# Nanoscale organization of rotavirus replication machineries

**Yasel Garcés Suárez[1†], Jose L Martínez[1†], David Torres Hernández[1,2], Haydee Olinca Hernández[1,2], Arianna Pérez-Delgado[1], Mayra Méndez[3], Christopher D Wood[1,2], Juan Manuel Rendon-Mancha[3], Daniela Silva-Ayala[1‡], Susana López[1], Adán Guerrero[1,2]\*, Carlos F Arias[1]\***

[1]Departamento de Genética del Desarrollo y Fisiología Molecular, Instituto de Biotecnología, Universidad Nacional Autónoma de México, Mexico City, Mexico; [2]Laboratorio Nacional de Microscopía Avanzada, Instituto de Biotecnología, Universidad Nacional Autónoma de México, Mexico City, Mexico; [3]Centro de Investigación en Ciencias, Instituto de Investigación en Ciencias Básicas y Aplicadas, Universidad Autónoma del Estado de Morelos, Cuernavaca, Mexico

**\*For correspondence:**
adanog@ibt.unam.mx (AáG);
arias@ibt.unam.mx (CFA)

[†]These authors contributed equally to this work

**Present address:** [‡]Harvard Medical School, Boston, United States

**Competing interests:** The authors declare that no competing interests exist.

**Abstract** Rotavirus genome replication and assembly take place in cytoplasmic electron dense inclusions termed viroplasms (VPs). Previous conventional optical microscopy studies observing the intracellular distribution of rotavirus proteins and their organization in VPs have lacked molecular-scale spatial resolution, due to inherent spatial resolution constraints. In this work we employed super-resolution microscopy to reveal the nanometric-scale organization of VPs formed during rotavirus infection, and quantitatively describe the structural organization of seven viral proteins within and around the VPs. The observed viral components are spatially organized as five concentric layers, in which NSP5 localizes at the center of the VPs, surrounded by a layer of NSP2 and NSP4 proteins, followed by an intermediate zone comprised of the VP1, VP2, VP6. In the outermost zone, we observed a ring of VP4 and finally a layer of VP7. These findings show that rotavirus VPs are highly organized organelles.
DOI: https://doi.org/10.7554/eLife.42906.001

## Introduction

Rotavirus is a non-enveloped virus composed of three concentric layers of proteins that enclose a genome constituted by eleven segments of double stranded RNA (dsRNA) that encode six structural proteins (VP1 to VP4, VP6 and VP7) and six non-structural proteins (NSP1 to NSP6). The inner layer is formed by dimers of VP2 that enclose the viral genome and small numbers of molecules of the viral RNA-dependent RNA polymerase, VP1, and the capping enzyme, VP3. This nucleoprotein complex constitutes the core of the virus, which is surrounded by an intermediate protein layer of trimers of VP6, to form double-layered particles (DLPs). The surface of the virion is occupied by two polypeptides, VP7, a glycoprotein, and VP4, which forms spikes that protrude from the VP7 shell (*Estes and Greenberg, 2013*). Replication of the rotavirus genome and assembly of DLPs take place in cytoplasmic electron dense inclusions termed viroplasms (VPs) (*Estes and Greenberg, 2013*). Once the double-shelled particles are assembled, they bud from the cytoplasmic VPs into the adjacent endoplasmic reticulum (ER). During this process, which is mediated by the interaction of DLPs with the ER transmembrane viral protein NSP4, the particles acquire a temporary lipid bilayer, modified by VP7 and NSP4, which after being removed in the lumen of the ER by an unknown mechanism, yields the mature triple-layered virions (*Estes and Greenberg, 2013*). It has been reported that VP4 is located between the VP and the ER membrane and it is incorporated into triple-layered

**eLife digest** Rotaviruses are small viruses that can infect cells in the intestine. They are responsible for most cases of severe infectious diarrhea, the most common cause of death among young children in developing countries. Controlling the spread of rotavirus infections is difficult, even with high levels of hygiene, so effective treatments are essential to curtail the virus' infections. Understanding how new rotaviral particles are made in infected cells is one of the first steps toward developing new therapies.

Once rotaviruses enter the cells, proteins from the virus and the cell aggregate into compact spheres called viroplasms to make new viral particles. Studying these viroplasms used to be difficult because they are too small to see with the resolution of standard microscopes. In recent years, advances in microscopy and mathematical methods have focused on breaking the existing resolution limits, leading to the development of super-resolution microscopy. This new technique has made it possible to study objects with sizes in the order of a billionth of a meter, known as nanoscopic structures, including viroplasms.

Garcés et al. use super-resolution microscopy to determine how viral proteins are arranged in the viroplasm and gain a better understanding of how the viruses are assembled. The images revealed that, in infected monkey kidney cells, rotavirus proteins inside the viroplasm form highly organized concentric layers. This arrangement is reliably repeated in viroplasms of different sizes, indicating that the organization of the proteins is likely set up when the viroplasm starts to form.

These findings make use of new microscopy, image analysis and statistical tools to study rotaviruses, providing a new framework to understand many aspects of rotaviral biology. Additionally, the result showing that proteins organize consistently in viroplasms is a first step towards understanding how the machinery that makes new rotaviruses works, which could lead to future treatments for severe infectious diarrhea.

DOI: https://doi.org/10.7554/eLife.42906.002

particles (TLPs) during the budding process and maturation of the virus particle inside the ER (*Estes and Greenberg, 2013*; *Navarro et al., 2016*).

The viral non-structural proteins NSP2 and NSP5 serve a nucleation role that is essential for the biogenesis of VPs (*Fabbretti et al., 1999*; *Silvestri et al., 2004*; *Vascotto et al., 2004*; *Campagna et al., 2005*). In addition to viral proteins and genomic dsRNA, cellular proteins such as ER chaperones (*Maruri-Avidal et al., 2008*), proteins associated with lipid droplets (*Cheung et al., 2010*), and ribonuclear proteins (*Dhillon et al., 2018*), have been shown to colocalize with VPs. Several studies have characterized the intracellular distribution of the rotavirus proteins (*González et al., 2000*; *Petrie et al., 1982*; *Petrie et al., 1984*; *Richardson et al., 1986*). Immunofluorescence studies, based upon epifluorescence or confocal microscopy, have described the viral proteins that conform the VPs, however the images are inherently diffraction-limited to a spatial resolution in the range of hundreds of nanometer, precluding the identification of the nanoscopic molecular scale organization of VPs (*González et al., 1998*; *González et al., 2000*; *Eichwald et al., 2004*; *López et al., 2005b*; *Criglar et al., 2014*; *Martin et al., 2011*; *Contin et al., 2010*). On the other hand, transmission electron microscopy (TEM) studies often provide images with nanometric resolution, nevertheless, immunoelectron microscopy is challenging when looking for the localization of more than a single protein (*Altenburg et al., 1980*; *Petrie et al., 1982*; *Petrie et al., 1984*). Over the past 15 years, a variety of super-resolution microscopy (SRM) techniques have been developed to observe subcellular structures beneath the diffraction limit of optical microscopes, with resolutions in the tens of nanometers (*Schnitzbauer et al., 2017*; *Deschout et al., 2014*; *Cox et al., 2011*). In this work, we determined the organization of rotaviral proteins within and around VPs through the 'Bayesian Blinking and Bleaching' (3B) SRM technique. We developed a segmentation algorithm to automatically analyze and quantify the relative distribution of seven viral proteins, and propose a model that describes their relative spatial distribution. Also, we present a dependency model that explains the relationship between the viral proteins. This work establishes a structural framework for VP organization that future mechanistic and functional studies must take into account, and establishes key methodologies for future investigations on this subject.

## Results

### Qualitative analysis of VP morphology and structure through SRM

Rotavirus VPs are complex signaling hubs composed of viral and cellular proteins, packed together with viral RNAs. By TEM, they roughly resemble circular electrodense structures whose internal components lack an obvious degree of spatial organization (*Altenburg et al., 1980*; *Eichwald et al., 2012*). In this work, we determined the relative spatial distribution of VPs components by immunofluorescence and SRM in MA104 cells infected with the rhesus rotavirus strain RRV at 6 hr post-infection (hpi), using protein-specific antibodies. Due to their important role as nucleating factors during VP biogenesis, we selected either NSP2 or NSP5 as spatial relative reference for the distribution of the VP1, VP2 and VP6 proteins. VPs were optically sectioned through total internal reflectence microscopy (TIRF), with an excitation depth of field restricted to 200 *nm* from the coverslip. This approach avoids excitation of fluorophores marking structural components located away from this plane, that is towards the inner cellular milieu. Additionally, NSP2 was also co-immunostained with the viral outer layer protein VP4 as well as with the ER resident proteins NSP4 and VP7, all of which have been reported to form separate ring-like structures that closely associate with VPs (*González et al., 2000*). In order to gain more insight into the morphogenesis of rotavirus, we analyzed the distribution of both VP7 monomers (VP7-Mon) and trimers (VP7-Tri) since this protein is assembled into virus particles in the latter form (*Kabcenell et al., 1988*). The nanoscale distribution of VPs was then analyzed through 3B-SRM, with improvements in the technique, developed in the present work, to solve nanoscopic structures ('Stochastic model fitted for 3B super resolution microscopy'Appendix 1). By different methods of analysis VPs exhibit roughly a circular shape (*Figure 1A–E*). However, unlike the diffraction-limited image (*Figure 1B*), in super-resolution microscopy structural details of VP are appreciated, like the different layer distributions of viral components with respect to NSP2 (*Figure 1C–E*). In addition to VPs, by diffraction-limited TIRF microscopy we

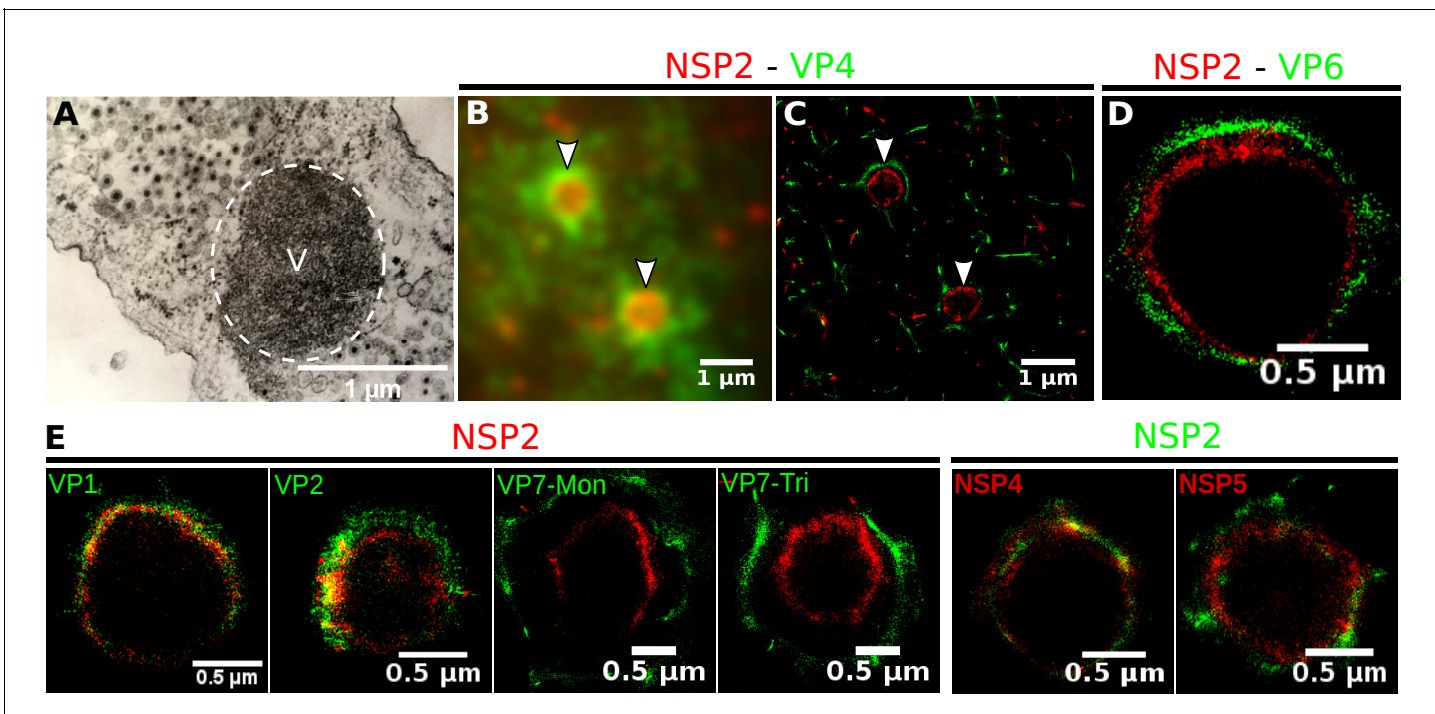

**Figure 1.** Relative distribution of viral components in rotavirus-VPs. RRV-infected MA104 cells (6hpi) were fixed and processed for transmission electron microscopy or immunofluorescence microscopy. (**A**) Transmission electron microscopy of a VP (identified by the dotted white ellipse). (**B**) Diffraction-limited image of VPs (white arrows). (**C**) 3B-SRM image reconstructed from B. (**D–E**) 3B-SRM images of individual VPs labeled with different antibodies (see Methods).

DOI: https://doi.org/10.7554/eLife.42906.003

detected in the cytoplasm several small and dispersed puncta of fluorescence (*Figure 1B*), and in these images it is also sometimes possible to differentiate the distribution of NSP2 from that of VP4, a closely viroplasm-associated viral protein (see also *González et al., 2000*); in this case, VP4 is detected as a ring-like structure that surrounds the VP. Nevertheless, the small size of the VPs effectively precludes measurement of component distribution for the majority of its structural elements, as their separation is below the spatial resolution of typical optical microscopes. In contrast, images obtained by 3B-SRM do allow the study of the relative distribution of the VP components (*Figure 1C–E*). In the case of SRM images of VP4 (*Figure 1C*), we observed that this protein forms a ring-like structure that does not colocalize with NSP2, and also ribbon-like projections that extend towards the cytoplasm, details that were not apparent in images captured with conventional fluorescence microscopy (*Figure 1B*). Additionally, we observed that the small puncta of proteins detected in the cytoplasm were in fact ribbon-like structures composed of various viral proteins that may represent different organization forms of the viroplasmic proteins (*Figure 1C*). In this regard, it is interesting to note that both NSP2 and VP4 have been reported to have at least two different intracellular distributions (*González et al., 2000*; *Nejmeddine et al., 2000*; *Criglar et al., 2014*). An examination of 3B-SRM images of VPs (*Figure 1C–E*) revealed that the viral components form ring like structures within the VPs and are arrayed as rather discrete concentric layers. As seen in *Figure 1C–E*, we find that although the structural proteins VP1, VP2 and VP6 partially overlap in position with NSP2, the bulk of the proteins form separate and distinct layers. Also, the monomeric as well as the trimeric forms of VP7 are clearly distinguished from NSP2, forming an outer ring. Of interest, the spatial distribution of NSP4 colocalized with that of NSP2, an unexpected result since, as mentioned, NSP4 is an ER integral membrane protein (see the Discussion section), and as such it was expected to colocalize with VP7 rather than with an internal viroplasmic protein (*Petrie et al., 1984*). With regard to NSP5, it was observed distributed inside the ring formed by NSP2 (*Figure 1E*).

## Quantitative characterization of VPs structure by a novel segmentation algorithm

A qualitative analysis of the distribution of the VP components through 3B-SRM suggested that these are arranged as concentric spherical shells; thus, we set out to quantitatively validate the circularity of the VP shape. For this, we developed a segmentation algorithm based on a least squares approach, which we called 'Viroplasm Direct Least Squares Fitting Circumference' (VP-DLSFC) (see 'Segmentation Algorithm' in Appendix 1), to measure the spatial distribution of the components within individual VPs by adjusting concentric circumferences. This method is automatic, deterministic, easy to implement, and has a linear computational complexity. The performance of VP-DLSFC was tested on approximately 40,000 'ground truth' (GT) synthetic images, showing a high robustness to noise and partial occlusion scenarios. Additionally, we compared our method with two other alternative methods (*Gander et al., 1994*), and our approach displayed an improved performance (see 'Algorithm Validation' in Appendix 1). Based on this new algorithm, we find that the mean radius of the NSP5 distribution was smaller than that of NSP2, suggesting that NSP5 is located in the innermost section, as a component of the core of VPs (*Figure 2A*). On the other hand, the distribution of the structural proteins VP1, VP2 and VP6 exhibit slightly larger mean radii than that of NSP2, and are thus primarily localized in a zone surrounding NSP2. Continuing further towards the outer regions of the VP, we observed a region occupied by the spike protein VP4. Finally, the two different forms of VP7 (VP7-Mon and VP7-Tri) were located together, close to the most external region of the VPs (*Figure 2A*). The distribution of the glycoprotein NSP4 showed a similar mean radius to that of NSP2 (around $0.4 \mu m$) suggesting, as described above, that these two proteins are located in the same structural layer of the VP (*Figure 2A*).

In order to confirm our preliminary observations and clarify the nanoscopic organization of the VPs, we evaluated the relative separation between NSP2 and each accompanying protein. Again, the results show a remarkable degree of organization in the structure of the VP (*Figure 2B*). As predicted from *Figure 2A*, we found that NSP5 is located in the internal part of the VP, in close proximity ($\approx 0.05 \mu m$) to the area occupied by proteins NSP2 and NSP4, which themselves show the closest association. After the NSP2-NSP4 region, VP6 occupies a middle region at $\approx 0.05 \mu m$ from NSP2, followed by the VP4 protein, which were located at a distance of $\approx 0.18 \mu m$. Finally, the VP7-Mon and VP7-Tri were situated at $\approx 0.38 \mu m$ from NSP2 (*Figure 2B*). A Mann-Whitney test showed that the

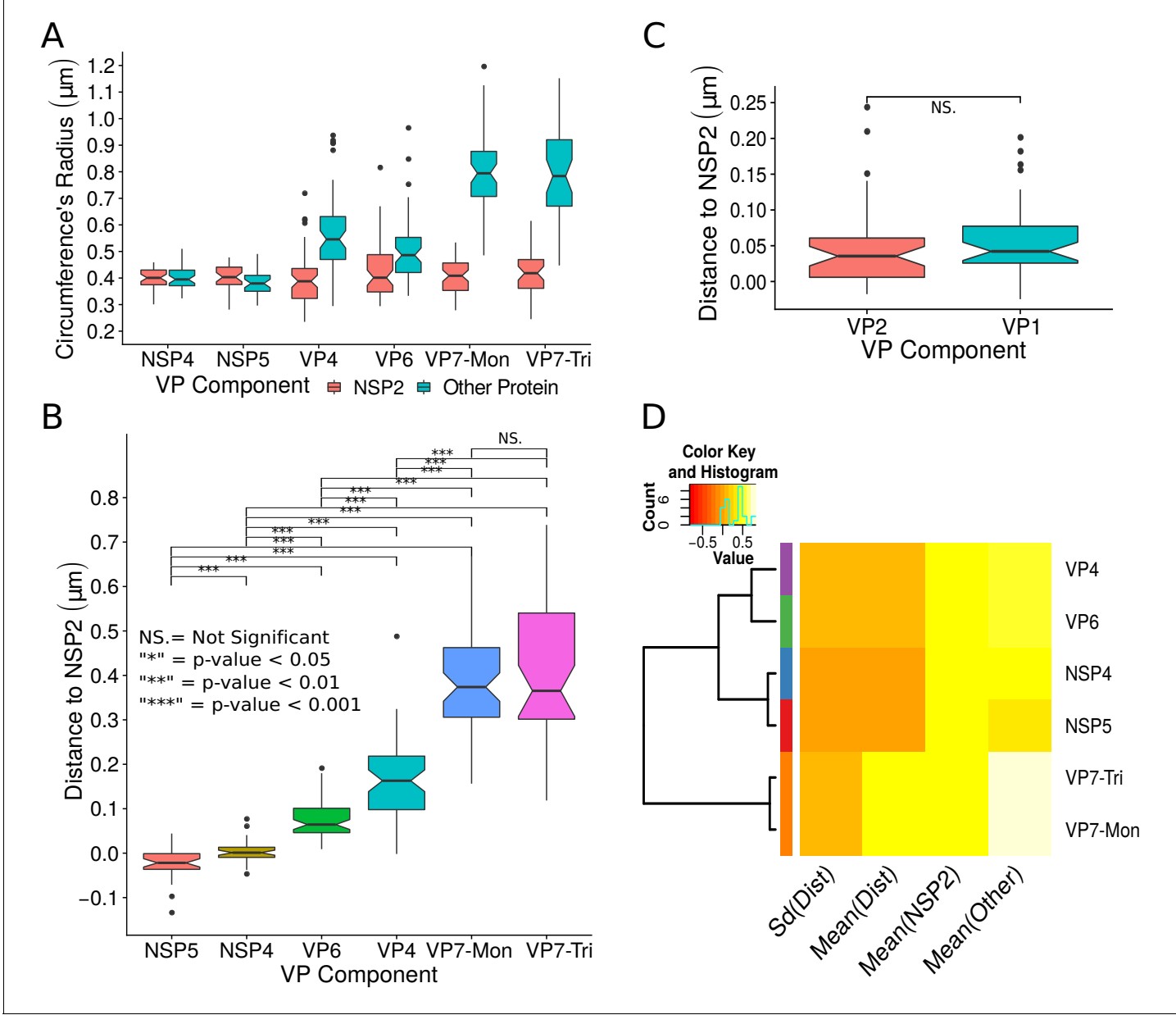

**Figure 2.** Exploratory analysis of the results obtained by the algorithm VPs-DLSFC. (**A**) Boxplot for the radius of the fitting circumferences. In each experimental condition we plot two boxes, the red box is for the radius of NSP2 (reference protein), and the blue box represents the radius of the accompanying VP components (names in x-axis). (**B**) Boxplot and results of the Mann-Whitney hypothesis test for the distance between each viral element and NSP2. Each combination of the Mann-Whitney test is linked by a line, and the result of the test it is above the line. Note that this test reports significant differences between the distribution of the distance to NSP2 of two different VP components. (**C**) Distance of VP1 and VP2 to NSP2 and result of the Mann-Whitney test. Because the distributions of NSP2 in combination with VP1 and VP2 are statistically different to the other NSP2 distributions (see *Appendix 1—figure 7B*), we show these two cases independently in this exploratory analysis. (**D**) Hierarchically clustered heatmap for the standard deviation of the distance to NSP2, the mean distance to NSP2, the mean radius of NSP2, and the mean radius of the accompanying protein layers, NSP5, NSP4, VP6, VP4, and VP7.

DOI: https://doi.org/10.7554/eLife.42906.004

distances of the various viral components in relation to NSP2 were significantly different (*Figure 2B*), suggesting that they are situated in specific areas of the VPs. The two forms of VP7 were located at the same distance to NSP2, suggesting that the formation of trimers of VP7 takes place at the ER membrane, where the VP7 monomers should also be located. Note that in *Figure 2B* the relative

distance of VP1 and VP2 to NSP2 was not included, since the radii obtained for NSP2 in these two combinations were significantly smaller than those found when it was determined in combination with the other VP components (see 'Supplementary Exploratory Analysis'). In addition to this, we found no significant differences between the distance of both VP1 and VP2 to NSP2 (*Figure 2C*). Nonetheless, based on the inferential analysis, we could place these two proteins in the same layer as VP6 (see below).

Next, through a hierarchical cluster analysis, we studied the relationship between the components of the VP, taking into account multiple variables at the same time, like the mean distance to NSP2 ['Mean(Dist)"], the standard deviation of the distance to NSP2 ['Std(Dist)"], the mean radius of NSP2 ['Mean(NSP2)"], and the radii of the other proteins ['Mean(Other)"] (*Figure 2D*). Note that the proteins within a cluster should be as similar as possible and proteins in one cluster should be as dissimilar as possible from proteins in another. Because our variables are related with the distance to NSP2 and the radii of the proteins, this is a no-parametric analysis that should provide evidence about the spatial distribution/order of the viral proteins into the VP. As we are considering the distance to NSP2, VP1 and VP2 were not included in this analysis. The first level of the hierarchical agglomerative cluster (*Figure 2D*, left) partitioned the VPs and the surrounding proteins in five clusters, composed by NSP4, NSP5, VP6, VP4 and {VP7-Mon, VP7-Tri}, which suggest that these five proteins compose different layers of the VP. The second agglomerative level merged into the same group the proteins NSP4 and NSP5, meanwhile VP6 and VP4 continue as independent clusters, which indicate that NSP5 and NSP4 are closer to each other than to VP6 and VP4 in the VP. In the third level, VP6 and VP4 are clustered in the same group, and as consequence are more related between them than with the others viral proteins. The subsequent groups in the clustering analysis indicate that VP7 remains as an independent layer with respect to the other proteins. Based on this analysis, the viral proteins seem to be highly organized, with VP7 conforming the most external layer, while NSP5, NSP4, VP6 and VP4 are distributed very close but as independent layers. The clusters between NSP5-NSP4 and VP6-VP4 suggest that these two pairs of proteins (in each cluster) conform continuous layers in the VP.

## The relative spatial organization of VPs is maintained regardless their size

The scatterplot between the radius of the spatial distribution of NSP2 (independent variable, x-axis) and the radius of the distribution of other viral components (response variable, y-axis) showed a strong linear relationship (*Figure 3A*). The distribution of NSP5 grows $0.87 \mu m$ for each $1 \mu m$ increase in the radius of NSP2 (slope interpretation), whereas the radius of the distribution of NSP4 increases $0.99 \mu m$ (*Figure 3B*). These findings indicate that NSP5 is distributed in a proportionally smaller region than NSP2 regardless of the absolute size of the VP, supporting our observation that NSP5 is a constituent of the core of the VP. Moreover, the fact that the increase in the radius of the fitted distribution of NSP4 is directly proportional to the same parameter measured for NSP2 supports the idea that these proteins are both constituents of a putative second layer. VP1, VP2 and VP6 exhibit similar slopes which diverge between 0.03 and 0.05 μm (*Figure 3B* and *Appendix 1—table 6*); thus, these results confirm that VP1, VP2 and VP6 are components of the same layer in the VPs which, from the data in *Figure 3*, is located just after the layer of NSP2 and NSP4. Finally, as noted in our quantitative analysis, VP4 and VP7 form consecutive external layers with a slope of 1.39 and 1.94 μm, respectively (*Figure 3B* and *Appendix 1—table 6*). These findings indicate that the spatial distribution of the viral components in the VPs and in the surrounding areas is conserved regardless of their absolute size, and also form the basis of a predictive model, where, for a given radius of distribution of NSP2, it is possible to predict the radii of the remaining VP components (NSP5, NSP4, VP1, VP2, VP6) and of VP4 and VP7 proteins. This predictive model is available as a web app at https://yasel.shinyapps.io/Nanoscale_organization_of_rotavirus_replication_machineries/. The mathematical details and the residual analysis that validate these linear models are available in the Appendix 1, section 'Linear dependency between the viral components', *Appendix 1—table 6* and *Appendix 1—figure 9*.

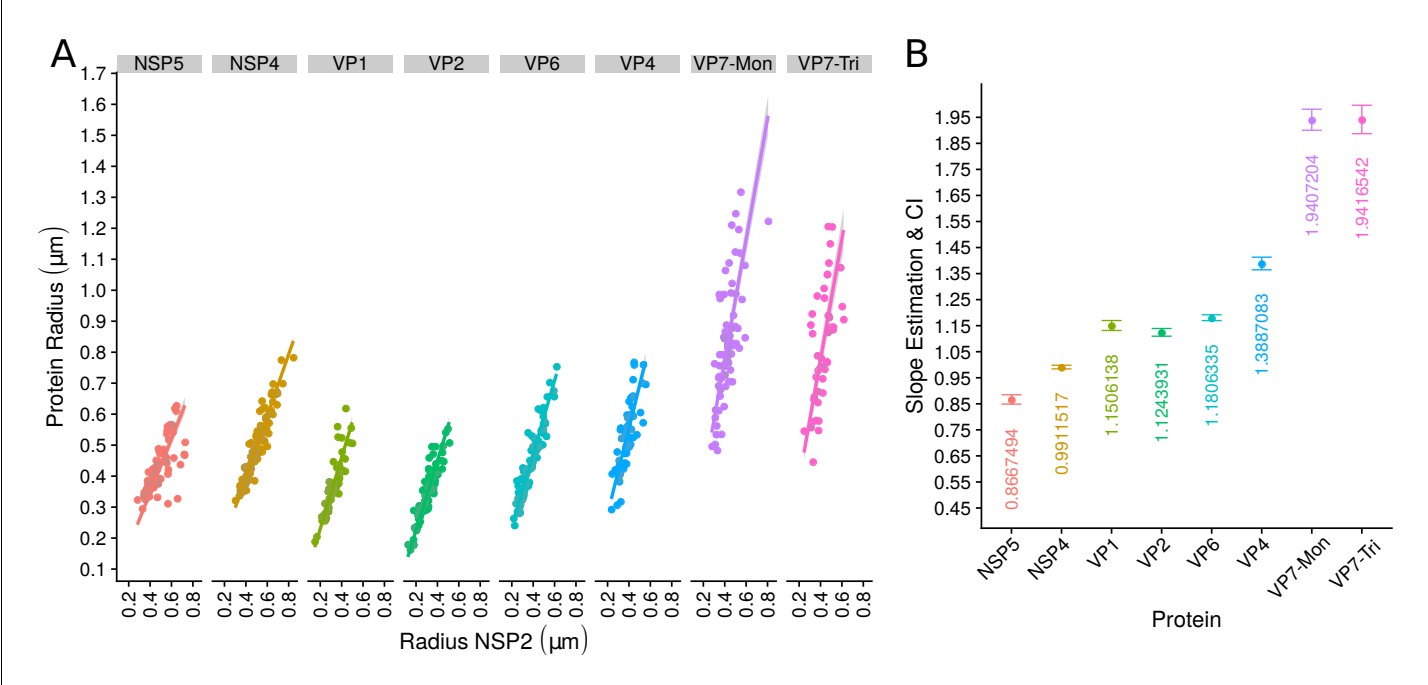

**Figure 3.** The organization of VPs scales with its size. (**A**) Simple linear regression analyses for each component combination (eight subpanels). In all subpanels, the x-axis represents the radius of the distributions of NSP2, and the y-axis the radius of the distribution of the accompanying VP component. The 95% confidence interval, marked in grey, is imperceptible due to goodness of fit of the linear regression (solid line). (**B**) Slope and confidence interval for each linear regression model (dependent variables in x-axis). The slopes values were shown under each confidence interval.

DOI: https://doi.org/10.7554/eLife.42906.005

## The structural organization of VPs is independent of the reference protein chosen for pairwise comparison

In order to confirm the observed structural organization of VPs, we analyzed two more experimental conditions in which we chose a different reference protein for pairwise comparisons. The first was based on the distribution of NSP5 and its comparison with the relative localizations of VP6 and VP4, and the second considered NSP4 as the reference protein to compare with the distribution of VP6. We found that both analyses produced an identical structural organization for the VPs, with a comparative localization error of approximately $0.05 \mu m$ between models (close to the effective resolution limit of the 3B algorithm; see 'NSP5 and NSP4 as reference proteins' in Appendix 1). An extensive quantitative validation regarding the congruence between the NSP2, NSP5 and NSP4 models is available in the Appendix 1.

Based on our extensive quantitative, descriptive and inferential statistical analyses, we propose that the VP and the surrounding viral proteins form an ordered biological structure composed of at least five concentric layers organized as depicted in *Figure 4*. In this structure, NSP5 constitutes the innermost layer, followed by a {NSP2-NSP4} layer. Then, there is a layer composed by {VP1-VP2-VP6} and two consecutive external layers formed by VP4 and VP7. The different layers of proteins are most likely highly porous to allow the entry of positive-sense single-stranded viral RNA (+RNA) during genome replication and also of the antibodies used for VP staining.

## Discussion

VPs have been previously studied using electron and fluorescence microscopy, however, due to the limited resolution of classic fluorescence microscopy techniques, and the difficulty of analysis of immunoelectron microscopy, the existence of any complex structural organization of the viral elements inside the VPs has not been reported. In recent years, the development of SRM has facilitated research into the nanoscale organization of a diverse range of cellular structures (*Grant et al., 2018*;

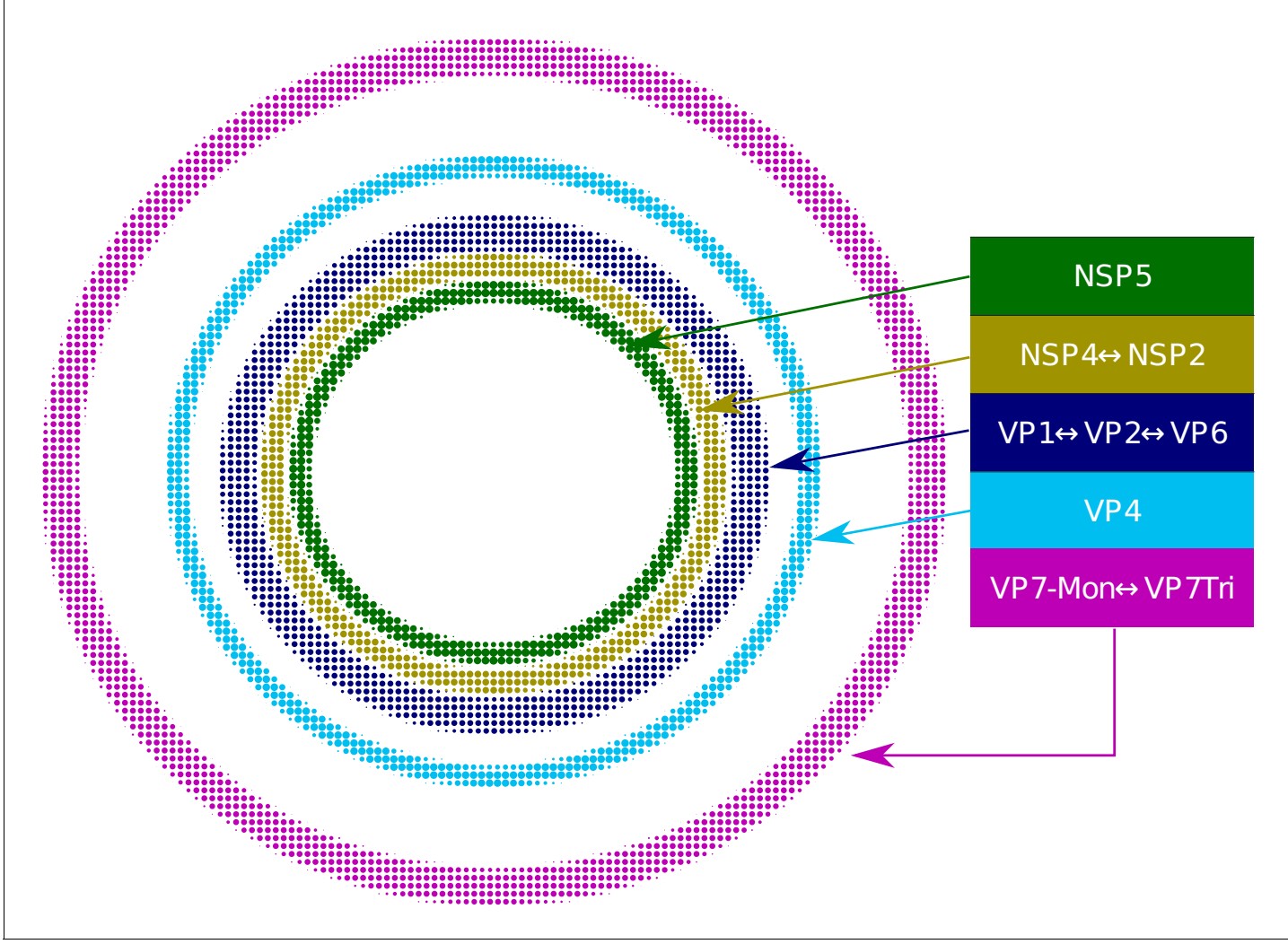

**Figure 4.** Relative structural distribution of VP components. The radii of the circumferences maintain the relative values determined for the different VP layers.

DOI: https://doi.org/10.7554/eLife.42906.006

*Reznikov et al., 2018*), however, until now SRM had not been applied to study the replication cycle of rotavirus. In this work, thanks to the use of the 3B SRM algorithm, we visualized and determined quantitatively the location of several viroplasmic proteins, leading us to propose a detailed model of the VP that should be of great value for understanding virus morphogenesis.

Other SRM algorithms had been used to study the organization of viral and cellular structures showing concentric arrangements, as those proposed by *Laine et al. (2015)* and *Manetsberger et al. (2015)*. The main similarities between those studies and our approach is the use of conics, such as circles or ellipses, to fitting structures showing concentric organization. The method provided by *Manetsberger et al. (2015)* could actually be implemented to analyze our data set, which as outcome will produce similar results. This method could also provide information about the degree of asymmetry within the VP, which may be valuable to establish functional relationships between the protein distribution belts that shape these intriguing structures. The selection of the 3B SRM algorithm over other super-resolution approaches was based on the fact that this method allows to deal with samples with high density of labeling, obtaining data with a reasonable resolution, although at the cost of higher computational effort.

The quantification of the viral protein distribution within the VPs was possible thanks to a novel segmentation algorithm (VPs-DLSFC) that was proven to be robust and efficient in noisy and partial

occlusion scenarios. The manual pre-segmentation step of this algorithm was necessary in our case because we did not want to introduce any bias in the isolation of the VPs through an automatic approach. Setting aside the manual pre-segmentation step, the VPs-DLSFC algorithm is automatic, deterministic, non-iterative and has a linear computational complexity.

Previous reports have suggested that VPs have a spherical-like structure (*Eichwald et al., 2004*; *Cabral-Romero and Padilla-Noriega, 2006*; *Campagna et al., 2013*); in this study we confirmed this suggestion by comparing the VPs-DLSFC approach with a similar approach based on an ellipse adjustment (*Garcés et al., 2016*). The results showed no significant statistical differences between these two models, and as consequence we can confidently model the structure of the VPs as a circumference. We also ratified that the center displacement of the circumferences that adjust two paired proteins are not statistically different.

Our study indicates that the viral components in the VPs, as well as VP4 and VP7, are arranged as largely discrete concentric layers (note that we are describing the structure of viroplasms, not of virus particles). This organization, however, does not preclude the interaction among the VP components proposed in this model as being located in separate layers since, for instance, NSP5 has been shown by different biochemical methods to interact with NSP2 (*Eichwald et al., 2004*; *Poncet et al., 1997*; *Afrikanova et al., 1998*; *Jiang et al., 2006*), VP1 (*Afrikanova et al., 1998*) and VP2 (*Berois et al., 2003*). In this regard, based on the super resolution microscopy images, it seems clear that there is some overlapping between different protein layers, as is the case for NSP2 and NSP5 in *Figure 1E*, but also between NSP2 and VP1, VP2, VP6 and NSP4 in *Figure 1*. These general overlapping zones between different proteins most likely are relevant for coordinating the genome replication and virion assembly, as suggested. Of interest, we observed the presence of protein projections ('spike-like') from different viral shells that could also contribute to the interaction of proteins mapped to different layers (*Appendix 1—figure 8*). Although our present analysis is limited to a general characterization of the spatial distribution of the viral proteins within VPs, and not to understand specific details about the interactions between proteins in different layers, it could be used as departure point to analyze these interactions. Taking as initial solution the result of the algorithm VPs-DLSFC and the SRM image, it is possible to employ other segmentation approaches, like deformable/active contours (snakes) (*Kass et al., 1988*), level-set (*Osher and Fedkiw, 2003*), or region growing methods (*Mehnert and Jackway, 1997*; *Synthuja et al., 2012*), to evolve the circular contour and fit precisely the spatial distribution of the viral proteins. Then, establishing a polar coordinate system in the VP, and considering the results of both segmentation algorithms, it would be possible to quantify the radial angle in which a spike from the central distribution of a viral protein that interacts with a different protein exists. It would be also possible to determine how strong these interactions are (intersection between two segmentation curves), and to study whether the spikes are randomly distributed between layers or a specific pattern in the connection between different protein layers exists. In the latter case, this could allow us to explore whether these patterns influence the assembly of the virus-like particles or only provide a skeleton that maintain the structure of the VP. The results obtained could also be used to study topological changes that the VP might experience at different times post infection, and associate these changes with maturation of the subviral particles. In this regard, in preliminary experiments carried out at three hpi, the viral proteins in he VP have been found to have a similar 'layered' organization as shown for the mature VP at six hpi (data not shown). This observation indicates that this organization is already present when the formation of viral particles has not yet taken place, suggesting that it might be important for the assembly of DLPs within VPs. In an additional application, SRM could also be used to observe the assembly of the virus particles and the interactions that may occur of these particles in different layers of the VP. Nevertheless, to develop this idea it would be important to establish an experimental protocol to observe the viral particles during the early stages of the assembly process, to distinguish simultaneously the layers of the viroplasm and the viral particles, and to collect the SRM images with a very short acquisition rate and a very high resolution (25–30 nm), which makes this experimental plan a challenge.

Previous studies based on conventional microscopy techniques have reported that NSP5 and NSP2 colocalize (*González et al., 2000*; *Eichwald et al., 2004*; *Fabbretti et al., 1999*); in contrast, we found that although NSP5 and NSP2 are located in close proximity, their positions in the VP were separable. This difference is attributable to the increased spatial resolution in the final image created by the super-resolution techniques employed in our study. Here, NSP5 was found to represent the innermost layer of the VPs, suggesting that this protein might serve as the core scaffold

upon which the subsequent viroplasmic proteins are assembled to form the VPs. This finding contrasts with a report by *Eichwald et al. (2004)*, who described that NSP5 locates to a region external to NSP2. In addition to the superior spatial resolution obtainable through 3B-SRM, compared to the traditional confocal microscopy employed in the previous report, the difference might be due to the fact that in our study we characterized the endogenous structures produced during virus replication, while Eichwald et al. characterized VP-like structures formed by transiently expressed proteins fused to GFP.

Immediately outside the NSP5 core, we observed a layer composed of NSP2 and NSP4 proteins. The finding that NSP4 is located in the inner part of VPs was unexpected, since it is known that NSP4 is an integral membrane protein of the ER and since it has been reported that functions as a receptor for the new DLPs located at the periphery of the VPs, during their budding towards the lumen of the ER (*Chasey, 1980*; *Petrie et al., 1982*; *Petrie et al., 1984*; *Au et al., 1989*). Furthermore, it has been shown that NSP4 associates with VP4 and VP7 to form a hetero-oligomeric complex that could be involved in the last steps of rotavirus morphogenesis (*Maass and Atkinson, 1990*). Based on these findings, NSP4 was expected to locate close to VP4 and VP7, in the surroundings of the VP. On the other hand, and in line with our observations, previous confocal microscopy studies have shown that a portion of NSP4 also shows a limited colocalization with NSP2 (*González et al., 2000*).

The dual location of NSP4 as an integral glycoprotein of the ER membrane and as internal to VPs, as our results indicate, is not easy to reconcile; however, in a previous work it was suggested that there are three pools of intracellular NSP4 molecules. The first pool is represented by NSP4 localized in the ER, a second minor pool localized in the ERGIC compartment, and the third pool distributed in cytoplasmic vesicular structures associated with the autosomal marker LC3 (*Berkova et al., 2006*). Furthermore, in that work the authors suggested that NSP4 and autophagic marker LC3-positive vesicles may serve as a lipid membrane scaffold for the formation of large VPs by recruiting early VPs or VP-like structures formed by NSP2 and NSP5 (*Berkova et al., 2006*). This observation is in line with our model that NSP4 lies in an internal protein shell within VPs.

An additional, and very interesting possibility to explain the internal location of NSP4 in VPs is the hypothesis that VP morphogenesis occurs on the surface of lipid droplets (LDs) (*Cheung et al., 2010*). In that work, it was proposed that LDs serve as a platform to which NSP2 and NSP5 proteins attach to form VP-like structures; NSP2 octamers, in turn, associate with the viral polymerase VP1 and rotavirus +RNAs. The assorted RNA complex containing NSP2, VP1, the capping enzyme VP3 and viral +ssRNA is predicted to nucleate VP2 core assembly. In this model, core assembly results in the displacement of +RNA-bound NSP2 octamers, while VP1 within new formed cores direct dsRNA synthesis, using +RNAs as templates (*Cheung et al., 2010*; *Borodavka et al., 2017*; *Borodavka et al., 2018*). These events are followed by incorporation of the middle virus capsid protein VP6 to form DLPs. At some stage, these assemblies become VPs containing cores and DLPs and may lose some or all of their lipids (*Cheung et al., 2010*). In this regard, it is important to have in mind that the currently accepted model for the LD biogenesis is that neutral lipids are synthesized between the leaflets of the ER membrane, and the mature LD is then thought to bud from the ER membrane to form an independent organelle that is contained within a limiting monolayer of phospholipids and associated LD proteins (*Walther and Farese, 2012*). Thus, during budding of the LDs from the ER membrane they could take along rotavirus NSP4 (topologically oriented towards the cell's cytoplasm) which could help as a scaffold on the surface of LDs for the assembly of other rotavirus viroplasmic proteins, localizing then to the interior of VPs.

Further support for our model of localization of at least one pool of NSP4 molecules inside of the VPs is the observation that knocking-down the expression of NSP4 by RNA interference significantly reduces the number and size of VPs present in the cell, as well as the production of DLPs (*López et al., 2005a*). That study also showed that during RNAi inhibition of NSP4 expression the NSP2 and NSP5 proteins maintained an intracellular distribution restricted to VPs, while the VP2, VP4, VP6 and VP7 proteins failed to locate to VPs. Based on these observations, it is tempting to suggest that, in addition to the role NSP4 has on the budding of DLPs into the ER lumen, it may also play an important role as a regulator of VP assembly.

After the NSP2/NSP4 layer, we observed a middle zone composed of the structural proteins VP1, VP2 and VP6. Their location in the same zone is expected given their close association in the assembled DLPs (*Estes and Greenberg, 2013*). Also, the fact that VP1, VP2 and VP6 form a complex with

NSP2 that has replicase activity (*Aponte et al., 1996*), suggests that the production of new DLPs could take place in this zone of the VP.

Finally, we found that VP4 and VP7 conform independent layers just external to the viroplasmic proteins. The position of these two proteins agrees with the proposed model of rotavirus morphogenesis in which VP4 is assembled first on DLPs, and subsequently VP7 binds the particles and locks VP4 in place (*Trask and Dormitzer, 2006*). Furthermore, the fact that VP7-Mon and VP7-Tri occupied the same layer in our model indicates that in the ER sites into which the DLPs bud, VP7 is already organized as trimers, which are subsequently assembled into the virus particles. Of interest, VP4 has been reported to exist in two different forms in infected cells. One of them is associated with microtubules (*Nejmeddine et al., 2000*), while the other one has been reported to be found between the VP and the ER membrane (*González et al., 2000*). In this regard, based on our findings, we suggest that the latter form of VP4 can be actually considered as an integral component of the VP. Since several studies have found the presence of different cellular proteins and lipids in association to VPs (*Maruri-Avidal et al., 2008*; *Cheung et al., 2010*; *Dhillon et al., 2018*), it will be interesting to study the relative localization of this components using the methodologies described here.

# Materials and methods

## Key resources table

| Reagent type (species) or resource | Designation | Source or reference | Identifiers | Additional information |
|---|---|---|---|---|
| Virus strain (Rhesus rotavirus) | RRV | Harry B. Greenberg, Stanford University. | | |
| Cell line (Cercopithecus aethiops) | MA014 cells | American Type Culture Collection | ATCC:CRL-2378.1; RRID:CVCL_3846 | |
| Antibody | Mouse monoclonal antibody 3A8 | Harry B. Greenberg, Stanford University. | | IF (1:1000) |
| Antibody | Mouse monoclonal antibody 2G4 | Harry B. Greenberg, Stanford University. PMID: 2431540 | | IF (1:1000) |
| Antibody | Mouse monoclonal antibody 255/60 | Harry B. Greenberg, Stanford University. PMID: 6185436 | | IF (1:1000) |
| Antibody | Mouse monoclonal antibody M60 | Harry B. Greenberg, Stanford University. PMID: 2431540 | | IF (1:2000) |
| Antibody | Mouse monoclonal antibody 159 | Harry B. Greenberg, Stanford University. PMID: 2431540 | | IF (1:2000) |
| Antibody | Mouse polyclonal antibody VP1 | Our Laboratory. | RRID:AB_2802095 | IF (1:500) |
| Antibody | Mouse polyclonal antibody NSP2 | Our Laboratory. PMID: 9645203 | RRID:AB_2802096 | IF (1:100) |
| Antibody | Rabbit polyclonal antibody NSP2 | Our Laboratory. PMID: 9645203 | RRID:AB_2802097 | IF (1:2000) |
| Antibody | Rabbit polyclonal antibody NSP4 | Our Laboratory. PMID: 18385250 | RRID:AB_2802094 | IF (1:1000) |
| Antibody | Rabbit polyclonal antibody NSP5 | Our Laboratory. PMID: 9645203 | RRID:AB_2802098 | IF (1:2000) |

*Continued on next page*

*Continued*

| Reagent type (species) or resource | Designation | Source or reference | Identifiers | Additional information |
|---|---|---|---|---|
| Software, algorithm | R | *R Development Core Team, 2017.* R: A language and environment for statistical computing. R Foundation for Statistical Computing, Vienna, Austria. URL https://www.r-project.org/ | RRID:SCR_001905 | Version 3.4.4 (2018-03-15) |
| Software, algorithm | Matlab | MATLAB and Statistics Toolbox Release 2018b, The MathWorks, Inc, Natick, Massachusetts, United States. | RRID:SCR_001622 | |
| Software, algorithm | Fiji | PMID:22743772 | RRID:SCR_002285 | |
| Software, algorithm | VP-DLSFC | This paper | | See 'Segmentation Algorithm' in Appendix 1. |

## Cell and virus

The rhesus monkey kidney epithelial cell line MA104 (ATCC) was grown in Dulbecco's Modified Eagle Medium-Reduced Serum (DMEM-RS) (Thermo-Scientific HyClone, Logan, UT) supplemented with 5% heat-inactivated fetal bovine serum (FBS) (Biowest, Kansas City, MO) at 37°C in a 5% $CO_2$ atmosphere. The cells were confirmed to be free of mycoplasm by testing with the INTRON Mycoplasma PCR Detection Kit (#25234). Rhesus rotavirus (RRV) was obtained from H. B. Greenberg (Stanford University, Stanford, Calif.) and propagated in MA104 cells as described previously (*Pando et al., 2002*). Prior to infection, RRV was activated with trypsin (10 µg/ml; Gibco, Life Technologies, Carlsbad, CA) for 30 min at 37°C.

## Antibodies

Monoclonal antibodies (MAbs) to VP2(3A8), VP4 (2G4), VP6 (255/60), VP7 (60) and VP7 (159) were kindly provided by H. B. Greenberg (Stanford University, Stanford, CA) (*Shaw et al., 1986*; *Greenberg et al., 1983*). The rabbit polyclonal sera to NSP2, NSP4 and NSP5, and the mouse polyclonal serum to NSP2 were produced in our laboratory (*González et al., 1998*). The hyperimmune serum to NSP4 (C-239) was generated in our laboratory by immunizing New Zealand white rabbits with a recombinant protein expressed in *E. coli* with a histidine-tail, representing the carboxy-terminal end (amino acids 120 to 175) of the rhesus rotavirus RRV NSP4 protein; see also *Maruri-Avidal et al. (2008)*, in which this serum was used. The hyperimmune serum to VP1 was also generated in our laboratory by immunizing BALB/c mice with a recombinant protein expressed in *E. coli* with a histidine-tail, representing amino acids 227 to 539 of the rhesus rotavirus RRV VP1 protein. Goat anti-mouse Alexa-488- and Goat anti-rabbit Alexa-568-conjugated secondary antibodies were purchased from Molecular Probes (Eugene, Oreg.).

## Immunofluorescence

MA104 cells grown on glass coverslips were infected with rotavirus RRV at a multiplicity of infection (MOI) of 1. Six hours post infection, the cells were fixed with and processed for immunofluorescence as described (*Silva-Ayala et al., 2013*). Finally, the coverslips were mounted onto the center of glass slides with storm solution (1.5% glucose oxidase $+100$ mM $\beta$-mercaptoethanol) to induce the blinking of the fluorophores (*Dempsey et al., 2011*; *Heilemann et al., 2009*).

## Transmission electron microscopy

Cells grown in 75-$cm^2$ flasks were infected with rotavirus RRV at an MOI of 3 as described above. Six hours postinfection the cells were fixed in 2.5% glutaraldehyde-0.1 M cacodylate (pH 7.2), postfixed with 1% osmium tetroxide, and embedded in Epon 812 resin. The ultrathin sections obtained were

stained with 2% uranyl acetate-1% lead citrate (Reynolds mix). The grids were examined with a Zeiss EM-900 electron microscope at 80 kV.

## Set up of the optical microscope

All super-resolution imaging measurements were performed on an Olympus IX-81 inverted microscope configured for total internal reflection fluorescence (TIRF) excitation (Olympus, cellTIRFTM Illuminator). The critical angle was set up such that the evanescence field had a penetration depth of ~200 nm (Xcellence software v1.2, Olympus soft imaging solution GMBH). The samples were continuously illuminated using excitation sources depending on the fluorophore in use. Alexa Fluor 488 and Alexa Fluor 568 dyes were excited with a 488 nm or 568 nm diode-pumped solid-state laser, respectively. Beam selection and modulation of laser intensities were controlled via Xcellence software v.1.2. A full multiband laser cube set was used to discriminate the selected light sources (LF 405/488/561/635 A-OMF, Bright Line; Semrock). Fluorescence was collected using an Olympus UApo N $100x/1.49$ numerical aperture, oil-immersion objective lens, with an extra 1.6x intermediate magnification lens. All movies were recorded onto a $128 \times 128$-pixel region of an electron-multiplying charge coupled device (EMCCD) camera (iXon 897, Model No: DU-897E-CS0-#BV; Andor) at 100 nm per pixel, and within a 50 ms interval (300 images per fluorescent excitation).

## Bayesian analysis of the blinking and bleaching

Sub-diffraction images were derived from the Bayesian analysis of the stochastic Blinking and Bleaching of Alexa Fluor 488 dye (*Cox et al., 2011*). For each super-resolution reconstruction, 300 images were acquired at 20 frames per second with an exposure time of 50 ms at full laser power, spreading the bleaching of the sample over the length of the entire acquisition time. The maximum laser power coming out of the optical fiber measured at the back focal plane of the objective lens, for the 488 nm laser line, was 23.1 mW. The image sequences were analyzed with the 3B algorithm considering a pixel size of 100 nm and a full width half maximum of the point spread function of 270 nm (for Alexa Fluor 488), measured experimentally with 0.17 µm fluorescent beads (PS-SpeckTM Microscope Point Source Kit, Molecular Probes, Inc). All other parameters were set up using default values. The 3B analysis was run over 200 iterations, as recommended by the authors in *Cox et al. (2011)*, and the final super-resolution reconstruction was created at a pixel size of 10 nm with the ImageJ plugin for 3B analysis (*Rosten et al., 2013*), using parallel computing as described in *Hernández et al. (2016)*. The resolution increase observed in our imaging set up by 3B analysis was up to five times below the Abbe's limit (~50 nm). The resolution provided by 3B was improved by computing the photo-physical properties of Alexa Fluor 488, and Alexa Fluor 568 dyes, which were provided to 3B algorithm, as an input parameter which encompass the probability transition matrix between fluorophore's states. The method was validated with 40 nm gattapaint nanorules (PAINT 40RG, gattaquant, Inc) labeled with ATTO 655/ATTO 542 dyes (see '3B Algorithm' in Appendix 1).

## Code and statistical analysis

The segmentation algorithm (VPs-DLSFC) was developed in Matlab R2018a (9.4.0.813654) software. A detailed explanation of each the developed methods is available in Appendix 1. Statistical analysis were performed using R version 3.4.4 (2018-03-15) software. All the codes are available at https://github.com/Yasel88/Nanoscale_organization_of_rotavirus_replication_machineries (*Garcés Suárez, 2019*; copy archived at https://github.com/elifesciences-publications/Nanoscale_organization_of_rotavirus_replication_machineries).

## Acknowledgements

YG received a postdoctoral fellowship from DGAPA-UNAM at the Institute of Biotechnology (IBt-UNAM). AG thanks DGTIC-UNAM for generous computing time on the Miztli supercomputer (Grant numbers: SC15-1-IR-89; SC16-1-IR-102). JLM and DTH are recipients of scholarships from CONACyT. HOH received a grant from the Programa de Apoyo a Proyectos de Investigación e Innovación Tecnológica (PAPIIT-UNAM), IN202312. Microscopy equipment was provided and maintained through CONACYT grants 123007, 232708, 260541, 280487, 293624. AG thanks CONACyT (No. 252213) and DGAPA-PAPIIT (No. IA202417), SL and CFA thank DGAPA-PAPIIT grant IG200317 for funding. We are thankful to IBt-UNAM for providing access to the computer cluster and to Jerome

Verleyen for his support while using it. We are also thankful to Arturo Pimentel, Andrés Saralegui and Xochitl Alvarado from LNMA-UNAM for their helpful discussions. The funders had no role in study design, data collection and interpretation, or the decision to submit the work for publication.

## Additional information

### Funding

| Funder | Grant reference number | Author |
|---|---|---|
| Dirección General de Asuntos del Personal Académico, Universidad Nacional Autónoma de México | IG200317 | Susana López Carlos F Arias |
| Dirección General de Asuntos del Personal Académico, Universidad Nacional Autónoma de México | IA202417 | Adán Guerrero |
| Universidad Nacional Autónoma de México | SC15-1-IR-89 | Adán Guerrero |
| Consejo Nacional de Ciencia y Tecnología | 252213 | Adán Guerrero |
| Dirección General de Asuntos del Personal Académico, Universidad Nacional Autónoma de México | IN202312 | Haydee Olinca Hernández |
| Universidad Nacional Autónoma de México | SC16-1-IR-102 | Adán Guerrero |

The funders had no role in study design, data collection and interpretation, or the decision to submit the work for publication.

### Author contributions

Yasel Garcés Suárez, Data curation, Software, Formal analysis, Supervision, Validation, Investigation, Visualization, Methodology, Writing—original draft, Project administration, Writing—review and editing; Jose L Martínez, Conceptualization, Supervision, Investigation, Visualization, Methodology, Writing—original draft, Project administration, Writing—review and editing; David Torres Hernández, Investigation, Visualization, Methodology; Haydee Olinca Hernández, Software, Validation, Investigation, Visualization, Methodology; Arianna Pérez-Delgado, Daniela Silva-Ayala, Investigation, Methodology; Mayra Méndez, Software, Validation, Investigation, Writing—review and editing; Christopher D Wood, Investigation, Methodology, Writing—review and editing; Juan Manuel Rendon-Mancha, Supervision, Investigation; Susana López, Conceptualization, Supervision, Funding acquisition, Investigation, Methodology, Project administration, Writing—review and editing; Adán Guerrero, Conceptualization, Software, Supervision, Validation, Investigation, Methodology, Writing—review and editing; Carlos F Arias, Conceptualization, Supervision, Funding acquisition, Investigation, Methodology, Writing—original draft, Project administration, Writing—review and editing

### Author ORCIDs

Yasel Garcés Suárez (iD) https://orcid.org/0000-0001-8401-8352
Jose L Martínez (iD) https://orcid.org/0000-0002-0529-7558
Juan Manuel Rendon-Mancha (iD) http://orcid.org/0000-0002-9629-7050
Adán Guerrero (iD) https://orcid.org/0000-0002-4389-5516
Carlos F Arias (iD) https://orcid.org/0000-0003-3130-4501

### Decision letter and Author response

Decision letter https://doi.org/10.7554/eLife.42906.040
Author response https://doi.org/10.7554/eLife.42906.041

# Additional files

## Supplementary files

• Transparent reporting form
DOI: https://doi.org/10.7554/eLife.42906.007

## Data availability

All data generated or analysed during this study are included in the manuscript and supporting files.

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

# Appendix 1

DOI: https://doi.org/10.7554/eLife.42906.008

This appendix is divided in seven sections. In –Segmentation algorithm–, we discuss the mathematical details of the algorithm 'Viroplasm Direct Least Square Fitting Circumference' (VPs-DLSFC). In Section –Algorithm validation–, we compare the algorithm VPs-DLSFC with other two classic methods for the adjustment of circumferences; this analysis was done thanks to the use of approximately 40000 synthetic 'ground truth' images. In Section – Model Considerations–, we prove that the distribution of the studied viral elements follow a concentric circumference spatial distribution. Sections –Exploratory Analysis– and – Linear Regression Model– contains a supplementary exploratory analysis about the spatial distribution of the viral elements, and details about the results and the residual error analysis of the linear regression models, respectively. In Section –NSP5 and NSP4 as reference proteins– we developed a similar study that was made with NSP2, but considering NSP5 and NSP4 as reference proteins. Also, we shown that there is no statistically significative differences between the distributions of the viral elements when we changed the reference protein. Finally, in Section –3B Algorithm– we explain the fitting model for the 3B algorithm where the transition matrix is modeled by ordinary differential equations (ODE).

## Segmentation algorithm

The use of primitive models for the segmentation of the SRM images has many benefits, like computational efficiency, simple programmation, and understandable information about the objects that were segmented. In this regard, we developed a simple method for fitting a circumference to scattered data, which we called 'Direct Least Squares Fitting Circumference' (DLSFC). This approach considers basic mathematical analysis tools for the computation of the extreme value of a continuous function with N variables.

## Direct least squares fitting circumference (DLSFC)

The spatial distribution of a viral proteins into the VP can be seen as a set of points $P = \left\{ (x_i, \ y_i) | i = \overline{1:N} \right\} \in \mathbb{R}^2$ (scattered data in the plane). Taking into account the implicit equation of a circumference $(x - x_c)^2 + (y - y_c)^2 = r^2$, where $(x_c, y_c)$ is the center, and $r$ is the radius. The problem of adjust a circumference to $P$, is a optimization problem given by:

$$\min_{x_c, y_c, r} \sum_{i=1}^{N} \left( (x_i - x_c)^2 + (y_i - y_c)^2 - r^2 \right)^2. \tag{1}$$

The center of mass of $P$ is given by:

$$(\bar{x}, \bar{y}) = \left( \frac{1}{N} \sum_{i=1}^{N} x_i, \ \frac{1}{N} \sum_{i=1}^{N} y_i \right). \tag{2}$$

Then, considerering **Equation (2)**, the problem (**Equation (1)**) can be rewritten as:

$$\min_{u_c, v_c, \alpha} \sum_{i=1}^{N} \left( (u_i - u_c)^2 + (v_i - v_c)^2 - \alpha \right)^2 \tag{3}$$

where $u_i = x_i - \bar{x}$, $v_i = y_i - \bar{y}$ (for all $i = 1, 2, \ldots, N$), $\alpha = r^2$, and $(u_c, v_c)$ is the center of the circumference after the change of variable.

For simplicity, we will define as $f_{u_c, v_c, \alpha}(u_i, v_i) := (u_i - u_c)^2 + (v_i - v_c)^2 - \alpha$ the distance function, and by $F(u_c, v_c, \alpha) := \sum_{i=1}^{N} f_{u_c, v_c, \alpha}(u_i, v_i)^2$ the objective function of our minimization problem.

**Note 1** Note that, with the new convention of notation, the problem (**Equation (3)**) is equivalent to:

$$\min_{u_c, v_c, \alpha} F(u_c, v_c, \alpha) \tag{4}$$

A simple alternative to resolve **Equation (4)** is to consider the extreme values of $F(u_c, v_c, \alpha)$, this is:

$$\nabla F(u_c, v_c, \alpha) = \left( \frac{\partial F}{\partial u_c}, \frac{\partial F}{\partial v_c}, \frac{\partial F}{\partial \alpha} \right) = (0, 0, 0). \tag{5}$$

The partial derivates of the function $F(u_c, v_c, \alpha)$ are:

1. For $\alpha$:

$$\frac{\partial F}{\partial \alpha} = -2 \sum_{i=1}^{N} f_{u_c, v_c, \alpha}(u_i, v_i),$$

Taking into account the condition given in **Equation (5)** for $\frac{\partial F}{\partial \alpha}$, we obtain that:

$$\sum_{i=1}^{N} f_{u_c, v_c, \alpha}(u_i, v_i) = 0. \tag{6}$$

2. For $u_c$:

$$\frac{\partial F}{\partial u_c} = -4 \sum_{i=1}^{N} \left[ f_{u_c, v_c, \alpha}(u_i, v_i)(u_i - u_c) \right]$$

$$= -4 \left[ \sum_{i=1}^{N} u_i f_{u_c, v_c, \alpha}(u_i, v_i) - u_c \sum_{i=1}^{N} f_{u_c, v_c, \alpha}(u_i, v_i) \right]$$

$$= -4 \sum_{i=1}^{N} u_i f_{u_c, v_c, \alpha}(u_i, v_i)$$

3. For $v_c$: It is the same process than $u_c$, the final result is,

$$\frac{\partial F}{\partial v_c} = -4 \sum_{i=1}^{N} v_i f_{u_c, v_c, \alpha}(u_i, v_i). \tag{8}$$

Expanding the **Equation (7)** we obtain:

$$\frac{\partial F}{\partial u_c} = -4 \sum_{i=1}^{N} u_i f_{u_c, v_c, \alpha}(u_i, v_i) = -4 \sum_{i=1}^{N} u_i \left[ (u_i - u_c)^2 + (v_i - v_c)^2 - \alpha \right]$$

$$= -4 \sum_{i=1}^{N} u_i \left[ (u_i^2 + u_c^2 - 2u_i u_c) + (v_i^2 + v_c^2 - 2v_i v_c) - \alpha \right]$$

$$= -4 \left[ \sum_{i=1}^{N} u_i^3 + u_c^2 \sum_{i=1}^{N} u_i - 2u_c \sum_{i=1}^{N} u_i^2 + \sum_{i=1}^{N} u_i v_i^2 + v_c^2 \sum_{i=1}^{N} u_i - 2v_c \sum_{i=1}^{N} u_i v_i - \alpha \sum_{i=1}^{N} u_i \right],$$

and taking into account that $u_i = x_i - \bar{x} \Rightarrow \sum_{i=1}^{N} u_i = \sum_{i=1}^{N} x_i - N\bar{x} = \sum_{i=1}^{N} x_i - N \frac{\sum_{i=1}^{N} x_i}{N} = 0$, we obtain that:

$$\frac{\partial F}{\partial u_c} = -4 \left[ \sum_{i=1}^{N} u_i^3 - 2u_c \sum_{i=1}^{N} u_i^2 + \sum_{i=1}^{N} u_i v_i^2 - 2v_c \sum_{i=1}^{N} u_i v_i \right]. \tag{9}$$

Let $C_{(u^k, v^t)} := \sum_{i=1}^{N} u_i^k v_i^t$, then the equation **Equation (9)** can be written as:

$$\frac{\partial F}{\partial u_c} = -4\left[C_{(u^3,\, v^0)} - 2u_c C_{(u^2,\, v^0)} + C_{(u^1,\, v^2)} - 2v_c C_{(u^1,\, v^1)}\right],$$

and considering the extreme condition $\frac{\partial F}{\partial u_c} = 0$ (see **Equation (5)**) we obtain:

$$u_c C_{(u^2,v^0)} + v_c C_{(u^1,v^1)} = \frac{1}{2}\left(C_{(u^3,v^0)} + C_{(u^1,v^2)}\right). \tag{10}$$

Following the same process, but now considering the **Equation (8)**, we obtain:

$$u_c C_{(u^1,v^1)} + v_c C_{(u^0,v^2)} = \frac{1}{2}\left(C_{(u^0,v^3)} + C_{(u^2,v^1)}\right). \tag{11}$$

The extreme values $\widehat{u}_c$ and $\widehat{v}_c$ of the function $F(u_c, v_c, \alpha)$ are obtained as solution of the linear system formed by **Equation (10)** and **Equation (11)**.

**Note 2** The coordinate system translation to the mass center of the original data, allow the simplification of the equation (**Equation (9)**).

The radius of the circumference is obtained developing the **Equation (6)**:

$$\begin{aligned}
0 &= \sum_{i=1}^{N}\left[u_i^2 + u_c^2 - 2u_i u_c + v_i^2 + v_c^2 - 2v_i v_c - \alpha\right] \\
&= \sum_{i=1}^{N} u_i^2 + N u_c^2 + \sum_{i=1}^{N} v_i^2 + N v_c^2 - N\alpha,
\end{aligned}$$

and writing the equation in function of $\alpha$, we obtain that:

$$\widehat{\alpha} = u_c^2 + v_c^2 + \frac{C_{(u^2,v^0)} + C_{(u^0,v^2)}}{N}. \tag{12}$$

The point $(\widehat{u}_c,\ \widehat{v}_c,\ \widehat{\alpha})$ is a critical point of the function $F(u_c, v_c, \alpha)$. Finally, we need to compute the Hessian matrix of $F(u_c, v_c, \alpha)$ to test if $(\widehat{u}_c,\ \widehat{v}_c,\ \widehat{\alpha})$ is a maximum, a minimum, or an inflection point:

$$H = \begin{pmatrix}
\frac{\partial^2 F}{\partial \alpha^2} & \frac{\partial^2 F}{\partial \alpha \partial u_c} & \frac{\partial^2 F}{\partial \alpha \partial v_c} \\
\frac{\partial^2 F}{\partial u_c \partial \alpha} & \frac{\partial^2 F}{\partial u_c^2} & \frac{\partial^2 F}{\partial u_c \partial v_c} \\
\frac{\partial^2 F}{\partial v_c \partial \alpha} & \frac{\partial^2 F}{\partial v_c \partial u_c} & \frac{\partial^2 F}{\partial v_c^2}
\end{pmatrix} = \begin{pmatrix}
2N & -4Nu_c & -4Nv_c \\
0 & 8\sum_{i=1}^{N} u_i^2 & 8\sum_{i=1}^{N} u_i v_i \\
0 & 8\sum_{i=1}^{N} u_i v_i & 8\sum_{i=1}^{N} v_i^2
\end{pmatrix} \tag{13}$$

The principal minors of $H$ are:

$$H_1 = 2N > 0,$$

$$H_2 = \begin{vmatrix} 2N & -4Nu_c \\ 0 & 8\sum_{i=1}^{N} u_i^2 \end{vmatrix} = 16N \sum_{i=1}^{N} u_i^2 > 0,$$

$$H_3 = \begin{vmatrix} 2N & -4Nu_c & -4Nv_c \\ 0 & 8\sum_{i=1}^{N} u_i^2 & 8\sum_{i=1}^{N} u_i v_i \\ 0 & 8\sum_{i=1}^{N} u_i v_i & 8\sum_{i=1}^{N} v_i^2 \end{vmatrix} = 128N \underbrace{\left[\sum_{i=1}^{N} u_i^2 \sum_{i=1}^{N} v_i^2 - \left(\sum_{i=1}^{N} u_i v_i\right)^2\right]}_{\text{Cauchy-Schwarz inequality}} > 0.$$

**Note 3** The Cauchy-Schwarz inequality is equal to zero if and only if exist a real value $r$ such that $u_i r + v_i = 0$, for all $i = 1, 2, \ldots, N$, that is, all the points are in a straight line. In our problem, the proteins do not follow a straight line distribution in the space, then, we can guarantee the strict inequality. Note that $H_2 = 0$ if and only if $u_i = 0$, for all $i = 1, 2, \ldots, N$, which is a particular case of a straight line.

The principal minors of the matrix $H$ are greater than zero, then, the point $(\widehat{u}_c,\ \widehat{v}_c,\ \widehat{\alpha})$ is a local minimum, but this point is the only critical point of the function $F(u_c, v_c, \alpha)$, and as consequence it is a global minimum.

The center of the fitted circumference in the original coordinate system is $(x_c, y_c) = (\widehat{u_c}, \widehat{v_c}) + (\overline{x}, \overline{y})$, and the radius is $R = \sqrt{\widehat{\alpha}}$.

**Note 4** This method is based on a simple and very popular mathematical techniques (specifically Differential Calculus). Even when we do not found a published paper with this approach, we want to take cares and not adjudicate this technique to our work.

**Note 5** Note that, by definition, the distance function $f_{u_c, v_c, \alpha}(u_i, v_i)$ is not a geometric distance to the circle, which is probably more intuitive to use in this kind of problems. Let consider for a second the geometric distance of a set of points $P$ to a circle, which is given by:

$$G(x_c, y_c, r) := \sum_{i=1}^{N} \left( r - \sqrt{(x_i - x_c)^2 + (y_i - y_c)^2} \right)^2,$$

where $(x_c, y_c)$ is the center of the circumference and $r$ is the radius. Considering the extreme condition for the function $G(x_c, y_c, r)$, it is obtained that:

$$\frac{\partial G}{\partial r} = -2 \sum_{i=1}^{N} \sqrt{(x_i - x_c)^2 + (y_i - y_c)^2} + 2Nr$$

$$\frac{\partial G}{\partial x_c} = 2r \sum_{i=1}^{N} \frac{x_i - x_c}{\sqrt{(x_i - x_c)^2 + (y_i - y_c)^2}} - 2N\overline{x} + 2Nx_c$$

$$\frac{\partial G}{\partial y_c} = 2r \sum_{i=1}^{N} \frac{y_i - y_c}{\sqrt{(x_i - x_c)^2 + (y_i - y_c)^2}} - 2N\overline{y} + 2Ny_c,$$

where $\overline{x}$ and $\overline{y}$ are the mean values of the variables $x$ and $y$ respectively. Simultaneously equating these partials to zero does not produce closed form solutions for $x_c$, $y_c$, and $r$. With numerical methods it is possible to carry out the optimization of these parameters, but these alternatives are iterative, no-deterministic process, which also are highly sensitives to the initial values, while our proposal not have these kind of unwished characteristics. In any case, in Appendix 1 Section –Algorithm validation– we compared our approach with an algorithm based on geometric distance and explained in details the benefits of each one.

The final segmentation algorithm is composed by three steps (**Appendix 1—figure 1**). The first is a manual pre-segmentation that guarantees the existence of one and only one viroplasm per image (**Appendix 1—figure 1A**). In the second step, we adjust a circumference to the reference protein (remember that these are paired experiments, where NSP2 is present in all the combinations as reference protein) through the algorithm DLSFC (**Appendix 1—figure 1B**), and finally the radius of the accompanying protein is adjusted using the **Equation (12)**. Note that, the center of the accompanying protein is the same that the center of the reference protein (concentric model). Out of the manual pre-segmentation step, this algorithm is automatic, deterministic, not iterative and with a linear computational complexity.

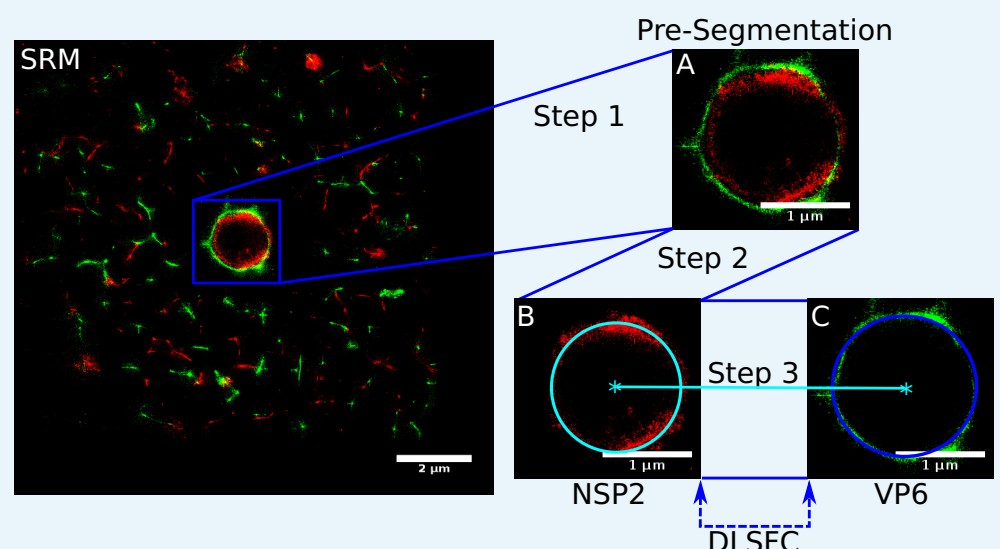

**Appendix 1—figure 1.** Scheme of the 'Viroplasm Direct Least Square Fitting Circumference' algorithm (VPs-DLSFC). SRM) Complete SRM image; **A**) Manual pre-segmentation step, an expert selects and isolates each viroplasm as a single image; **B**) Fit a circumference to the reference protein through the algorithm DLSFC; **C**) The center of the reference protein is taken as the center of the accompanying protein, and then the radius of the adjust circumference for this second protein is computed.
DOI: https://doi.org/10.7554/eLife.42906.009

**Note 6** The hypothesis that the viral proteins in the viroplasm are distributed like as concentric circumferences was proved in Appendix 1 Section – Model Considerations– ('Model Considerations').

## Algorithm validation

The algorithm DLSFC was compared with other two approaches proposed by *Gander et al. (1994)*. For a more comprehensive and clear comparison we named these others two methods as 'Algebraic Least Square Fitting Circle' (ALSFC) and 'Geometric Least Square Fitting Circle' (GLSFC).

The algorithm ALSFC considers the algebraic representation of a circle in the plane:

$$F(p) = Ap^t p + B^t p + C; \tag{14}$$

where $p = (x, \ y)$ is a point in $\mathbb{R}^2$, $B = (B_1, \ B_2) \in \mathbb{R}^2$, $A \neq 0$, and $C$ is the independent term. Then, let $p_i = (x_i, \ y_i) \in \mathbb{R}^2$, $i = 1, 2, \ldots, N$ the set of points that we want to adjust. The objective function for this minimization problem is $\sum_{i=1}^{n} F(p_i)$, which can be represented as matrix form:

$$\Psi u = \underbrace{\begin{pmatrix} x_1^2 + y_1^2 & x_1 & y_1 & 1 \\ x_2^2 + y_2^2 & x_2 & y_2 & 1 \\ \vdots & \vdots & \vdots & \vdots \\ x_n^2 + y_n^2 & x_n & x_n & 1 \end{pmatrix}}_{\psi} \underbrace{\begin{pmatrix} A \\ B_1 \\ B_2 \\ C \end{pmatrix}}_{u} \tag{15}$$

The final optimization problem is given by:

$$\begin{cases} \min_{u} \|\Psi u\| \\ \text{subjet to: } \|u\| = 1 \end{cases},\tag{16}$$

where $\|u\| = 1$ is a contrain to avoid the trivial solution $u = (0, 0, 0, 0)$. The notation $\|v\| = \sqrt{v_1^2 + v_2^2 + \ldots + v_m^2}$ represent the Euclidean norm. Finally, the problem (*Equation (16)*) is solved considering the right singular eigenvector associated with the smallest singular eigenvalue of $\Psi$ (*Gander et al., 1994*).

On the other hand, the algorithm GLSFC deals with the minimization of the geometric distance:

$$\min_{x_c, y_c, r} \sum_{i=1}^{n} (\|(x_c, \ y_c) - p_i\| - r)^2.\tag{17}$$

Note that $(\|(x_c, \ y_c) - p_i\| - r)^2$ is the geometric distance of the point $p_i$ to the circle $(x - x_c)^2 + (y - y_c)^2 = r^2$. This problem is nonlinear, and in *Gander et al. (1994)*, the authors consider to resolve it trought a Gauss-Newton algorithm. The Gauss-Newton method is an iterative algorithm that depends of an initial approximation, and in this case, Gander and collaborators consider the solution of the problem (*Equation (16)*) as the initial value for the iterative process. More details about the algorithms ALSFC and GLSFC can be consulted in *Gander et al. (1994)*. In summary, the method ALSFC is very simple and computational efficient, but the algebraic distance might not reflect with accuracy the real distance between the points and the circumference and, as consequence, the parameters of the model might be biased. On the contrary, the algorithm GLSFC is highly precise, but it is an iterative process (demand more computation time), and depends of good initial values to achieve the convergence. The Matlab codes for the algorithms GLSFC and ALSFC were obtained from *Brown (2007)*.

For the validation, we simulated the spatial distribution of viral elements as circumferences, for which we know their parametric form ('ground truth' dataset) (*Appendix 1—figure 2A*). The use of 'ground truth' allows us to quantify the error in the adjustment of the algorithms DLSFC, ALSFC and GLSFC at different noise levels (*Appendix 1—figure 2B*) and partial occlusion conditions (*Appendix 1—figure 2C*). We generated over 40 000 images (size $512 \times 512$ pixels) taking into account different levels of additive white Gaussian noise (AWGN) in order to consider auto-fluorescence and the error in the localization of the fluorophores by the algorithm 3B-ODE (see Appendix 1 Section –3B Algorithm–), and partial occlusion angles. Several radii and position of the synthetic viroplasms were generated randomly through a uniform distribution function. The codes for this simulation are availables at https://github.com/Yasel88/Nanoscale_organization_of_rotavirus_replication_machineries (*Garcés Suárez, 2019*; copy archived at https://github.com/elifesciences-publications/Nanoscale_organization_of_rotavirus_replication_machineries).

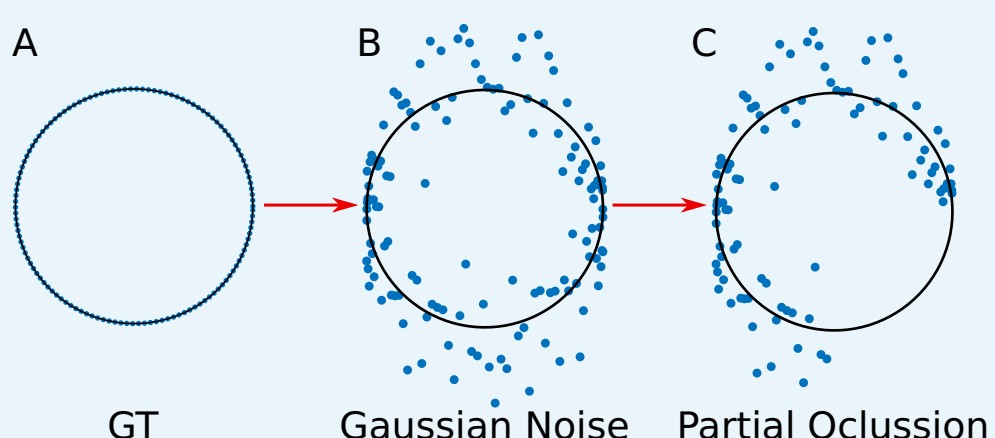

**Appendix 1—figure 2.** Simulation of the viral proteins. (**A**) 'Ground truth' (GT) circumference; (**B**) Addition of gaussian noise to the GT circumference (see Appendix 1 subsection –noise generation–); (**C**) Generation of partial occlusion (see Appendix 1 subsection –Partial Occlusion Generation).

DOI: https://doi.org/10.7554/eLife.42906.010

## Noise generation

Let $\mathcal{I}$ a synthetic image and $p_i \in \mathbb{R}^2$, $i = 1, 2, \ldots, N$ points of the 'ground truth' circumference $\mathcal{C} \in \mathcal{I}$, the AWGN over $p_i$, $\forall i = \overline{1, N}$ was generated as:

$$p_i^c = p_i + \sigma X_i, \tag{18}$$

where $p_i^c$ is the corrupted point after the contamination with AWGN, $\sigma$ is a scalar that represents the level of noise, and $X_i$ is a trial of a random variable $X \sim N(0, 1)$ (standard gaussian distribution). In our experimental design we included 20 different levels of noise $\sigma = 0 : 0.5 : 10$ (increase by 0.5 from 0 to 10). Note that the noise was added to the points of the circumferences and not to the images. This is because we are working with SRM images, and as consequence does not exist background noise.

## Partial occlusion generation

The partial occlusion can be seen as the existence of incomplete information of the objects of interest, for example, an image where we have only information about the half of the VPs. In our perspective, it is important that a segmentation algorithm have a good performance even in these situations.

**Definition 1** We called an angle $\theta$ as a partial oclussion angle for a set of points $p_i = (x_i, \ y_i) \in \mathbb{R}^2, i = 1, 2, \ldots, N$, if not exist $j \in \{1, 2, \ldots, N\}$, $r \in \mathbb{R}$ and $\beta \in [\alpha, \alpha + \theta]$ such as:

$$p_j = (\bar{x} + r \cos\beta, \ \bar{y} + r \sin\beta),$$

where $\alpha$ is an angle in $[0, 2\pi]$, and $(\bar{x}, \ \bar{y}) = \left( \frac{1}{N} \sum_{i=1}^{N} x_i, \ \frac{1}{N} \sum_{i=1}^{N} y_i \right)$ is the center of mass of the points $p_i$.

**Note 7** Note that, in the Definition 1 we considered the interval $[\alpha, \alpha + \theta]$ as a partial oclussion angle $\theta$, but it is possible to obtain the same oclussion angle with $[\alpha, \alpha - \theta]$. For our validation we took $\alpha = 0$ for simplicity, but this has not consequences in our experiments, because we generate the points as a circumference.

The relationship between cartesian and polar coordinates is given by:

$$\begin{cases} x &= a + r \cos\beta \\ y &= b + r \sin\beta \end{cases}, \tag{19}$$

where, (a, b) is the circumference center, $r$ is the radius and $\beta$ is the angular coordinate. Then, for example, to provoke a partial occlusion of $\theta$, it is just needed to avoid the generation of points in the angle $[0, \theta]$ or $[2\pi, 2\pi - \theta]$ through the *Equation (19)*. *Appendix 1—figure 3* shows this strategy, in this example, a half of the information is removed from the 'ground truth' circumference.

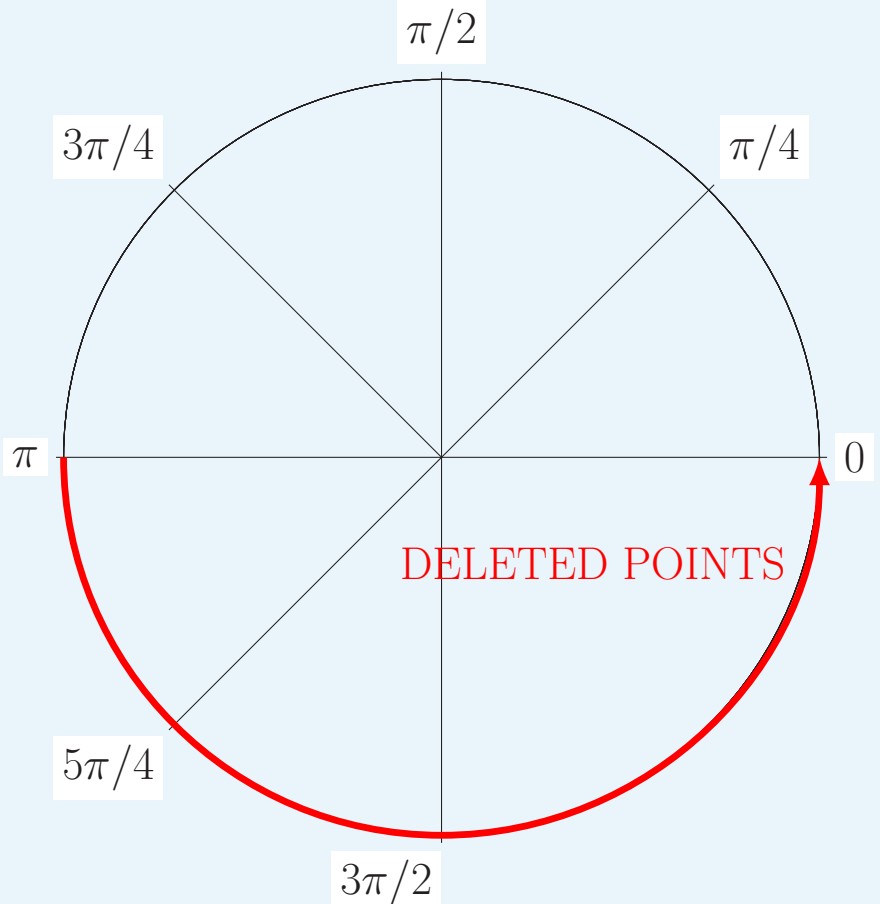

**Appendix 1—figure 3.** Generation of partial occlusion in the angle $\theta = \pi$ (red line). The circle just conserves the information relative to the angle $[0, \pi]$.

DOI: https://doi.org/10.7554/eLife.42906.011

In general, the average of the points that will be conserved after the generation of partial occlusion is given by:

$$P_\% = 100(1 - \theta 2\pi).$$

In the validation of the algorithm DLSFC, we considered four different partial occlusion conditions: 1- Not partial occlusion: Conserves all the points of the 'ground truth' circumference; 2– 25% of partial occlusion: Partial occlusion angle $\theta = \pi/2$; 3– 50% of partial occlusion: Partial occlusion angle $\theta = \pi$; 4– 75% of partial occlusion: Partial occlusion angle $\theta = 3\pi/2$.

In the experiments, we represent the noise through the mean and the standard deviation of the distance of the points $p_i^c$, $i = 1, 2 \dots, n$ to the 'ground truth' circumference $\mathcal{C}$, this is:

$$\underbrace{\overline{d_{\mathcal{C}}} = \frac{\sum_{i=1}^{n} d_{i\mathcal{C}}}{n}}_{\text{Mean Value}}, \quad \underbrace{\sigma_{d_{\mathcal{C}}} = \sqrt{\frac{\sum_{i=1}^{n} \left(d_{i\mathcal{C}} - \overline{d_{\mathcal{C}}}\right)^2}{N-1}}}_{\text{Standard Deviation}} \tag{20}$$

where $d_{i\mathcal{C}} = \left|\|\mu - p_i^c\| - r\right|$ is the geometric distance between the point $p_i^c$ and the circumference $\mathcal{C}$, $|\cdot|$ represent the absolute value, $\mu = (x_c, y_c)$ is the center of the circumference and $r$ is the radius. The mean and standard deviation are an useful alternative to understand the amount of dispersion generated by a noise $\sigma$ in a circumference $\mathcal{C}$.

The **Appendix 1—figure 4** shows the adjustment through the algorithms DLSFC, GLSFC and ALSFC of one 'ground truth' circumference (solid black line) that was corrupted by different levels of AWGN and partial occlusion angles. As was expected, the performance of the algorithms seems to get worse when increase the level of partial occlusion, but the three algorithms shows good results for high AWGN levels. This example suggests that the algorithm GLSFC (solid red line) has a better performance than DLSFC (solid blue line) and ALSFC (solid green line) when the partial occlusion is less than $\pi$ (columns A-C), but when $\theta = 3\pi/2$, this method breaks down, and this observation is more relevant when increase the level of noise. The algorithm ALSFC shows relative good results for all the experimental conditions, but it is interesting to note that, in most cases, the adjustment is worse than in the DLSFC and GLSFC alternatives, probably as consequence of the algebraic distance (see Appendix 1 Section –Algorithm validation–). This example is just for a first visual analysis, but a complete study of the results obtained by the three algorithms over the 40 000 synthetic images will be presented below.

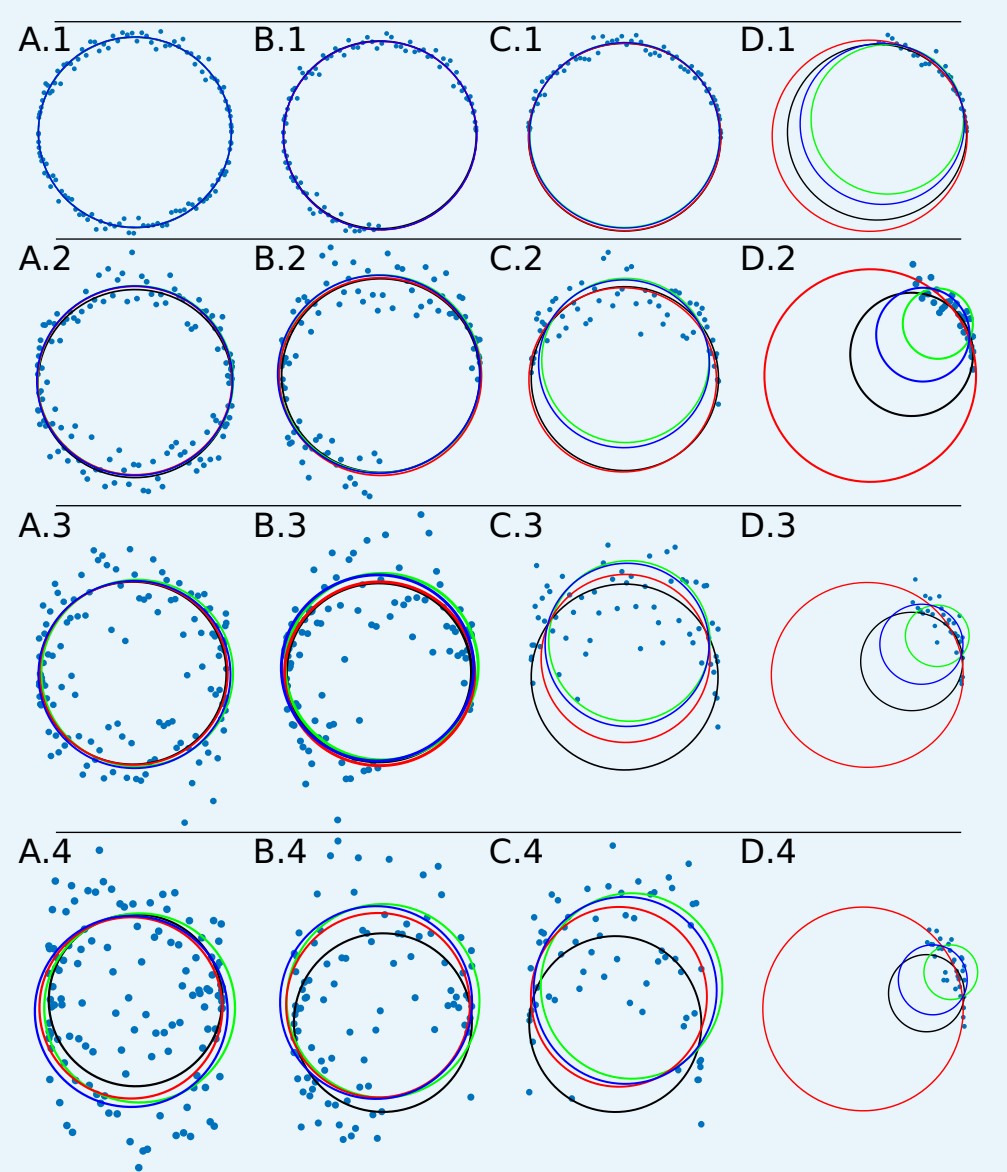

**Appendix 1—figure 4.** Data adjustment through the algorithms DLSFC (solid blue line), ALSFC (solid green line) and GLSFC (solid red line). The data was generated corrupting the points of a 'ground truth' circumference (solid black line) with different noise levels (rows) and four differents occlusion angles conditions (columns). Column 1 (A.1-A.4) Not Occlusion angle, Column 2 (B.1-B.4) $\theta = \pi/2$, Column 3 (C.1-C.4) $\theta = \pi$, Column 4 (D.1-D.4) $\theta = 3\pi/2$. The noise increase by rows: Row 1: $\overline{d_\mathcal{C}} \in [1.5\mu m, \ 1.54\mu m]$, $\sigma(d_\mathcal{C}) \in [1.29, \ 1.74]$, $\sigma_{\mathrm{AWGN}} = 2.1053$; Row 2: $\overline{d_\mathcal{C}} \in [4.19\mu m, \ 5.28\mu m]$, $\sigma(d_\mathcal{C}) \in [3.64, \ 5.71]$, $\sigma_{\mathrm{AWGN}} = 4.7368$; Row 3: $\overline{d_\mathcal{C}} \in [7.5\mu m, \ 9.23\mu m]$, $\sigma(d_\mathcal{C}) \in [7.49, \ 10.38]$, $\sigma_{\mathrm{AWGN}} = 7.3684$; Row 4: $\overline{d_\mathcal{C}} \in [12.58\mu m, \ 17.78\mu m]$, $\sigma(d_\mathcal{C}) \in [11.12, \ 17.96]$, $\sigma_{\mathrm{AWGN}} = 10$.

DOI: https://doi.org/10.7554/eLife.42906.012

A circumference can be represented as a vector $\mathcal{C} = (r, x_c, y_c) \in \mathbb{R}^3$, where $r$ is the radius, and $(x_c, \ y_c)$ the center. Let $\hat{\mathcal{C}} = (\hat{r}, \hat{x}_c, \hat{y}_c)$ and $\mathcal{C} = (r, x_c, y_c)$ the 'ground truth' and the adjusted circumference, respectively. The fit error committed by the algorithms can be quantified as:

$$\varepsilon = \left\| \hat{\mathcal{C}} - \mathcal{C} \right\| \tag{21}$$

Now, thanks to the use of 'ground truth' circumferences, it is possible to compare the obtained outcome with the ideal result. We consider that an adjustment is good enough if the estimation error of each circumference's component $\mathcal{C}$ is less than 0.1 $\mu$m. This value is approximately two-fold smaller than the theoretical Abbe's limit ($\approx 250nm$), and taking into account that the synthetic images resolution vary between 4.6 and 25.8 $\mu$m in mean (see *Appendix 1—figure 5*), we are demanding that the algorithm has a nanoscopic precision (less than 100 nm) even when the images have a very bad resolution. Combining the threshold explained above with *Equation (21)*, we obtain the final condition based on the euclidean norm between these two vectors:

$$\varepsilon = \left\| \hat{\mathcal{C}} - \mathcal{C} \right\| \leq \|(0.1, \, 0.1, \, 0.1)\| = 0.1732 \mu m. \tag{22}$$

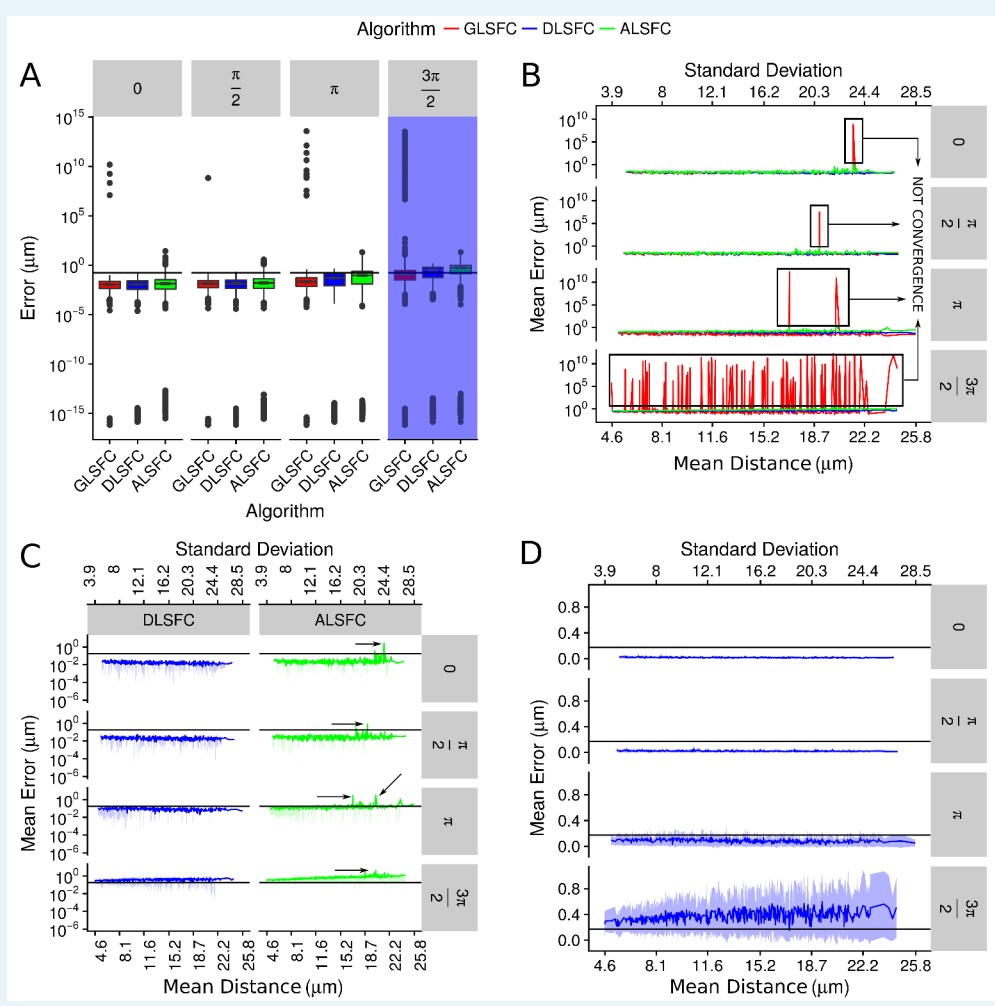

**Appendix 1—figure 5.** Error in the adjustment of the algorithms DLSFC, GLSFC and ALSFC. The error was quantified through *Equation (21)*. In all panels, the horizontal black line represents the value $y = 0.1732 \mu m$ (see *Equation (21)*). (**A**) Boxplot of the error distribution for each algorithm taking into account the partial occlusion angles (four sub-panels). The x-axis specifies the name of the algorithm and the y-axis the error in microns. The blue shadow in the sub-panel $3\pi2$ represents the occlusion angle in which the mean value of the errors are greater than $0.1732 \mu m$. (**B**) Mean error of the adjustment by the algorithms DLSFC, GLSFC and ALSFC.

The bottom x-axis is the Mean distance of the corrupted points to the 'ground truth' circumference (see *Equation (20)*), and the up x-axis is the Standard Deviation. The figure is split out in four sub-panels in accordance with the occlusion angle. The black boxes show examples in which the algorithm GLSFC does not reach the convergence (extremely high error). The arrows mark out some examples where the algorithm ALSFC does not have have a good adjustment. (**C**) Zoom of the performance of the algorithms DLSFC and ALSFC. (**D**) Results of the algorithm DLSFC. The graphics in the panel (**C**)) and (**D**)) also shows the confidence interval around the mean (see green and blue shadows), it was computed as $\overline{d_{\mathcal{C}}} \pm \sigma(d_{\mathcal{C}})$.

DOI: https://doi.org/10.7554/eLife.42906.013

The *Appendix 1—figure 5(a)* shows the descriptive analysis (through boxplots) of the errors generated by the algorithms DLSFC, GLSFC and ALSFC over the 40000 synthetic images. In some experiments we obtained extremely good adjusts ($\varepsilon \approx 10^{-15}\mu m$), but in general the 75% of the errors are in the interval $[10^{-3}, 1]\mu m$. For all occlusion angles, the algorithm GLSFC had a bad behaviour in a wide range of experiments with an error of $\approx 10^{10}\mu m$. This kind of situation was expected for big partial occlusion angles (for example $\theta = 3\pi/2$), but in conditions where does not exist partial occlusion ($\theta = 0$), the bad adjustment of GSFC is evidence that this algorithm is not a good alternative in this kind of problems. Note that a similar situation happens with the algorithm ALSFC, but in this case the maximum error is $\approx 10^3 \mu m$, which is high too, but significantly less than that obtained by GLSFC. The algorithm DLSFC does not show the previous problems; observe that the maximun error is under $1\mu m$ for all experimental conditions, even when $\theta = 3\pi/2$. Now, if we forget the outliers and take into account only the 75% of the experiments (boxes), a direct relationship between the error of each algorithm and the partial occlusion angles (increase occlusion angle imply a major error) is observed. However, the raise of the mean error is small in relationship with the occlusion angle, which suggests that these three methods are robust to partial occlusion. The blue shadow band highlights the occlusion angle ($\theta = 3\pi/2$) for which the algorithms have a median error greater than $0.1732\mu m$. Therefore, it is necessary at least the 50% of the data to obtain a good adjustment (with an error under $0.1732\mu m$) in the 75% of the experiments.

The *Appendix 1—figure 5(b)* displays the mean error obtained by each algorithm in relationship with the resolution of the images (mean and the standard deviation of the points distance to the 'ground truth' circumference, *Equation (20)*). This graph shows again that the method GLSFC did not reach the convergence in some experiments (remember that GLSFC is based on an iterative optimization algorithm), even for small partial occlusion angles, but it is interesting to point out that in all these cases (at least for the occlusion angles $\theta = \left\{0, \frac{\pi}{2}, \pi\right\}$), these high errors ($\approx 10^{10}\mu m$) seem to be related with bad initial values. Note that in these experiments the algorithm ALSFC do not has a good performance (remember that the result of ALSFC is the initial value for GLSFC), which is an evidence of the sensibility of the GLSFC to the initial values. For the occlusion angle $\theta = \frac{3\pi}{2}$, a lot of bad fits were obtained through the algorithm GLSFC. In our opinion, even when the algorithm GLSFC can reach an extremely good fit, its sensitivity to initial values, the computational cost (iterative process), and the possibility of not convergence, are strongs reasons to consider that this alternative is not viable to use in this kind of problems.

In the *Appendix 1—figure 5(c)*, the mean error of the algorithms DLSFC and ALSFC was splitted to see more clearly the performance of each one. For small partial occlusion angles ($\theta = \left\{0, \frac{\pi}{2}\right\}$) the algorithm DLSFC has errors $\leq 0.1732\mu m$, while ALSFC, in high noise scenes, get a poor adjustment (black arrows). When $\theta = \pi$, the method ALSFC has errors $\geq 0.1732\mu m$ in many cases, that increase when the level of noise grows. Both algorithms generate an error $\geq 0.1732\mu m$ when $\theta = \frac{3\pi}{2}$, but in the case of DLSFC, in some experiments had errors $<0.1732\mu m$, even for high noise levels. The *Appendix 1—figure 5(d)* show only the performance of the algorithm DLSFC. As noted, the method DLSFC shows a behaviour extremely robust and stable to noise and partial acclusion; note that for $\theta = \left\{0, \frac{\pi}{2}, \pi\right\}$ does not exist an increment in the mean error when the noise level is incremented. In extreme experimental conditions,

when only the 25% of the data remain in the image $(\theta = \frac{3\pi}{2})$, DLSFC has a mean error between $[0.3, 0.5]\mu m$ with a 95% confidence interval equal to $[0.1, 0.9]\mu m$.

So far we have shown the behaviour of the three algorithms considering several noise levels and partial occlusion angles conditions. However, it is interesting to know through a hypothesis test (inferential statistics) the performance of DLSFC and ALSFC taking into account the experiments we developed (samples). Note that we are not going to study the algorithm GLSFC, because, as we mentioned above, its sensitivity to initial values and its computational cost (iterative process) are strong reasons to consider that this algorithm is not viable to use in this kind of problems. Taking into account the threshold in *Equation (22)*, for each experiment we have two possible outcomes:

$$\begin{cases} \text{success,} & \text{if } \varepsilon \leq 0.1732\mu m \\ \text{failure,} & \text{if } \varepsilon > 0.1732\mu m \end{cases}. \tag{23}$$

From this point of view, each experiment follows a Bernoulli's distribution, and then, all the experiments for the same experimental condition (partial occlusion angle) follows a Binomial distribution. The generated samples are independent (random generation), the decision rule (*Equation (23)*) is dichotomous, and our experimental design is a fair representation of the population (we considered many experimental conditions in more than 40000 images); then, it is possible to develop a 'Binomial Test' (*Prybutok, 1989*; *Howell, 1982*) to evaluate the population proportion of success, which we will denote by $p$. The null (H0) and alternative (H1) hypothesis for this test are:

$$\begin{cases} H0: & p = 0.7 \\ H1: & p > 0.7 \end{cases}. \tag{24}$$

The null hypothesis is that the population proportion with an error less that 0.1732 is 0.7, and the alternative hypothesis is that the proportion is more that 0.7. The significance level for these tests is fixed at $\alpha = 0.05$. For more information about this test and its implementation review (*R Development Core Team, 2017*; *Prybutok, 1989*; *Howell, 1982*) Online documentation is available in url https://stat.ethz.ch/R-manual/R-devel/library/stats/html/00Index.html.

The *Appendix 1—table 1* shows the results of the 'Binomial Test' for the algorithms DLSFC and ALSFC. For small occlusion angles $\theta = \{0, \frac{\pi}{2}\}$, the results of both methods are very similar and indicate a high performance. Note that in these cases the null hypothesis was rejected with a probability error of type I of $\approx 10^{-16} \leq \alpha$, and the 95% confidence intervals are equal to $[0.99, 1]$, which means that, at a population level, we hope that at least the 99% of the adjustments had an error $\leq 0.1732\mu m$. For $\theta = \pi$, the algorithm DLSFC has a probability of success greater than 0.7 with a p-value of $\approx 10^{-16} \leq \alpha$. On the contrary, for the same oclussion angle, the test revealed that the null hypothesis can't be rejected in the case of the algorithm ALSFC. Finally, as was analyzed above, the performance of the algorithms decrease significaly for $\theta = \frac{3\pi}{2}$, and we can't reject the null hypothesis, in others words, we don't have enough evidence to assume that in this case the probability of success is greater than 0.7.

**Appendix 1—table 1.** Results of the Binomial Test. Column 1: Names of the algorithms. Column 2: 'P. Success' is the Probability of Success; 'P. Value' denote the probability of the error type 1, that is, reject the null hypothesis when it is true; 'C. Interval' is the 95% confidence interval. Column 3–6: Partial Occlusion Angles. The blue shadowed area in the table cell indicates the conditions in which it is not possible to reject the null hypothesis.

| Algorithm | Statistics | Partial occlusion angles | | | |
|---|---|---|---|---|---|
| | | **0** | $\pi/2$ | $\pi$ | $3\pi/2$ |
| | P. Success | 1 | 1 | 0.804 | 0.384 |
| DLSFC | P. Value | $2.2 \times 10^{-16}$ | $2.2 \times 10^{-16}$ | $2.2 \times 10^{-16}$ | 1 |

*Appendix 1—table 1 continued on next page*

**Appendix 1—table 2.** Two-sample Mann-Whitney hypothesis test between $a$ (major semi-axis of the ellipse ) and $r$ (circumference radius). H0(H1): The radii of the circumferences and the values of the major semi-axis of the ellipses came from the same distribution function (different). Column 1: W represent the distribution value of the statistical test; Difference is the estimation of the location parameter (median difference between $a$ and $r$); P. Value is the p-value of the test; C. Interval is the 95% confidence interval. For each one of the nine viral elements combinations, we carry up the hyphotesis test taking into account NSP2 independently in each combination, just for the semi-major axis this table shows the results of 18 Mann-Whitney hypothesis tests. More information about this test is available in *Mann and Whitney (1947)* and *Hollander et al. (2013)*.

**Proteins Combinations**

| Statistics | NSP2/NSP4 | | NSP2/NSP5 | | NSP2/VP1 | | NSP2/VP2 | |
|---|---|---|---|---|---|---|---|---|
| | NSP2 | NSP4 | NSP2 | NSP5 | NSP2 | VP1 | NSP2 | VP2 |
| W | 900 | 902 | 1426 | 1415 | 559 | 539 | 174 | 184 |
| Difference | 0.0352 | 0.03369 | 0.0358 | 0.0357 | 0.0213 | 0.026 | 0.0529 | 0.0512 |
| P. Value | 0.1233 | 0.127 | 0.091 | 0.08025 | 0.1776 | 0.117 | 0.1137 | 0.1789 |
| 95% CI | [−0.082, 0.011] | [−0.08,0.013] | [−0.086, 0.007] | [−0.08,0.006] | [−0.054,0.012] | [−0.057, 0.006] | [−0.1168,0.0204] | [-0.15, 0.028] |

| Statistics | NSP2/VP4 | | NSP2/VP6 | | NSP2/VP7-Tri | | NSP2/VP7-Mon | |
|---|---|---|---|---|---|---|---|---|
| | NSP2 | VP4 | NSP2 | VP6 | NSP2 | VP7-Tri | NSP2 | VP7-Mon |
| W | 771 | 760 | 119 | 134 | 1425 | 1421 | 551 | 528 |
| Difference | 0.0264 | 0.0221 | 0.0393 | 0.031 | 0.0301 | 0.032 | 0.0487 | 0.055 |
| P. Value | 0.1012 | 0.0833 | 0.0747 | 0.1814 | 0.1567 | 0.1503 | 0.1513 | 0.092 |
| 95% CI | [−0.0549, 0.0044] | [−0.051,0.003] | [−0.0909,0.0067] | [−0.103,0.034] | [−0.0675,0.0105] | [−0.073, 0.014] | [−0.1165,0.0221] | [−0.119,0.01] |

DOI: https://doi.org/10.7554/eLife.42906.015

**Appendix 1—table 3.** Two-sample Mann-Whitney hypothesis test between $b$ (minor semi-axis of the ellipse ) and $r$ (circumference radius). H0(H1): The radii of the circumferences and the values of the minor semi-axis of the ellipses came from the same distribution function (different). This table is equivalent to *Appendix 1—table 2*. We considered to split the results for each semi-axis of the ellipse for better analysis and visualization.

**Proteins Combinations**

| Statistics | NSP2/NSP4 | | NSP2/NSP5 | | NSP2/VP1 | | NSP2/VP2 | |
|---|---|---|---|---|---|---|---|---|
| | NSP2 | NSP4 | NSP2 | NSP5 | NSP2 | VP1 | NSP2 | VP2 |
| W | 1263 | 1261 | 1966 | 1960 | 804 | 824 | 1135 | 1134 |
| Difference | 0.0325 | 0.041 | 0.0282 | 0.03 | 0.0308 | 0.0278 | 0.0185 | 0.0216 |
| P. Value | 0.1499 | 0.1936 | 0.2334 | 0.2394 | 0.2259 | 0.2385 | 0.1995 | 0.1335 |
| 95% CI | [−0.027,0.092] | [−0.026,0.108] | [−0.019,0.077] | [−0.0213,0.08] | [−0.018,0.085] | [−0.021,0.082] | [−0.015,0.05] | [−0.012,0.057] |

| Statistics | NSP2/VP4 | | NSP2/VP6 | | NSP2/VP7-Tri | | NSP2/VP7-Mon | |
|---|---|---|---|---|---|---|---|---|
| | NSP2 | VP4 | NSP2 | VP6 | NSP2 | VP7-Tri | NSP2 | VP7-Mon |
| W | 227 | 213 | 1931 | 1895 | 798 | 814 | 291 | 279 |
| Difference | 0.038 | 0.029 | 0.0259 | 0.0288 | 0.0427 | 0.046 | 0.0435 | 0.038 |
| P. Value | 0.1814 | 0.3543 | 0.17 | 0.2407 | 0.2233 | 0.1641 | 0.2578 | 0.3953 |
| 95% CI | [−0.023,0.097] | [−0.031,0.106] | [−0.013,0.066] | [−0.02, 0.074] | [−0.022,0.107] | [−0.019,0.113] | [−0.0317,0.107] | [−0.058,0.135] |

DOI: https://doi.org/10.7554/eLife.42906.016

Appendix 1—table 1 continued

| Algorithm | Statistics | Partial occlusion angles | | | |
|---|---|---|---|---|---|
| | | 0 | $\pi/2$ | $\pi$ | $3\pi/2$ |
| | C. Interval | [0.999, 1] | [0.999, 1] | [0.797, 1] | [0.376, 1] |
| | P. Success | 0.993 | 0.99 | 0.643 | 0.273 |
| ALSFC | P-value | $2.2 \times 10^{-16}$ | $2.2 \times 10^{-16}$ | 1 | 1 |
| | C. Interval | [0.99, 1] | [0.99, 1] | [0.635, 1] | [0.265, 1] |

DOI: https://doi.org/10.7554/eLife.42906.014

The study carried up in this section shows that the algorithm GLSFC has a good performance and is capable to obtain a low error in the adjustment, but this alternative is sensitive to the initial values (*Appendix 1—figure 5 (a,b)*), and as any iterative process, consumes more computational resources. The method ALSFC proved to have a good behavior in many experimental conditions, although in diverse occasions a bad fit was obtained. Also, the behaviour of this algorithm was always inferior than that of DLSFC (*Appendix 1—figure 5(C)* and *Appendix 1—table 1*). Finally, the algorithm DLSFC had a great performance and stability in all the experimental conditions, even in extreme partial occlusion angles (*Appendix 1—figure 5* and *Appendix 1—table 1*). This method proved to be robust in high noise conditions and it is a deterministic and not an iterative algorithm. For all these reasons we consider that the algorithm DLSFC is the best choice in comparison with GLSFC and ALSFC, to fitting the spatial distribution of the viral elements.

## Model considerations

To proof the hypothesis that the viral elements of the VPs can be approximated through circumferences, we carried up a series of experiments based on the comparison between the circumference obtained by the algorithm DLSFC, and a least squared ellipse resulting of the 'Direct Least Square Fitting Ellipse' (DLSFE) algorithm (*Fitzgibbon et al., 1999*). The algorithm DLSFE has been used previously to adjust the viral replication centers of adenoviruses in fluorescence images (limited by diffraction) (*Garcés et al., 2016*), and has also been shown to be robust to noise, computationally efficient, and easy to implement (*Fitzgibbon et al., 1999*).

Let us denote by $(x_i, y_i)$, $i = 1, 2, \ldots, N$ the set of N points relative to the protein $\mathcal{P}$, and by $C_{\mathcal{P}}$ and $E_{\mathcal{P}}$ the adjusted implicit form of the conics obtained by the algorithms DLSFC and DLSFE respectively. The variable $C_{\mathcal{P}} = (r, x_c, y_c)$ is a vector where $r$ is the radius, and $(x_c, y_c)$ is the center of the circumference. On the other hand, the implicit form of the ellipse can be represented as $E_{\mathcal{P}} = (a, b, x_c^E, y_c^E, \theta)$, where $\{a, b\}$ are the semi-major and semi-minor axis respectively, $(x_c^E, y_c^E)$ is the center of the ellipse, and $\theta$ is the rotation angle. Under the assumption that the protein $\mathcal{P}$ can be approximate by a circumference, we expect that does not exist significative statistical differences between the radius of the circumference $(r)$ and each one of the ellipse semi-axis $(a, b)$.

**Note 8** Note that this approach is independent of which protein we choose to test the circularity hypothesis.

For each protein combination, we accomplish a Shapiro-Wilk hypothesis test to study if any of the three distributions functions (circumference radius $(r)$, semi-major axis $(a)$ and semi-minor axis $(b)$) came from a normally distributed population (*Shapiro and Wilk, 1965*). The results revealed that, our data does not seem to follow a Gaussian distribution, and as consequence, it is impossible to use a parametric test (for example t-student). Based on our data conditions, we considered the Mann-Whitney test (*Mann and Whitney, 1947*; *Hollander et al., 2013*) to evaluate the differences between the radii of the circumferences and each of the ellipses semi-axis. This is a nonparametric test that can be applied when the observations are independent (our variables are independent because the radius and the semi-axes were obtained by different algorithms), the variables are ordinal (also our variables

meet this requirement because they are numeric), and finally, we want to test if the radius of the circumferences and each one of the ellipses semi-axis, are equal (null hypothesis) or are not (alternative hypothesis).

The statistical analysis reveal that does not exist a significative statistical differences between $R_C$ and $a$ (see **Appendix 1—table 2**) with a significance level of $\alpha = 0.05$. The same results were obtained for $R_C$ and $b$ (see **Appendix 1—table 3**), and as consequence not matter if we use circumferences or ellipses to fitt the spatial distribution of the viral proteins.

On the other hand, we carried up a study to validate the viral elements concentricity hypothesis based on the distance between the centers of the adjusted circumferences of both proteins. For all the proteins, the median value is very close or smaller than $0.05\mu m$ (**Appendix 1—figure 6** and **Appendix 1—table 4** (Column 2)), which is in accordance with the resolution limit of the algorithm 3B-ODE $(0.04 - 0.05\mu m)$ (**Cox et al., 2011**). Also, the 95% confidence interval (**Appendix 1—table 4**, Column 3) shows that the distance between the centers in each case is, 95% of the times in the order of $10^{-2}\mu m$, which we consider a small error taking into account the limitations of the algorithm 3B-ODE.

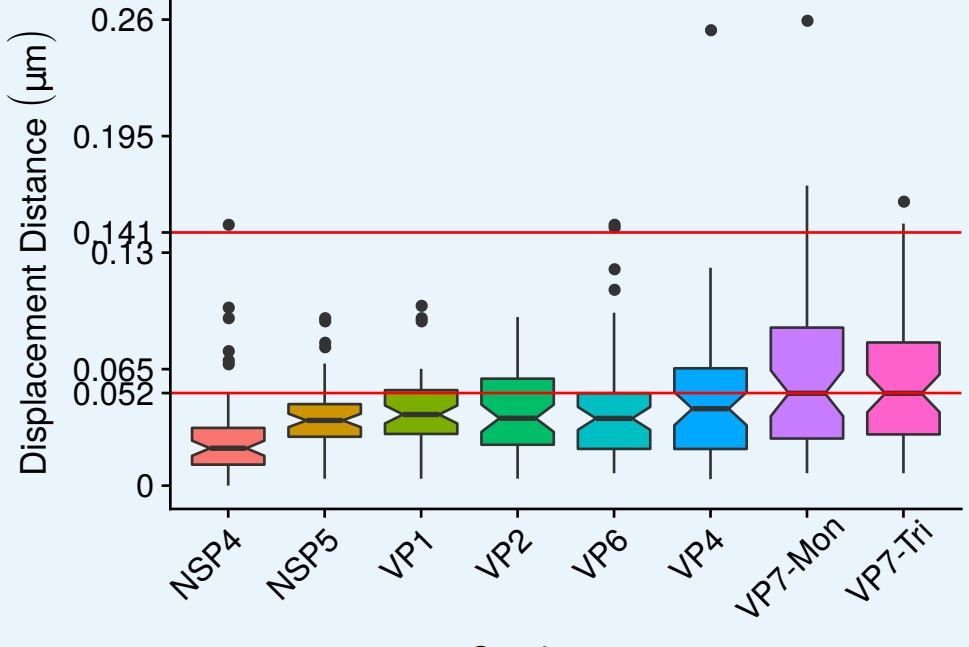

**Appendix 1—figure 6.** Boxplot of the differences in location between the center of the circumference that adjust NSP2 and the centers of the others nine viral elements. The red line $y = 0.052\mu m$ represents approximately the maximun median value of distance between NSP2 and each of the others viral elements, while the second red line $(y = (0.1, 0.1) = 0.141\mu m)$ is the maximun error that we considered in the algorithm validation section (see Appendix 1 Section –Algorithm validation–). Each box contains the 50% of the observations, the bottom and the top of the box are the first and third quartiles (25% and 75% of the observations respectively). The line inside the box is the median (second quartiles (50% of the observations)). The upper whisker extends from the hinge to the largest value no further than $1.5 \times IQR$ from the hinge (where IQR is the interquartile range, or distance between the first and third quartiles). The lower whisker extends from the hinge to the smallest value at most $1.5 \times IQR$ of the hinge. The notches around the median extend to $\pm(1.57 \times IQR/n)$. This gives a roughly 95% confidence interval for comparing medians (**McGill et al., 1978**).
DOI: https://doi.org/10.7554/eLife.42906.017

**Appendix 1—table 4.** Difference in location between the centers of NSP2 and {NSP5, NSP4, VP1, VP2, VP6, VP4, VP7-Tri, VP7-Mon}. The 95% Confidence Interval (Column 3) was computed in accordance with **Chambers et al. (1983)** as: $\hat{x} \pm (1.57 \times IQR/\sqrt{n})$, where, $\hat{x}$ is the median of $x$, IQR is the interquartile range, and $n$ is the length of $x$.

| Combination | Difference in location | 95% CI |
| --- | --- | --- |
| NSP2/NSP5 | 0.0367391 | [0.03302618, 0.04045203] |
| NSP2/NSP4 | 0.02126683 | [0.01708335, 0.02545032] |
| NSP2/VP2 | 0.03795674 | [0.03043561, 0.04547787] |
| NSP2/VP1 | 0.04006871 | [0.03508308, 0.04505433] |
| NSP2/VP6 | 0.03787043 | [0.03153263, 0.04420824] |
| NSP2/VP4 | 0.0432802 | [0.03410000, 0.05246041] |
| NSP2/VP7-Tri | 0.05184999 | [0.04138659, 0.06231340] |
| NSP2/VP7-Mon | 0.05184999 | [0.03922319, 0.06447680] |

DOI: https://doi.org/10.7554/eLife.42906.018

**Note 9** The Mann-Whitney test was used to study the differences in population medians (difference in location) (**Mann and Whitney, 1947**).

## Supplementary exploratory analysis

The algorithm VPs-DLSFC allows the quantification of the radius of two proteins in the same VP, and also the relative distance between them. At a first stage, we analyzed approximately 590 images (see histogram in **Appendix 1—figure 7A**) that were obtained by the pre-segmentation step of the algorithm VPs-DLSFC (**Appendix 1—figure 1**). In agreement with previous reports (**Eichwald et al., 2012**; **Eichwald et al., 2004**; **Carreño-Torres et al., 2010**), we found a wide heterogeneity in the size of VPs measured by NSP2, which exhibited radii ranging from 0.2 to 0.75 μm. In order to compare the distribution of each protein taking NSP2 as reference, we selected a subset of the individual VPs, in a way that there were no significant statistical difference between the radii of NSP2 from each condition.

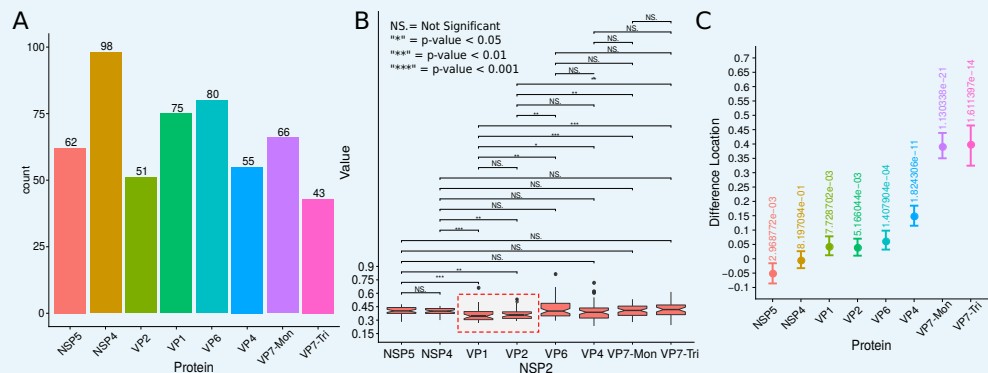

**Appendix 1—figure 7.** Supplementary exploratory analysis of the results obtained by the algorithm VPs-DLSFC. (**A**) Histogram with the numbers of VPs that were obtained as part of the pre-segmentation step of the algorithm described in **Appendix 1—figure 1A**. Note that because each image contains one and only one VP, this histogram represents also the numbers of pre-segmented images that we had in our study. (**B**) Graphical representation of the Mann-Whitney hypothesis test in two-by-two comparison between the radii distributions of NSP2 in each experiment (**Mann and Whitney, 1947**). To avoid confusions, for each box, under the NSP2 name in x-axis, we included the name of the accompanying protein that specifies in which experiment we obtain that distribution of NSP2. Each combination is linked

by a line, and the result of the test is up of the line. The red dashed square highlights the only two combinations (VP1 and VP2) in which the distribution of NSP2 had a significant statistical difference with the other NSP2 distributions. (**C**) Two-sample Mann-Whitney hypothesis test, considering as variables the radius of NSP2 in contrast to the radius of the others seven proteins. In each combination, we show the difference in location between NSP2 and the other protein (x-axis) and the confidence interval at a level of 95%. The numbers over the intervals confidence are the p-value of the statistical test.
DOI: https://doi.org/10.7554/eLife.42906.019

In accordance with this criterion, we were able to adjust the size of NSP2, such that the mean in the distribution of the radius was similar when this protein was paired with NSP5, NSP4, VP6, VP4 and VP7 (**Appendix 1—figure 7B**). Unfortunately, in the case of NSP2~VP1 and NSP2~VP2, even though we could selected a group of VPs in which the mean radius of NSP2 (0.35 μm) was closer to the size of the NSP2 from the other conditions ($0.4\mu m$), the analysis of the distributions in the radius of NSP2 by the Mann-Whitney hypothesis test (**Appendix 1—figure 7B**), revealed that the VPs of VP1 and VP2 were significative smaller than the mean size of viroplasms of the others proteins. As consequence, was not possible to compare directly the position of VP1 and VP2 with the others viral elements through an exploratory analysis. However, this problem did not limit the inferential study that we developed.

The use of super-resolution microscopy and the algorithm VPs-DLSFC allows the detection of small differences (resolution of nanometers) in the relative position of the viral elements. To validate that the different viral elements were in fact located in a different region from NSP2, we developed a Mann-Whitney test to study the existence of significative differences between the radii of NSP2 and the radii of each of the other viral elements. According with our previous results (see main text), this test revealed that only NSP4 has a similar spatial distribution that of NSP2 (**Appendix 1—figure 7C** and **Appendix 1—table 5**), while the others viral elements have different radii. Moreover, the small negative differences in location of NSP5 turned out to be significative different with respect to NSP2 (the p-values of the test are available in the (**Appendix 1—figure 7C** and **Appendix 1—table 5**). These results support that NSP5 is located in the innermost part of the viroplasm and that NSP4 shares the same layer with NSP2. The structural proteins VP1, VP2 and VP6 seem to be in the same layer of the viroplasm, with similar values, but with a significative statistical difference in location with NSP2 at a level $\alpha = 0.01$. Also, this test suggests that VP4 and VP7 form independent layers in the most external zone of the VPs (**Appendix 1—figure 7C** and **Appendix 1—table 5**), which is in accordance with our proposed model.

**Appendix 1—table 5.** Two-sample Mann-Whitney hypothesis test, considering as variables the radii of NSP2 in contrast with the radii of the others seven viral proteins. Under the null hypothesis, both samples come from the same distribution (H0: true location is equal to zero), while the alternative hypothesis (H1: true location is not equal to zero) establishes that exist a difference between the median of the distributions. Protein: Name of the viral element compared with NSP2; W: value of the Mann-Whitney statistical hypothesis test; Difference: Estimation of the location parameter (median difference between the radii of NSP2 and the radii of the other viral element); C. Interval: 95% confidence interval; P. Value: p-value of the test.

| Protein | W | Difference in location | 95% CI | p-value |
|---------|------|------------------------|--------|---------|
| NSP5 | 1327 | −0.048972049 | [−0.086040620,−0.01563190] | $2.968772 \times 10^{-3}$ |
| NSP4 | 4711 | −0.003348142 | [−0.032696239, 0.02632726] | $8.197094 \times 10^{-1}$ |
| VP1 | 1699 | 0.044239569 | [0.012214263, 0.07804365] | $7.728702 \times 10^{-3}$ |
| VP2 | 3557 | 0.040956503 | [0.010802771, 0.06990018] | $5.166044 \times 10^{-3}$ |
| VP6 | 4316 | 0.063282322 | [0.032020596, 0.09793399] | $1.407904 \times 10^{-4}$ |

*Appendix 1—table 5 continued on next page*

*Appendix 1—table 5 continued*

| Protein | W | Difference in location | 95% CI | p-value |
|---|---|---|---|---|
| VP4 | 2637 | 0.150597251 | [0.115162890, 0.18499337] | $1.824306 \times 10^{-11}$ |
| VP7-Mon | 4280 | 0.392334895 | [0.350192949, 0.43852366] | $1.130338 \times 10^{-21}$ |
| VP7-Tri | 1814 | 0.400145108 | [0.324450625, 0.46457274] | $1.611397 \times 10^{-14}$ |

DOI: https://doi.org/10.7554/eLife.42906.020

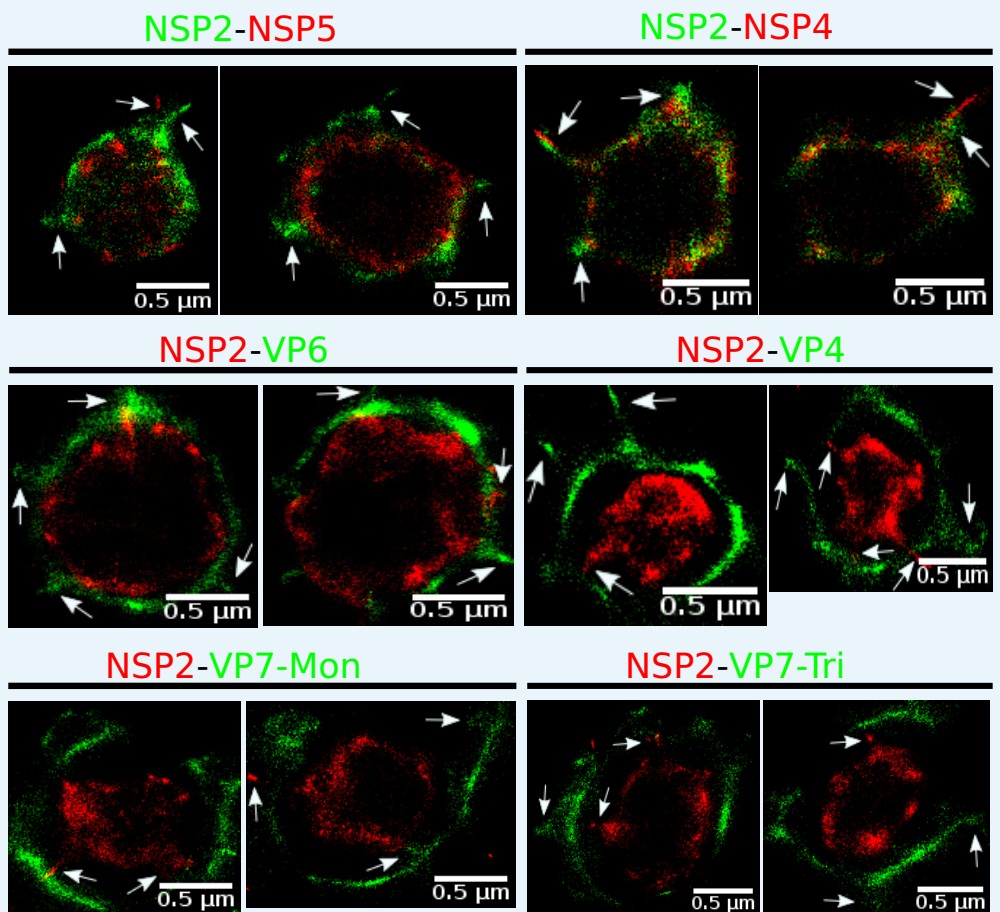

**Appendix 1—figure 8.** Examples of small zones of colocalization between differents viral proteins. The interations between the viral elements could be explained throught the spikes that come from the central distribution of the viral elements which colocalize with other proteins. A more specific study it is necessary in order to explain these interactions.
DOI: https://doi.org/10.7554/eLife.42906.022

## Linear dependency between the viral components

The spatial distribution of the viral elements in the VPs is conserved regardless of their size; that is, the radius of each protein shows a linear dependency with the radius of NSP2.

Let $P_i = (x_i, \ y_i) \in \mathbb{R}^2, \ i = 1, 2, \ldots, N$ a set of points, where $x_i$ and $y_i$ are the radii of NSP2 and the acompayning viral element Y, respectively. The adjustment of the linear model can be solved through the following optimization problem:

$$\min_{\beta} \sum_{i=1}^{N} (y_i - \beta x_i)^2. \tag{25}$$

The model (**Equation (25)**) does not include an offset term because we don't have any quantification about the radius of the accompanying viral element when NSP2 have been silenced.

The solution of this problem is given by **Kiefer (1987)**:

$$\widehat{\beta} = cor(x,y) \times \frac{\sigma(y)}{\sigma(x)}$$

$$= \frac{cov(x,y)}{\sigma(x)\sigma(y)} \times \frac{\sigma(y)}{\sigma(x)}$$

$$= \frac{\frac{1}{n}\sum(x_i - \bar{x})(y_i - \bar{y})}{Var(x)},$$

where $cor(x,y)$ is the linear correlation between x and y, $cov(x,y)$ is the covariance between x and y, Var(x) is the variance of x, and $\bar{x}$ is the mean value of the variable $X$.

The 'Residual Standard Error' (RSE or $(\sigma^2)$) is a measure of fit of the linear regression model.

$$\sigma^2 = \frac{1}{N-2} \sum_{i=1}^{N} \epsilon_i^2, \tag{27}$$

where $\epsilon_i = y_i - \widehat{\beta}x_i$. The 'Standard Error' $\left(\sigma_{\widehat{\beta}}^2\right)$ in the estimation of the slope $\widehat{\beta}$ is:

$$\sigma_{\widehat{\beta}}^2 = \frac{\sigma^2}{\sum(x_i - \bar{x})^2}. \tag{28}$$

Consider the statistic $t_\beta = \frac{\widehat{\beta} - \beta}{\sigma_{\widehat{\beta}}}$ and the null (H0) and alternative hypothesis (H1):

$$\begin{cases} H0 & : \beta = 0 \\ H1 & : \beta \neq 0 \end{cases} \tag{29}$$

Under the null hypothesis, $t_0 = \frac{\widehat{\beta}}{\sigma_{\widehat{\beta}}}$ follow a t-Student distribution with N-1 degrees of freedom. The p-value of this tests is computed as:

$$p.value = 2 \times (1 - P(T \leq t_0)), \tag{30}$$

where $P(T \leq t_0)$ is the probability that a t-Student variable with N-1 degrees of freedom be less than $t_0$. Note that through this hypothesis test it is possible to evaluate if the slope $\beta$ is significative to explain the linear relationship between the variables.

The variance of the dependent variable $y$ is:

$$\underbrace{\sum_{i=1}^{N}(y_i - \bar{y})^2}_{\text{Full Variance}} = \underbrace{\sum_{i=1}^{N}(y_i - \widehat{y}_i)^2}_{\text{Residual Variance}} + \underbrace{\sum_{i=1}^{N}(\widehat{y}_i - \bar{y})^2}_{\text{Regression Variance}}, \tag{31}$$

where, $\widehat{y}_i = \widehat{\beta}x_i$. In accordance with the **Equation (31)**, the variance rate that could be explained by the regression model is the regression variance divided by the full variance, this is:

$$R^2 = \frac{\sum(\widehat{y}_i - \bar{y})^2}{\sum(y_i - \bar{y})^2} \in [0, 1]. \tag{32}$$

Let $\Psi$ be the lineal regression model that adjusts the data $(x_i, \; y_i)$, and be $R^2$ the associated r-square value, then, $100 \times R^2$ is the percent of the data variance that it was explained by the model. For strong linear dependencies, the expected $R^2$ value should be close to 1. For more information about linear regression models consult (**Kiefer, 1987**).

The results of the regression models are available in **Appendix 1—table 6**. In all regression models, the standard errors are in the order of $10^{-2}\mu m$, with a $p - value{<}10^{-33}$ which is a strong evidence of the linear relationship that exist between NSP2 with the other seven viral proteins. Also, the RSE have values in the order of $10^{-2}\mu m$, except for VP7 which have a RSE of $0.15\mu m$, which could suggest that exist a greater dispersion of VP7 in comparison with the others viral proteins, something expected since, according to our model, VP7 is the most external protein.

**Appendix 1—table 6.** Results and validation of the linear regression models. NSP2 was considered as the independent variable and the other viral elements as the dependent variable (see **Equation (25)**). The slope and standard error for each regression model (Column 2 and 3) were computed in accordance with the **Equation (26) and (28)** respectively. The t-value of the t-Student distribution function with N-1 degrees of freedom are in Column 4 (see Appendix 1 Section – Linear Regression Model– for details). The p-value for each linear regression model are shows in Column five and were measure as was described in (**Equation (29)**) and (**Equation (30)**). Finally, the RSE and the $R^2$ summarize the adjustment of the data through the proposed lineal models. As was advised, the RSE (Column 6) is a fit measure of the linear model to the data (see **Equation (27)**). The '% Error' is the average error in the prediction and it is computed as: $100 \times RSE/\beta$, while the $100 \times R^2$ is the percent of the data variance that it is explained by the model (see **Equation (32)**).

| Model | Slope (β) | Std. error $\left(\sigma^2_\beta\right)$ | t-value | p-value | RSE | % Error | $R^2$ |
|---|---|---|---|---|---|---|---|
| $NSP5 = \beta_{NSP5} \times NSP2$ | 0.8667494 | 0.017947738 | 48.29296 | $2.550154 \times 10^{-50}$ | 0.07 | 8% | 0.974 |
| $NSP4 = \beta_{NSP4} \times NSP2$ | 0.9911517 | 0.006973701 | 142.12708 | $2.271571 \times 10^{-114}$ | 0.035 | 3.5% | 0.995 |
| $VP2 = \beta_{VP2} \times NSP2$ | 1.1243931 | 0.014817985 | 75.88030 | $6.242800 \times 10^{-72}$ | 0.04 | 3.5% | 0.987 |
| $VP1 = \beta_{VP1} \times NSP2$ | 1.1506138 | 0.019069730 | 60.33718 | $2.238867 \times 10^{-48}$ | 0.044 | 3.8% | 0.986 |
| $VP6 = \beta_{VP6} \times NSP2$ | 1.1806335 | 0.011218918 | 105.23595 | $1.087989 \times 10^{-86}$ | 0.038 | 3.2% | 0.992 |
| $VP4 = \beta_{VP4} \times NSP2$ | 1.3887083 | 0.024319983 | 57.10153 | $5.757257 \times 10^{-50}$ | 0.07 | 5% | 0.983 |
| $\text{VP7-Mon} = \beta_{\text{VP7-Mon}} \times NSP2$ | 1.9407204 | 0.040354225 | 48.09212 | $1.560719 \times 10^{-52}$ | 0.142 | 7.3% | 0.972 |
| $\text{VP7-Tri} = \beta_{\text{VP7-Tri}} \times NSP2$ | 1.9416542 | 0.054656327 | 35.52478 | $5.778910 \times 10^{-33}$ | 0.153 | 7.8% | 0.967 |

DOI: https://doi.org/10.7554/eLife.42906.021

The average errors are under 8% in all models (**Appendix 1—table 6**), which confirm the capacity of our linear models to predict with high accuracy the radius of the described viral elements. On the other hand, the percentage variance rate that could be explained by the regression models ($100 \times R^2$, see **Equation (32)**), was greater than the 96% in all cases (**Appendix 1—table 6**), demonstrating again the high linear dependency between NSP2 and the others viral elements into the viroplasm.

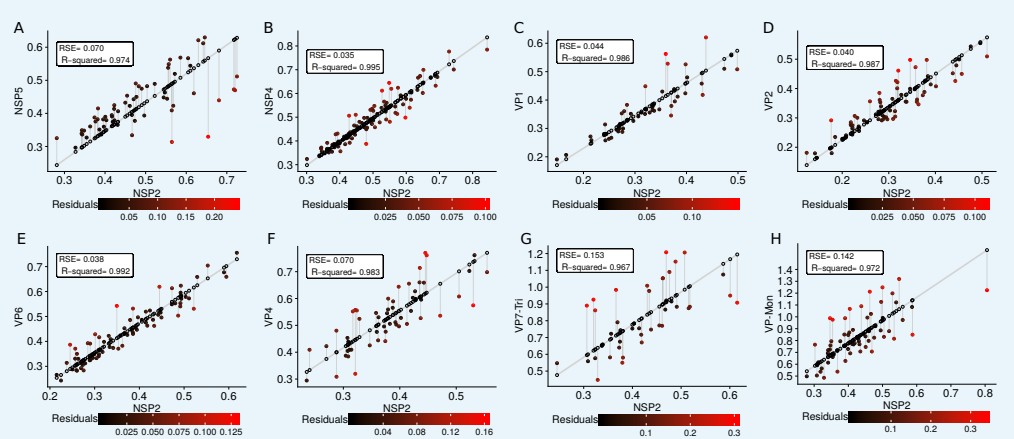

**Appendix 1—figure 9.** Residuals errors for each linear regression model. Gray line represent the regression model, the points over the line are the predicted values $\widehat{y_i}$ (see **Equation (31)**) by the models, and the dots filled with a color gradient are the real values $y_i$. The errors between the predicted and real values, $\epsilon = |\widehat{y_i} - y_i|$, are represented as a gradient of colors as follows (from lowest/coldest to highest/warmest). For each model, the RSE and $R^2$ (R-squared) were included, note that this statistics are the same that were presented in **Appendix 1—table 6**. (**A**) $NSP5 \approx 0.8667 \times NSP2$, (**B**) $NSP4 \approx 0.9912 \times NSP2$, (**C**) $VP1 = 1.1506 \times NSP2$, (**D**) $VP2 \approx 1.1244 \times NSP2$, (**E**) $VP6 \approx 1.1806 \times NSP2$, (**F**) $VP4 \approx 1.3887 \times NSP2$, (**G**) VP7-Tri $\approx 1.9417 \times NSP2$, (**H**) VP7-Mon $\approx 1.9407 \times NSP2$.
DOI: https://doi.org/10.7554/eLife.42906.023

## NSP5 and NSP4 as reference proteins

As was mentioned in the main text, we carried out a similar study using NSP5 and NSP4 as reference proteins. Since these experiments were developed to validate our previous results with NSP2, they were not as deep and extensive as in the case of the NSP2 model.

The number of VPs studied was 38 and 23 for NSP5~VP6 and NSP5~VP4 respectively (**Appendix 1—figure 10A.1**). The distribution of NSP5 relative to VP6 and VP4 did not show significative statistical differences according to the Mann-Whitney test, and the radius of VP6 was smaller than VP4, which is in accordance with our model of the viroplasm (**Appendix 1—figure 10A**). Furthermore, the distance of VP6 and VP4 towards NSP5, reflects that VP6 is closer to NSP5 than VP4, with a significance level $\leq 0.001$ (**Appendix 1—figure 10B**). On the other hand, we found that exists a linear relationship between the radii of NSP5 and {VP6, VP4} (**Appendix 1—figure 10 C-E**). The linear models of both, VP6 and VP4, explain approximately 97% of the data variability and exhibit a RSE in the order of $10^{-2} \mu m$ (see **Appendix 1—figure 10 (D,E)** and **Appendix 1—table 7**). Additionally, we performed a Mann-Whitney test to study the differences of the radii of NSP5 and each of the accompanying proteins (VP6 and VP4). This test demonstrated that the differences in location between NSP5 and VP6 is $\approx 0.1 \mu m$, while in the case of VP4 is $\approx 0.23 \mu m$, with a p-value in the order of $10^{-7}$ and $10^{-8}$ respectively (see **Appendix 1—table 8**), indicating that although these protein are near to each other, they occupy different layers on the VPs.

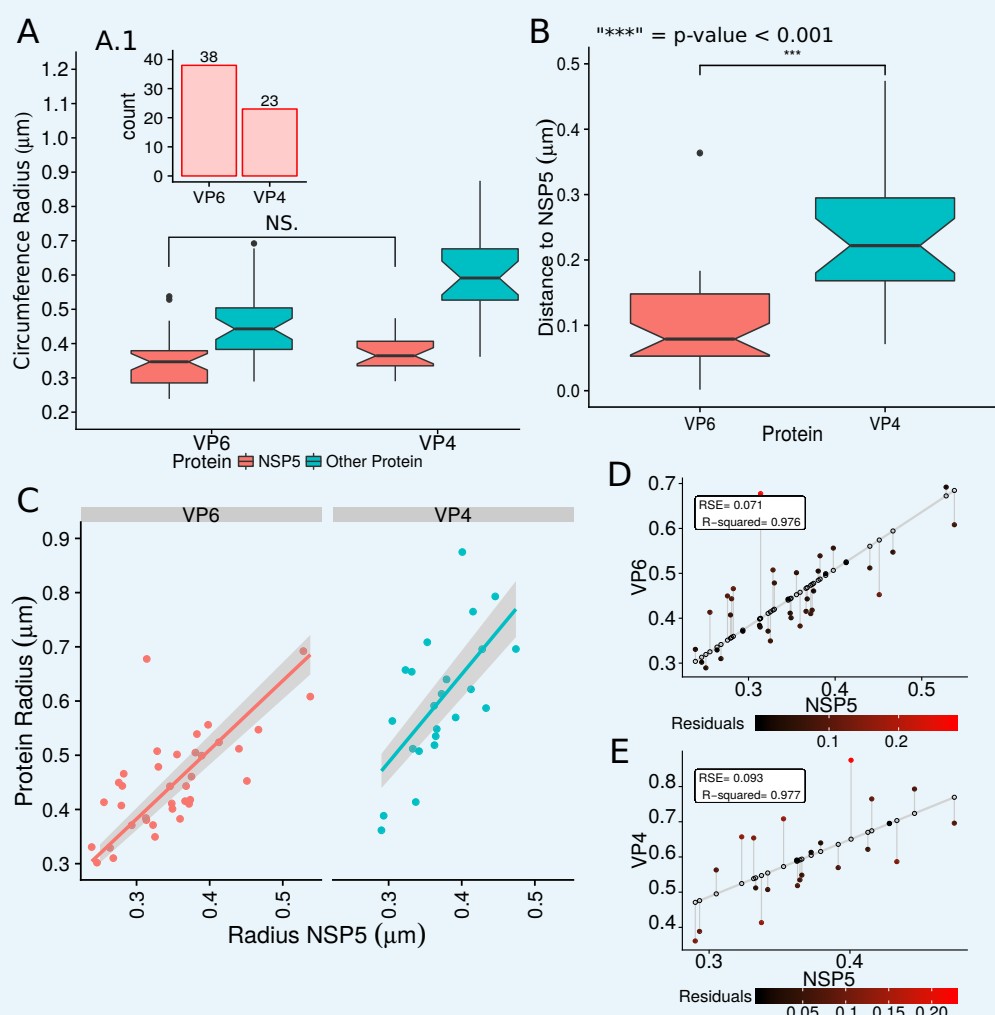

**Appendix 1—figure 10.** VP6 and VP4 spatial distribution taking NSP5 as reference protein. (**A**) Boxplot for the radii of the fitting circumferences for NSP5 and {VP6,VP4}. Mann-Whitney hypothesis test for the radius distribution of NSP5 shows that does not exist a significative statistically differences between the reference protein (NSP5) in the two experiments. A.1) Histogram of the numbers of viroplasm by combination. (**B**) Boxplot for the distance of the viral proteins VP6 and VP4 to NSP5, and result of the Mann-Whitney test between these two distributions. (**C**) Linear regression model taking the radius of NSP5 as independent variable and the radius of {VP6, VP4} as dependent variable. The gray shadow represent the 95% confidence interval for the regression adjustment. (**D and E**) Residual error for the VP6 and VP4 linear models, respectively. The details about this kind of representation can be consulted in *Appendix 1—figure 9*.
DOI: https://doi.org/10.7554/eLife.42906.024

**Appendix 1—table 7.** Linear regression results with NSP5 used as independent variable. For more information about the variables involved (columns) consult *Appendix 1—table 6* or Appendix 1 Section – Linear Regression Model.

| Model | Slope (β) | Std.error $\left(\sigma_{\hat{\beta}}^2\right)$ | t-value | p-value | RSE | R-squared |
|---|---|---|---|---|---|---|
| $VP6 = \beta_{VP6} \times NSP5$ | 1.273246 | 0.03278467 | 38.83663 | $1.364503 \times 10^{-31}$ | 0.071 | 0.976 |
| $VP4 = \beta_{VP4} \times NSP5$ | 1.623226 | 0.05248750 | 30.92596 | $1.262448 \times 10^{-19}$ | 0.093 | 0.977 |

DOI: https://doi.org/10.7554/eLife.42906.025

In the case of the model based on NSP4, we studied 60 viroplasms through the algorithm VPs-DLSFC. The results show that the radius of VP6 were bigger than the radius of NSP4 in approximately 0.04 μm (*Appendix 1—figure 11 A-B*). This difference is statistically significative ($p - value \leq 0.05$) according with the Mann-Whitney test (*Appendix 1—figure 11A*). Like the models based on NSP2 and NSP5, we found a linear relationship between NSP4 and VP6 (*Appendix 1—figure 11C*). This linear model was validated through a residual analysis (*Appendix 1—figure 11D* and *Appendix 1—table 9*), obtaining a $RSE = 0.079 \mu m$, and a $R^2 = 0.97$.

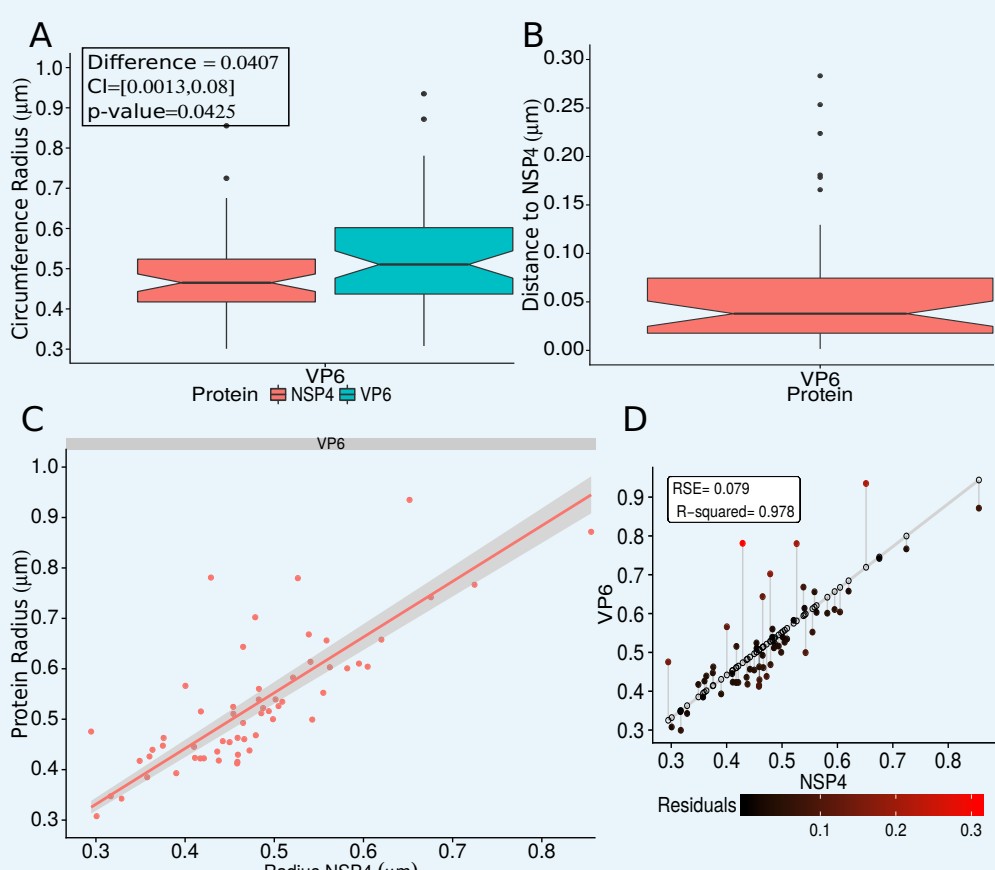

**Appendix 1—figure 11.** VP6 spatial distribution taking NSP4 as reference protein. (**A**) Boxplot for the radii of the fitting circumference of NSP4 and VP6. The inside panel shows the results of the Mann-Whitney hypothesis test (see the *Appendix 1—table 5* for details). (**B**) Boxplot for the distance between NSP4 and VP6. (**C**) Linear regression fitting taking the radius of NSP4 as independent variable and the radius of VP6 as dependent variable. The gray shadow represents the confidence interval at a level of 95%. (**D**) Residual error analysis of the model. This graph is analogous to *Appendix 1—figure 9*; the details about this kind of representation can be consulted in that figure.

DOI: https://doi.org/10.7554/eLife.42906.027

## Consistency between models

As was described, our study involves three differents reference proteins (NSP2, NSP5 and NSP4). Since the three models should respond to an unique spatial organization of the viral elements into the VP, it is expected to obtain similar results independently of the selected model.

The equivalence of the differents models (models based on NSP2 and NSP5) can be proven through a comparison of the distances between NSP5 and VP6. If the two models

describe the same distribution of the VPs, we expect that the distance of NSP5 to VP6 computed by the NSP5 model be similar to the the distance between NSP5 and NSP2, plus the distance of NSP2 to VP6 that were computed by the model based on NSP2. Hence:

$$d_{NSP2}(NSP5, NSP2) + d_{NSP2}(NSP2, VP6) \approx d_{NSP5}(NSP5, VP6) \tag{33}$$

The subindex ('name') in $d_{name}(\cdot)$ specifies the model employed to compute the distance. Therefore, replacing the corresponding values (available in **Appendix 1—table 5** and **Appendix 1—table 8**) in the above equation, we obtain that:

$$d_{NSP2}(NSP5, NSP2) + d_{NSP2}(NSP2, VP6)$$
$$= 0.048972049 + 0.063282322 = 0.1122544$$
$$\Rightarrow \varepsilon = |d_{NSP5}(NSP5, VP6) - 0.1122544|$$
$$= |0.096555 - 0.1122544|$$
$$= 0.0156994 \mu m.$$

**Appendix 1—table 8.** Two-sample Mann-Whitney hypotheses test, considering the radius of NSP5 in contrast with the radius of VP6 and VP4. For more information about the parameters (columns) see the **Appendix 1—table 5**.

| Protein | W | Difference in location | 95% | p-value |
|---|---|---|---|---|
| VP6 | 1174 | 0.096555 | [0.05986,0.13473] | $9.359731 \times 10^{-07}$ |
| VP4 | 502 | 0.227230 | [0.17693,0.28280] | $3.574209 \times 10^{-09}$ |

DOI: https://doi.org/10.7554/eLife.42906.026

Since the level of precision in our radii estimation of the viral components are limited by the resolution of the algorithm 3B ($0.04 - 0.05 \mu m$), we set this parameter as a threshold for the errors when we compared the results obtained for each model. For example, the error between the models based on NSP2 and NSP5 for VP6 is $\approx 0.016 \mu m$ (see **Equation (34)**), which is under the resolution limit that can be reached by the algorithm 3B, and as consequence the error is asociated to the limitations of the optic and not to the reference protein that we selected for the estimation of the VP6's radius.

This same approach was applied for VP4. In this case, and considering again the results in **Appendix 1—table 5** and **Appendix 1—table 8** we obtain:

$$d_{NSP2}(NSP5, NSP2) + d_{NSP2}(NSP2, VP4)$$
$$= 0.048972049 + 0.150597251 = 0.1995693$$
$$\Rightarrow \varepsilon = |d_{NSP5}(NSP5, VP4) - 0.1995693|$$
$$= |0.227230 - 0.1995693|$$
$$= 0.0276607 \mu m,$$

which again is an error under the resolution limit of the alorithm 3B.

On the other hand, taking into account the models based on NSP2 and NSP4, and the results in **Appendix 1—table 5** and **Appendix 1—figure 11A**, we obtain that:

$$d_{NSP2}(NSP4, NSP2) + d_{NSP2}(NSP2, VP6)$$
$$= 0.003348142 + 0.063282322 = 0.06663046$$
$$\Rightarrow \varepsilon = |d_{NSP4}(NSP4, VP6) - 0.06663046|$$
$$= |0.04070526 - 0.06663046|$$
$$= 0.0259252 \mu m.$$

As before, the error is under the limit of resolution of the algorithm 3B. Altogether, the comparison of the three models shows up that different experimental approaches describe the same spatial distributions of the viral elements inside the viroplasm.

The previous analysis only considers the consistency of the models based on the difference in the radius of the viral elements, but besides this, we developed a similar study using the lineal regression models.

Again, consider the model NSP5 and NSP2 as predictors of VP6:

$$VP6 = \beta_{\text{NSP2}}^{\text{VP6}} \times NSP2 \tag{37}$$

$$VP6 = \beta_{\text{NSP5}}^{\text{VP6}} \times NSP5, \tag{38}$$

where $\beta_{\text{NSP2}}^{\text{VP6}}$ and $\beta_{\text{NSP5}}^{\text{VP6}}$ are the slope associated with VP6 in the model based on NSP2 and NSP5 respectively. Also, the radius of the NSP5 can be estimated with the model based on NSP2, this is:

$$NSP5 = \beta_{\text{NSP2}}^{\text{NSP5}} \times NSP2. \tag{39}$$

Substituting *Equation (39)* into *Equation (38)* we obtain:

$$VP6 = \left(\beta_{\text{NSP5}}^{\text{VP6}} \times \beta_{\text{NSP2}}^{\text{NSP5}}\right) \times NSP2, \tag{40}$$

and taking into acount *Equation (37)*, we can assert that:

$$\beta_{\text{NSP2}}^{\text{VP6}} \approx \left(\beta_{\text{NSP5}}^{\text{VP6}} \times \beta_{\text{NSP2}}^{\text{NSP5}}\right) \tag{41}$$

$$\Rightarrow \epsilon = \left|\beta_{\text{NSP2}}^{\text{VP6}} - \left(\beta_{\text{NSP5}}^{\text{VP6}} \times \beta_{\text{NSP2}}^{\text{NSP5}}\right)\right| \quad \text{(absolute error)} \tag{42}$$

$$\Rightarrow \delta\epsilon = \frac{\left|\beta_{\text{NSP2}}^{\text{VP6}} - \left(\beta_{\text{NSP5}}^{\text{VP6}} \times \beta_{\text{NSP2}}^{\text{NSP5}}\right)\right|}{\beta_{\text{NSP2}}^{\text{VP6}}} \quad \text{(relative error).} \tag{43}$$

**Note 10** The relative error was computed under the assumption that the slope of the model based on NSP2 is the ideal value. This consideration was based in the fact that we have more evidence of a linear relationship in the NSP2 model (see the $R^2$ and RSE in *Appendix 1—table 6*) that in the models based on NSP5 and NSP4 (*Appendix 1—table 7* and *Appendix 1—table 9* respectively).

**Appendix 1—table 9.** Linear regression results with NSP4 as independent variable. For more information about the variables involved (columns) consult *Appendix 1—table 6* or Appendix 1 Section – Linear Regression Model.

| Model | Slope (β) | Std.error$\left(\sigma_{\hat{\beta}}^2\right)$ | t-value | p-value | RSE | R-squared |
|---|---|---|---|---|---|---|
| $VP6 = \beta_{VP6} \times NSP4$ | 1.103683 | 0.0212345 | 51.97592 | $5.626321 \times 10^{-51}$ | 0.079 | 0.978 |

DOI: https://doi.org/10.7554/eLife.42906.028

For the prediction of VP6 through the models NSP2 and NSP5, the relative error is (*Equation (43)*):

$$\delta\epsilon = \frac{|1.1806335 - (1.273246 \times 0.8667494)|}{1.1806335} = 0.06526013. \tag{44}$$

The relative error asociated with the prediction of the radius of VP4 through the models NSP2 and NSP5 is:

$$\delta\epsilon = \frac{|\beta_{\mathrm{NSP2}}^{\mathrm{VP4}} - (\beta_{\mathrm{NSP5}}^{\mathrm{VP4}} \times \beta_{\mathrm{NSP2}}^{\mathrm{NSP5}})|}{\beta_{\mathrm{NSP2}}^{\mathrm{VP4}}} = \frac{|1.3887083 - (1.623226 \times 0.8667494)|}{1.3887083} = 0.01312145. \quad (45)$$

Finally, in the case of VP6 considering the models NSP2 and NSP4 the relative error is:

$$\delta\epsilon = \frac{|\beta_{\mathrm{NSP2}}^{\mathrm{VP6}} - (\beta_{\mathrm{NSP4}}^{\mathrm{VP6}} \times \beta_{\mathrm{NSP2}}^{\mathrm{NSP4}})|}{\beta_{\mathrm{NSP2}}^{\mathrm{VP6}}} = \frac{1.1806335 - (1.103683 \times 0.9911517)}{1.1806335} = 0.07344889. \quad (46)$$

Since the relative errors represent the average error of the slopes computed with the models NSP5 and NSP4 in relation with the NSP2 model. These results indicate that the model based on NSP5 has an error of 6.5% for the prediction of the radius of VP6, while for VP4 the error is around 1.3% compared to the prediction of the model based on NSP2. On the other hand, if we use the model based on NSP4, we get approximately a 7.3% of differences in the radius of VP6. As in the previous analysis, these errors are small too, and could be associated with experimental variations and with the resolution limit of the algorithm 3B.

Even when the relative error provide important information about the differences in the prediction between differents models, we are not taking into account the standard error associated with the slopes of each model. In the case of the prediction of VP6 through the models NSP2 and NSP5 (all others cases have the same deduction), if we consider $\beta_{\mathrm{NSP5}}^{\mathrm{VP6}}$ and $\beta_{\mathrm{NSP2}}^{\mathrm{NSP5}}$ as random variables with a standard error $\sigma\left(\beta_{\mathrm{NSP5}}^{\mathrm{VP6}}\right)$ and $\sigma\left(\beta_{\mathrm{NSP2}}^{\mathrm{NSP5}}\right)$ respectively, the variance of the variable $\left(\beta_{\mathrm{NSP5}}^{\mathrm{VP6}} \times \beta_{\mathrm{NSP2}}^{\mathrm{NSP5}}\right)$ could be computed through the 'Delta Method' (**Oehlert, 1992**; **Ver Hoef, 2012**). The 'Delta Method' makes possible to obtain an approximation of the variance of a function $f(X_1, X_2, \ldots, X_n)$ as:

$$Var(f) = \sum_i \left(\frac{\partial f}{\partial X_i}\right)^2 Var(X_i) + \sum_i \sum_{j \neq i} \left(\frac{\partial f}{\partial X_i}\right)\left(\frac{\partial f}{\partial X_j}\right) Cov(X_i, X_j), \quad (47)$$

were $Cov(X_i, X_j)$ denote the covariance between $X_i$ and $X_j$ (**Lawrence, 1953**).

**Note 11** The standard error is defined as:

$$\mathrm{Standard\ Error} = \mathrm{Standard\ Deviation}/\sqrt{n},$$

but, in our case $n = 1$ because we only have the final slope estimation and the associated standard error, hence the standard error is equal to the standard deviation (**Altman and Bland, 2005**).

Because $\beta_{\mathrm{NSP5}}^{\mathrm{VP6}}$ and $\beta_{\mathrm{NSP2}}^{\mathrm{NSP5}}$ are independent (obtained by the linear regression analysis of the segmentation results in differents experiments and images) the covariance $Cov(\beta_{\mathrm{NSP5}}^{\mathrm{VP6}}, \beta_{\mathrm{NSP2}}^{\mathrm{NSP5}}) = 0$. This point can be proved easily because $Cov(X, Y) = \mathbf{E}(XY) - \mathbf{E}(X)\mathbf{E}(Y)$, were $\mathbf{E}(\cdot)$ is the expected value. If X and Y are independents $\mathbf{E}(XY) = \mathbf{E}(X)\mathbf{E}(Y) \Rightarrow Cov(X, Y) = 0$. Therefore, **Equation (47)** can be simplified to:

$$Var(f) = \sum_i \left(\frac{\partial f}{\partial X_i}\right)^2 Var(X_i). \quad (48)$$

Now, considering the function $f(p_1, p_2) = p_1 p_2$, we can rewrite **Equation (48)** as:

$$Var(f) = p_2^2 Var(p_1) + p_1^2 Var(p_2),$$

and substituting the variance for the standard deviation (same that the standard error in this case, see Note 11), we have:

$$Var(f) = p_2^2 \sigma^2(p_1) + p_1^2 \sigma^2(p_2). \tag{49}$$

The 99% confidence interval for $p_1 p_2$ can be computed as:

$$CI = p_1 p_2 \pm 2.576 \sqrt{Var(p_1 p_2)}. \tag{50}$$

Set $p_1 = \beta_{\text{NSP5}}^{\text{VP6}}$ and $p_2 = \beta_{\text{NSP2}}^{\text{NSP5}}$. We can compute the confidence interval of $\beta_{\text{NSP5}}^{\text{VP6}} \times \beta_{\text{NSP2}}^{\text{NSP5}}$ through **Equation (49)** and **Equation (50)**.

$$
\begin{aligned}
Var\left(\beta_{\text{NSP5}}^{\text{VP6}} \times \beta_{\text{NSP2}}^{\text{NSP5}}\right) &= \left(\beta_{\text{NSP2}}^{\text{NSP5}}\right)^2 \sigma^2(\beta_{\text{NSP5}}^{\text{VP6}}) + \left(\beta_{\text{NSP5}}^{\text{VP6}}\right)^2 \sigma^2(\beta_{\text{NSP2}}^{\text{NSP5}}) \\
&= (0.8667494)^2 (0.03278467)^2 + (1.273246)^2 (0.017947738)^2 \\
&= 0.001329683
\end{aligned}
\tag{51}
$$

$$\Rightarrow \sigma\left(\beta_{\text{NSP 5}}^{\text{VP6}} \times \beta_{\text{NSP2}}^{\text{NSP5}}\right) = \sqrt{0.001329683} = 0.03646482. \tag{52}$$

Now, considering **Equation (50)** we obtain:

$$
\begin{aligned}
CI_{\beta_{\text{NSP5}}^{\text{VP6}} \times \beta_{\text{NSP2}}^{\text{NSP5}}} &= \left(\beta_{\text{NSP5}}^{\text{VP6}} \times \beta_{\text{NSP2}}^{\text{NSP5}}\right) \pm 2.576 \left(\sigma\left(\beta_{\text{NSP5}}^{\text{VP6}} \times \beta_{\text{NSP2}}^{\text{NSP5}}\right)\right) \\
&= (1.273246 \times 0.8667494) \pm (2.576 \times 0.03646482) \\
&= [1.009652, 1.197519].
\end{aligned}
\tag{53}
$$

The slope of VP6 in the NSP2 model ($\beta_{\text{NSP2}}^{\text{VP6}}$, see **Appendix 1—table 6**) is $1.1806335 \in [1.009652, 1.197519]$, which proves that, in reference to VP6, the models based on NSP2 and NSP5 are consistent.

The rest of combinations runs as before. In the case of VP4 and the models based on NSP2 and NSP5, we obtained:

$$
\begin{aligned}
Var\left(\beta_{\text{NSP5}}^{\text{VP4}} \times \beta_{\text{NSP2}}^{\text{NSP5}}\right) &= \left(\beta_{\text{NSP2}}^{\text{NSP5}}\right)^2 \sigma^2\left(\beta_{\text{NSP5}}^{\text{VP4}}\right) + \left(\beta_{\text{NSP5}}^{\text{VP4}}\right)^2 \sigma^2\left(\beta_{\text{NSP2}}^{\text{NSP5}}\right) \\
&= (0.8667494)^2 (0.05248750)^2 + (1.623226)^2 (0.017947738)^2 \\
&= 0.002918405.
\end{aligned}
\tag{54}
$$

$$
\begin{aligned}
\Rightarrow \sigma\left(\beta_{\text{NSP 5}}^{\text{VP4}} \times \beta_{\text{NSP2}}^{\text{NSP5}}\right) &= \sqrt{0.002918405} = 0.05402226. \\
\Rightarrow CI_{\beta_{\text{NSP5}}^{\text{VP4}} \times \beta_{\text{NSP2}}^{\text{NSP5}}} &= \left(\beta_{\text{NSP5}}^{\text{VP4}} \times \beta_{\text{NSP2}}^{\text{NSP5}}\right) \pm 2.576 \left(\sigma\left(\beta_{\text{NSP5}}^{\text{VP4}} \times \beta_{\text{NSP2}}^{\text{NSP5}}\right)\right) \\
&= (1.623226 \times 0.8667494) \pm (2.576 \times 0.05402226) \\
&= [1.267769, 1.546092].
\end{aligned}
$$

Again, the slope of VP4 in the NSP2 model is $1.3887083 \in [1.267769, 1.546092]$ (see **Appendix 1—table 6**), and as consequence the models are consistent between them.

Finally, the stability between the models NSP4 and NSP2 based on the protein VP6 was evaluated.

$$
\begin{aligned}
Var\left(\beta_{\text{NSP4}}^{\text{VP6}} \times \beta_{\text{NSP2}}^{\text{NSP4}}\right) &= \left(\beta_{\text{NSP2}}^{\text{NSP4}}\right)^2 \sigma^2\left(\beta_{\text{NSP4}}^{\text{VP6}}\right) + \left(\beta_{\text{NSP4}}^{\text{VP6}}\right)^2 \sigma^2\left(\beta_{\text{NSP2}}^{\text{NSP4}}\right) \\
&= (0.9911517)^2 (0.0212345)^2 + (1.103683)^2 (0.006973701)^2 \\
&= 0.0005021999
\end{aligned}
\tag{55}
$$

$$\Rightarrow \sigma\left(\beta_{\mathrm{NSP4}}^{\mathrm{VP6}}\right) = \sqrt{0.0005021999} = 0.02240982$$

$$\Rightarrow CI_{\beta_{\mathrm{NSP4}}^{\mathrm{VP6}} \times \beta_{\mathrm{NSP2}}^{\mathrm{NSP4}}} = \left(\beta_{\mathrm{NSP4}}^{\mathrm{VP6}} \times \beta_{\mathrm{NSP2}}^{\mathrm{NSP4}}\right) \pm 2.576\left(\sigma\left(\beta_{\mathrm{NSP4}}^{\mathrm{VP6}} \times \beta_{\mathrm{NSP2}}^{\mathrm{NSP4}}\right)\right)$$

$$= (1.103683 \times 0.9911517) \pm (2.576 \times 0.02240982)$$

$$= [1.036190, 1.151645].$$

(56)

In this case the slope $\beta_{\mathrm{NSP2}}^{\mathrm{VP6}} = 1.1806335 \notin [1.036190, 1.151645]$, which is surprising taking into account that the relative error of NSP4 as predictor of VP6 instead of the model based on NSP2 is around the 7.3% (see **Equation (46)**). Even when this percent doesn't represent a great difference, it could be enough to generate small variations that cause that the slope $\beta_{\mathrm{NSP2}}^{\mathrm{VP6}}$ be out of the confidence interval given in **Equation (56)**. Moreover, note that the difference between the upper endpoint of the confidence interval of the slope between NSP2 and VP6 is approximately 0.03 $\mu$m, which is under the resolution limit of the 3B algorithm. A deeper study with different antibodies for NSP4 and the contrast of NSP4 with others viral elements would help to clarify our results.

## Stochastic model fitted for 3B super resolution microscopy.

The algorithm 3B (**Cox et al., 2011**) creates a super resolution image from experimental data (hundreds of images acquired by the microscope). The 3B algorithm generates a probability map of fluorophore location through Markov Chains and Bayesian inference. The implementation of the algorithm requires a priori information on the dynamics of the photophysical properties of the fluorophores under study; however, the software does not provide a method to calculate it. In this section we propose a method to obtain this information from the experimental data.

3B microscopy generates a probability map of fluorophore localization by deriving a weighted average over all possible models; the set of models includes varying numbers of emitters, emitter localizations and temporal dynamics of the states of the fluorophores. The information stored in the image sequence is iteratively compared with the models using Bayesian inference, producing a SRM image. This technique allows the reduction of the number of necessary images, so it reduces the photo damage in the sample. However, the necessary computation time is significant (**Hu et al., 2013**); therefore, we employed 3B parallelization techniques for a cluster and PC from **Hernández et al. (2016)** in order to decrease the time to enhance the SR images.

The dynamics of the photophysical properties of a fluorophore is modeled in 3B-algorithm as a Markov chain, where, for each time, the fluorophore can be in three posible states: the emitting state $S_0$, the non-emitting state $S_1$, and the bleached state $S_2$ (**Appendix 1—figure 12**). Hence, in the complete sequence of images a fluorophore can be transiting between these states with a probability $P_i$.

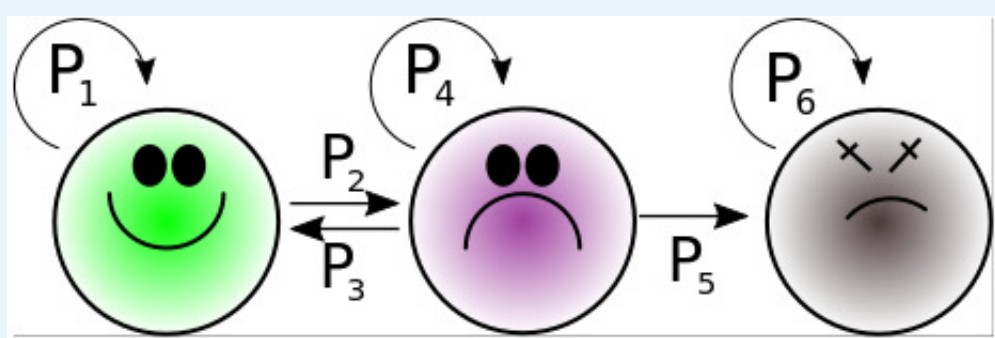

**Appendix 1—figure 12.** Diagram of probabilities for the transitions between the states of a fluorophore for the 3B-algorithm. From left to right are represented the emitting state, non-emitting state and bleached state for a fluorophore, with their respective transition probabilities among the states. As an example, a fluorophore can transit from the emission state to the non-emission state or remain in this state with a probability $P_2$ and $P_1$, respectively.

DOI: https://doi.org/10.7554/eLife.42906.029

In this way, the temporary stack of images acquired by the microscope contains information on the dynamics of an unknown number of fluorophores transiting between these three states, with a probability $P_i$. This information is represented as a transition matrix:

$$P := \begin{pmatrix} P_1 & P_2 & 0 \\ P_3 & P_4 & P_5 \\ 0 & 0 & 1 \end{pmatrix}, \tag{57}$$

where the entry in the $i$th row and $j$th column is the probability that a fluorophore in the state $S_i$ moves to the state $S_j$.

The initial probability distribution

$$\overline{\phi}_0 := (\phi_0(0), \phi_0(1), \phi_0(2)), \tag{58}$$

describes the probability of a fluorophore to be in each energy state in the beginning of the process: $\phi_0(i)$ corresponds to the state $S_i$.

The 3B algorithm considers the Markov chain determined by the transition matrix (**Equation (57)**) and the initial probability distribution (**Equation (58)**). Then,

$$\begin{aligned} \overline{\phi}_1 &= \overline{\phi}_0 P \\ \overline{\phi}_2 &= \overline{\phi}_1 P \\ &\vdots \\ \overline{\phi}_n &= \overline{\phi}_{n-1} P, \end{aligned} \tag{59}$$

are the probabilities that a fluorophore is in the states $S_0, S_1$ and $S_2$ in time $1, \ldots, n$; that is, for all $i = 1, \ldots, n$, $\overline{\phi}_i$ is the 3-vector whose entries are the probabilities that the fluorophore is in each energy state at time $i$.

The 3B algorithm has the transition matrix (**Equation (57)**) as an input, together with the initial probability distribution (**Equation (58)**). However, the software does not provide a method to calculate the transition matrix from the experimental data.

We deal with this issue by proposing a method to determine $P$ from the data. Namely, we propose to adjust the experimental data (the stack of images) with an ordinary differential equation (ODE) model that allows us to calculate the probabilities of the system.

Following the Jablonski diagram for a given fluorophore, we propose a model with three states ($S_0, S_1$ and $S_2$) and three transition probabilities ($k_{isc}, k_T$ and $k_b$) between them (**Appendix 1—figure 13**).

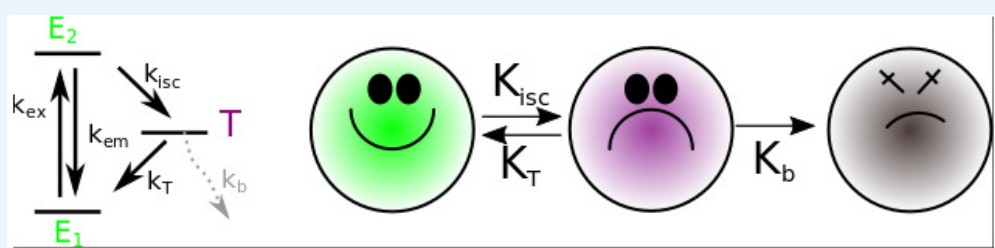

**Appendix 1—figure 13.** Left panel: The reduced Jablonski diagram for the fluorophore model (*Appendix 1—figure 12*). Right panel: Diagram of probabilities for the transitions between the states of a fluorophore for our fluorophore's model. In our model, the basal ($E_1$) and the excited ($E_2$) states, from the reduced Jablonski diagram, are collapsed in a new excited state $S_0$ (green). The justification for this lies in the fact that the fluorescence phenomenon ($k_{ex}, k_{em}$) occurs on the scale of nanoseconds or less; however, an image collected with an EM-CCD camera regularly has millisecond exposure times, therefore, it involves the integration of $105 - 106$ cycles of photon emission; as a consequence, the $E_1$ state is never detected. The entrance into the triplet excited state ($k_{isc}, k_T$) happens on the scale of seconds; if the emission process occurs, it releases photons of lower energy that are not detected and, therefore, the triplet state is considered as a dark state $S_1$ (violet). Finally, the photobleaching state $S_2$ (gray) is a irreversible process ($k_b$) culminating with the destruction of the coordinating center of resonant electrons (orbitals $\pi$) which is responsible for absorbing photons.
DOI: https://doi.org/10.7554/eLife.42906.030

For $i = 0, 1, 2$, we define the dynamics of the energy state $S_i$ as the time evolution of the concentration $[E_i]$ of molecules (fluorophores) emitting photons:

$$\frac{d[S_0]}{dt} = k_T[S_1] - k_{isc}[S_0],$$

$$\frac{d[S_1]}{dt} = k_{isc}[S_0] - (k_T + k_b)[S_1],$$

$$\frac{d[S_2]}{dt} = k_b[S_1].$$

The sequence of images taken through the time keeps track of the state the fluorophore is in. If in the $i$th image the fluorophore has intensity (the value of the pixel approaches the maximum value), then the fluorophore is in its emitting state; in any other case, the fluorophore is off, and it can be in two states: the non-emitting state or the bleached state. We normalize the data and compute the mean intensity at time $t$ of the fluorophores in the emitting state: by definition, the initial mean value is one; after that, it is going to decay depending on the acquiring protocol.

Following the classical regression methods, we fit the experimental data to the proposed differential equations system. The estimate of the rate constant $k_{isc}$, $k_T$ and $k_b$ from the stack of images is obtained following the example taken from *Rawlings and Ekerdt (2013)*.

Let us consider the matrix $P$ from *Equation (57)* as the transition matrix of our model. Then, by the definition of our model, $P_2 = k_{isc}$, $P_3 = k_T$ and $P_5 = k_b$. Also, by the general properties of the probability function, the sum of the entries of each row of this matrix is one; so we obtain the transition matrix as follows:

$$P = \begin{pmatrix} P_1 & P_2 & 0 \\ P_3 & P_4 & P_5 \\ 0 & 0 & 1 \end{pmatrix} = \begin{pmatrix} ? & k_{isc} & 0 \\ k_T & ? & k_b \\ 0 & 0 & 1 \end{pmatrix} = \begin{pmatrix} 1-k_{isc} & k_{isc} & 0 \\ k_T & 1-(k_T+k_{isc}) & k_b \\ 0 & 0 & 1 \end{pmatrix} \tag{60}$$

As we mentioned earlier, the estimate of the transition matrix depends on the acquisition protocol. There are image sequences for which the fitting does not converge; this happens when the average intensity decays exponentially. However, when the average intensity

decreases linearly, the adjustment of the data to the converging model results in a transition matrix that fits the data.

After the fitting to the experimental data we simulate a Markov chain for the calculated transition matrix (*Appendix 1—figure 14*). We also observe that greater number of fluorophores leads to better concordance between the simulation and the experimental data.

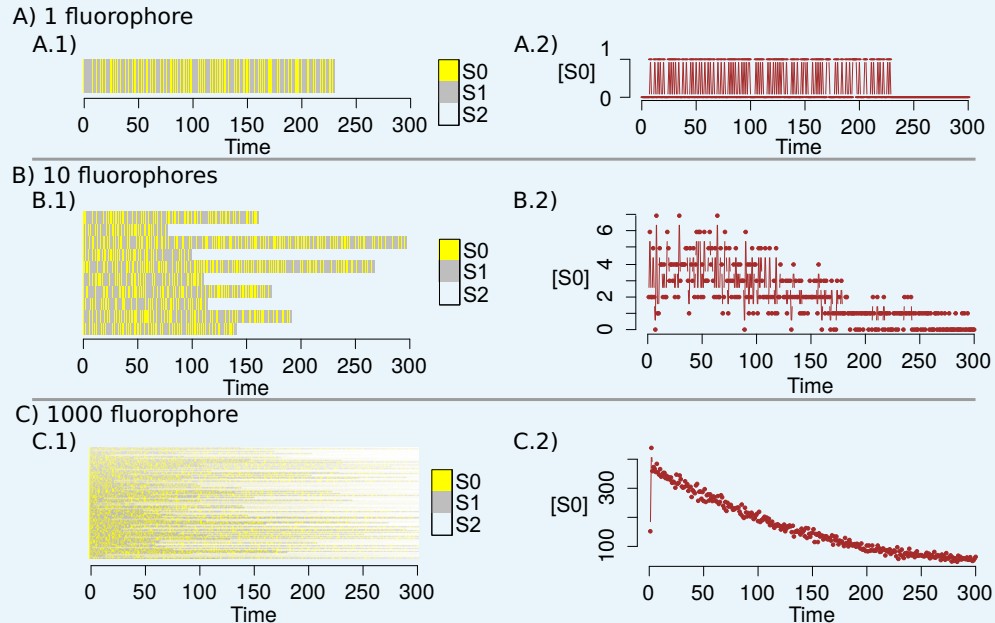

**Appendix 1—figure 14.** Markov Chain simulations with the new transition matrix. (**A–C**) shows the simulation of the dynamics for 1, 10 and 1000 fluorophores respectively. A.1, B.1 and C.1 indicates the state of each fluorophore in the time; A.2, B.2 and C.2 point out the amount of fluorophores that are in the state $S_0$ in the time.

DOI: https://doi.org/10.7554/eLife.42906.031

To corroborate that the proposed fitting is adjusted properly we use the experimental data provided by the authors of the 3B SRM microscopy algorithm. The podosomes data is available at the ThreeB plugin for ImageJ (*Rosten et al., 2013*), is a time serie of 300 images. Taking this stack as initial experimental data, we simulate Markov chains for 1000 fluorophores with the original transition matrix and the transition matrix fitted by our model. Although we found that our model fits better to the experimental data (*Appendix 1—figure 15*); the comparision between the SR images of the podosomes are not conclusive about the resolution enhancement with our model (*Appendix 1—figure 16*).

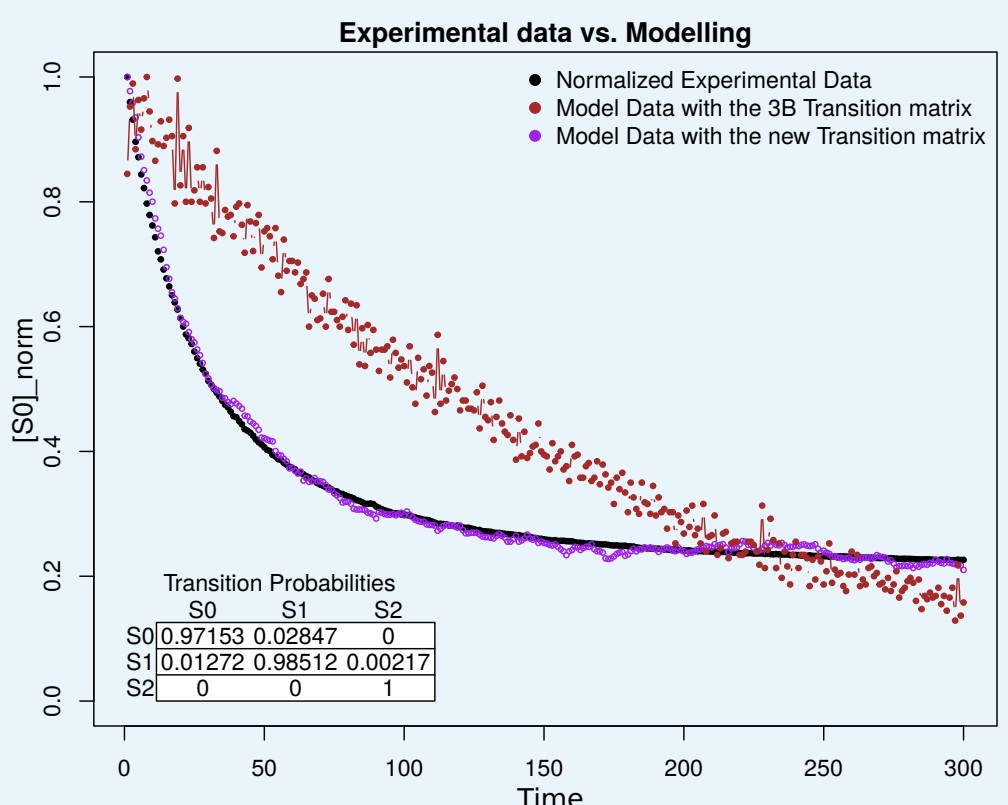

**Appendix 1—figure 15.** Simulation estimation chart of 1000 fluorophores. The $y$-axis is the normalization of fluorophores in the $S_0$ state (emission) in the time $i$ ($x$-axis). The normalized data extracted from the images are shown in black. The purple and the brown dots represent the simulation of 1000 fluorophores as a Markov chain with the transition matrix that we propose and with the original 3B matrix, respectively. The transition matrix that we propose is shown in the bottom of the figure.

DOI: https://doi.org/10.7554/eLife.42906.032

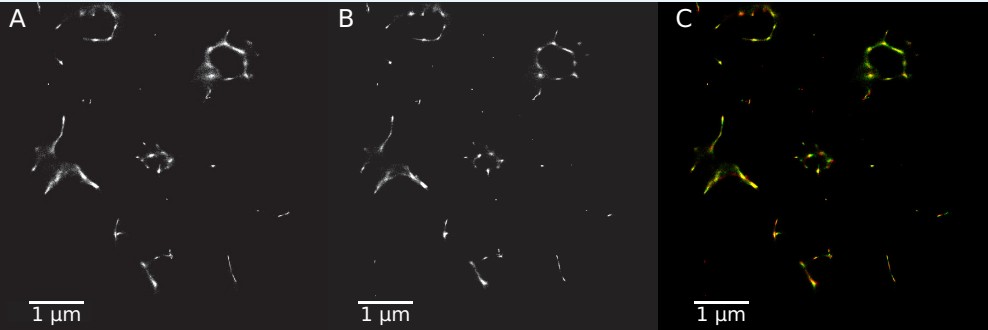

**Appendix 1—figure 16.** Super-resolution images generated with different transition matrices. (**A**) SR image using the transition matrix constructed by our model. (**B**) SR image with the original transition matrix. (**C**) Composite image comparing the two SR images in green (**A**) and in red (**B**).

DOI: https://doi.org/10.7554/eLife.42906.033

Furthermore, in order to validate our method, we use Gatta-paint nanorulers, DNA molecules label with 488, 550 and 655 fluorophores at a distance of $40nm$ (GATTA-PAINT

40RG/40B, immobilized in buffer on glass-slide) to resolve the resolution enhacement of our method.

As in the previous analysis, we find that our model fits better to the experimental data obtained from the different ATTOS. Because our model involves a greater number of fluorophores, there is more information of the fluorophores localization. For example, in the *Appendix 1—figure 17A*.1, the information in the SR image with the original matrix, resolves only one fluorophore; however, with our method, three fluorophores in the same area can be resolved. In this way, we observed that in all cases the distance between the fluorophores (each peak) of the grahps shown in panels A.1-A.3 from *Appendix 1—figure 17* is approximately 40 nm; this coincides with the information provided by the manufacturers of the samples. All together, these results demonstrate that the new proposed method improves the resolution capacity of 3B algorithm by reflecting better the photophysical behavior of the fluorophores.

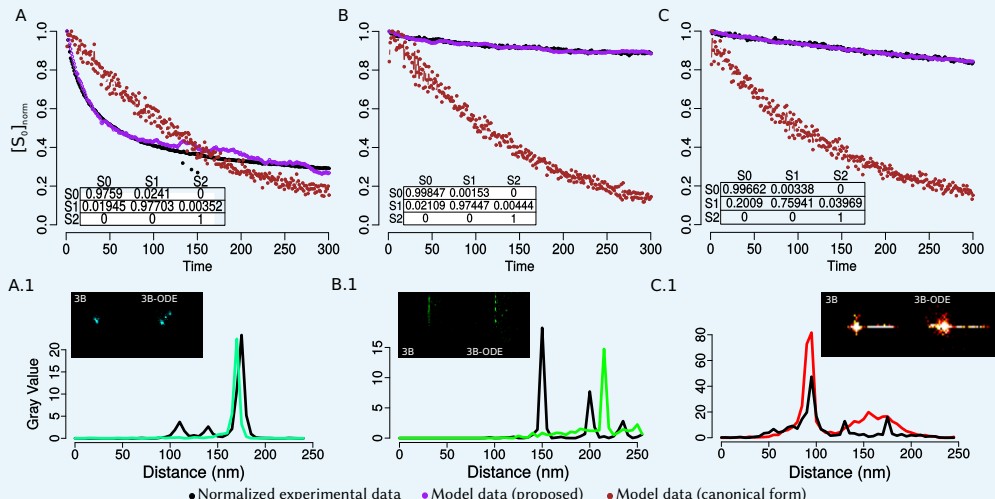

**Appendix 1—figure 17.** Comparison between the original model of 3B against the obtained with the transition matrix that we propose. Columns A-C represent the nanorulers GATTA-PAINT labelled with ATTO 488, 550 and 655 respectively. Panels A-C show the normalized experimental data (black dots) and the simulation for 1000 fluorophores with the matrix that we propose (purple dots) and with the original 3B matrix (burgundy dots); each point represents the mean value of the fluorescence. The time is indicated in frames, acquisition time between images is 100 ms. A complete experiment consists of 300 images collected in a CellTIRF microscope with a 160× magnification. Panels A.1- C.1 show two ROI's from the complete SR images: 3B) canonical reconstruction and 3B-ODE) reconstruction with the ODE's model. The graph shows a line profile from the two reconstructions, in green the results from original 3B and in black with the transition matrix that we propose, the x-label is the distance in nm and the y-label is the intensity pixel value. The peaks from the graph denote the localization of a fluorophore.
DOI: https://doi.org/10.7554/eLife.42906.034

Once the transition matrix for the nanorulers was calculated and taking into account that the data consists of hundreds of images, we parallelized the data on a cluster and on a personal computer (*Hernández et al., 2016*) (the cluster has 314 nodes with RedHat Enterprise Linux Version 6.2, 2 Intel Xeon Sandy Bridge E5-2670 processors at 2.6 GHz with 8 cores each 64 GB of RAM) to reduce the computation time.

