## [Decision Letter]

[Editors’ note: this article was originally rejected after discussions between the reviewers, but the authors were invited to resubmit after an appeal against the decision.]

Thank you for submitting your work entitled "Nanoscale organization of rotavirus replication machineries" for consideration by *eLife*. Your article has been reviewed by three peer reviewers, and the evaluation has been overseen by a Reviewing Editor and a Senior Editor. The following individuals involved in review of your submission have agreed to reveal their identity: Ulrich Desselberger (Reviewer #1); Clemens F Kaminski (Reviewer #3).

Our decision has been reached after consultation between the reviewers. Based on these discussions and the individual reviews below, we regret to inform you that your work will not be considered further for publication in *eLife*.

Your conclusions could be of considerable interest to the field, but reviewers believe that a significant number of experiments would have to be done to validate the conclusions that are reached. We feel that this additional work would require more than two months, and it is our policy to not ask for a revision in such a circumstance. However, if the points made by the reviewers in the attached reviews could be addressed, and the conclusions in the paper validated adequately, we would be open to considering such a manuscript.

*Reviewer #1:*

This manuscript describes the application of super-resolution microscopy to the analysis of the structure of viroplasms (vpls), the 'factory' for rotavirus (RV) early morphogenesis and RNA replication. It was found that within vpls 7 RV-encoded proteins and RV double-stranded (ds) RNA are organized in 6 concentric rings, with NSP5 localized at the centre, surrounded by a layer of NSP2 and NSP4, those in turn being surrounded by VP1, VP2, VP6 and viral dsRNA in an intermediate zone, and the outmost zone consisting of rings of VP4 and VP7 the latter in monomeric and trimeric form (Figure 4). From this it was concluded that the RV vpls are highly organized organelles.

The theoretical basis of the super-resolution microscopy used is explained and explored in Appendix 1, starting with mathematical details of the applied algorithm 'Viroplasm Direct least square fitting circumference', and followed by algorithm validation and model considerations. The distribution of the studied viral elements (proteins and dsRNA) as an arrangement of concentric rings was considered as proven, since the result was independent of which viral protein is used as a reference.

This manuscript is a bold proposal, which, however, in this reviewer's view contains several inconsistencies:

1) The RV NSP4 is considered to be located near the centre of the concentric ring structures. However, it was previously found to co-localize with the autophagosomal marker LC3 in cap-like structures on the outside of vpls (Berkova et al., 2006).

2) RV NSP2 octamers and NSP5 interact, and NSP5 and RV RNA compete for a binding site in the major groves of NSP2 octamers (Jiang X, et al., 2006). From the data presented it is not clear how this can happen.

3) For the localization of the viral RNA an antibody was used (SciCons J2) with apparent specificity for dsRNA. It is not clear whether it recognized the stems of stem-loop structures of viral ssRNA or the complete genomic dsRNA segments, nor that it is specific for RV dsRNA.

4) The analysis of vpl structure was carried out at 6 h post infection (pi). At this time point, vpls and infected cells can be shown to contain numerous single-layered (core) and double-layered particles which, however, are asymmetrically distributed. Since vpls appear at 2-3 h pi, those earlier structures should also be analysed.

5) For further testing of the model, authors should produce transmission electron micrographs (TEM) of sections of cells infected for 6 h. Those will show a highly asymmetric distribution of the content of vpls.

6) The authors' proposal reminded reviewer of concentric rings representing the density of RV dsRNA in the cores of particles reconstructed from cryo-EM photographs. These rings were, however, only seen when eicosahedral symmetry conditions were imposed on the image reconstruction, and not when reovirus cores were explored by cryo-EM tomography. This reviewer has the impression that the super-resolution microscopy data are based on an averaging procedure which does not recognize asymmetric structures.

7) This reviewer is not a mathematician and therefore cannot competently assess most of the content presented in the Appendix. It is strongly suggested to request the view of an alternative reviewer, who is experienced in and conversant with methods of super resolution light microscopy.

*Reviewer #2:*

In this manuscript by Garces et al., the authors investigated the nanometric-scale spatial organization rotavirus replication machinery using super-resolution microscopy. It is indeed the first quantitative visual analysis of rotavirus viroplasm components using this technique.

I have several comments that the authors need to address.

1) The authors state that the viral components form ring-like structures and are arrayed as concentric layers in Figures 1B and 1C. However, it is really hard to see the ring-like NSP2 in these figures, especially in the panels of NSP2 with VP1 and dsRNA.

2) The innermost layer of the viroplasm is composed of NSP5 that acts as the core scaffold for viroplasm formation. In the third paragraph of subsection “Quantitative characterization of VPs structure by a novel segmentation algorithm”, it is described that NSP5 is distributed inside the ring formed by NSP2 (Figure 1C). But it looks like there is NSP2 with the red NSP5 ring. And why there is nothing within the NSP5 circle? Is it hollow in the viroplasm sphere or is it that the anti-NSP5 antibody is not able to penetrate the protein layers enough to stain it?

3) It is interesting to observe the co-localization of NSP2 and NSP4, which may play a significant role in viroplasm formation and genome replication. Please give more details about how the rabbit polyclonal sera to NSP4 was generated. How was it characterized? Please give the references if it has been used in previous publications. Does it stain the NSP4 on ER?

4) Control imaging experiments should be included. For instance, an image with anti-rabbit secondary antibody in the absence of the rabbit polyclonal sera to NSP4. Does the anti-rabbit secondary antibody cross-react with the mouse polyclonal sera to NSP2?

5) The proposed model in Figure 4 show the viroplasm viral proteins as in different layers. How would this model explain the interplay between these proteins that coordinate the genome replication and virion assembly? Does the overlapping region (Figure 2) of different layers play a role in these processes?

*Reviewer #3:*

In the paper by Garces et al., the authors report on the use of an optical super resolution method to report on the nanoscale organisation of rotavirus replication machineries. The authors use the Bayesian Blinking and Bleachung (3B) super resolution method and developed a semiautomatic segmentation algorithm that analyses and quantifies the relative organisation of several key proteins in viroplasms (VP). Using optical methods featuring inherently high molecular specificity and signal to noise ratio, they are able to discriminate a number of key structural and non-structural proteins and infer their relative positions with high resolution and precision within VPs. The method exploits stochastic fluctuations in the emission characteristics of fluorophores to locate fluorophore positions at much higher precision than would be possible with standard fluorescence imaging. The nice feature of 3B is that it does not require the use of special buffers or photochemistry to enable photoswitching, which is in contrast to other super resolution methods, such as STORM. Its downside is the considerable computational effort required for analysis. Based on their analyses, the authors present a structural model for the radial structural distribution of VP components, consisting of concentric shells in which individual VP components are tightly organised. There are new findings, e.g. a suggestions of a spatially distinct location for NSP2 and NSP5 and, unexpectedly, that NSP4 is located on the inner part of VP rather than on the outside, given the fact that it is an integral membrane protein. These are nice findings, and the data appear solid. The style of writing and the presentation of data are clear. I am largely supportive of this effort and think that it could add useful information to the existing literature on rotavirus organisation. However, I do have concerns and suggestions that I think the authors must address.

1) There are no reported controls for the antibodies. Given that these are made in the laboratory and that there are no references for them in the paper, it is essential that data are included to verify their specificity. This is particularly true for the potentially controversial finding of NSP4 – the authors must exclude potential artifacts and show that binding is specific. In general, controls are lacking in the paper.

2) Most of the technical detail in the paper is in the SI and devoted to the description of the algorithms for fitting the fluorescence data and locating VP components within shell structures. Given the sensitivity of the method to the photophysical behaviour of the specific dyes used, how representative are the validation and controls presented in the SI for the AF488 antibodies?

3) More details on the chosen approach should be included. Why was the 3B method chosen? For example, could the authors have used high signal to noise (but low resolution) widefield imaging and fitted concentric shells with appropriate noise models (see e.g. https://www.cell.com/biophysj/fulltext/S0006-3495(15)00993-5)? Nanometer fitting precision has been achieved with such an approach. The method would be easily implemented for the available data, and may work very well, given the comparatively large size of the VP structures investigated. Also, the paper gives the impression that this is the first time optical super resolution methods have been used to infer virus architecture, and notable efforts with related techniques have not been cited or discussed. E.g. in Laine et al., 2015, dSTORM was used and model based fitting was used to infer the concentric arrangement of virus proteins in HSV1 with nanometer precision. There are many similarities between the efforts presented there and those presented here. What are relative advantages and disadvantages of the available methods? I encourage the authors to review the existing literature more thoroughly and to provide guidelines to the reader on what the relative merits and disadvantages are for each technique.

[Editors’ note: what now follows is the decision letter after the authors submitted for further consideration.]

Thank you for choosing to send your work entitled "Nanoscale organization of rotavirus replication machineries" for consideration at *eLife*. Your letter of appeal has been considered by a Senior Editor, a Reviewing editor, and an additional reviewer. We are prepared to consider a revised submission with no guarantees of acceptance. Our discussions about your appeal lead us to believe that it would be important to consider the following points prior to resubmission:

1) The authors have accepted that the dsRNA recognizing antibody used is non-specific (within the RNAs of RVA and beyond) and have now omitted all their dsRNA related data. RVA ssRNA-ssRNA segment interactions are at the core of pre-genome assembly in viroplasms and require interaction with NSP2. This has been convincingly described by: Borodavka et al., 2017.

The authors should comment on the processing of RVA RNAs in viroplasms in some more detail than by a single sentence in Discussion (p7, paragraph 3). The question of what is inside the NSP5 layer remains to be answered.

2) The data by Berkova et al., 2006 have now been compared with the new findings in more detail in Discussion.

3) The potential for interaction of the different viral protein layers identified has now been recognized.

4) Regarding potential structural asymmetries in viroplasms, authors provide a TEM photograph (Figure 1A) which shows an electron-dense viroplasm, but also many core particles and DLPs in the surrounding cytoplasm, proving that a first round of RVA particle formation has already taken place. See also Figure 1 in Altenburg et al., 1980, taken at 8 h p.i and figures in refs (Petrie et al., 1982, 1984; Richardson et al., 1986 and Eichwald et al, 2012). While the limits of TEM for multiple viral protein localization are recognized, it is still considered worthwhile to use the novel procedure on viroplasms at 2-3 h p.i., when viral particle formation has not yet taken place; preliminary data of this nature are not 'beyond the scope of this work' as authors suggest, but would be an important application.

5) Authors' argument that they “describe the structure of viroplasms, not virus particles” should be emphasized in Discussion.

6) Authors state that they “proved statistically that the spatial distribution of the viral proteins can be considered as concentric layers”. Although they further state that the novel observations cannot be used to understand specific details, such as the interactions between different layers, Discussion could be more outspoken about the use of the novel procedure to analyse RVA replication steps within viroplasms.

---

## [Author Response]

[Editors’ note: the author responses to the first round of peer review follow.]

Reviewer #1:[…] 1) The RV NSP4 is considered to be located near the centre of the concentric ring structures. However, it was previously found to co-localize with the autophagosomal marker LC3 in cap-like structures on the outside of vpls (Berkova et al., 2006).

It is important to keep in mind that immunofluorescence images based upon epifluorescence or confocal microscopy are inherently diffraction-limited to a spatial resolution in the range of hundreds of nanometer, with a resolution that precludes a detailed, nanoscopic molecular scale organization of VPs. Thus, in the work by Berkova et al. (1), it would be difficult to determine the exact spatial location of NSP4 in relation to other viroplasmic proteins and to the autophagosomal marker LC3. Also, in that work they used antibodies to NSP5 to identify viroplasms, and in the present work we show that NSP5 lies in a layer internal to that conformed by NSP4 and NSP2. If they had used NSP2 as the reference protein for viroplasm staining they could have found no difference in the distribution of NSP4 and NSP2, even at the level of resolution provided by confocal microscopy.

Also, it is interesting that in the paper by Berkova et al. (1) they hypothesize that there are three pools of intracellular NSP4. The first pool they propose is represented by NSP4 localized in the ER, a second minor pool of NSP4 molecules localized in the ERGIC compartment, and the third pool of NSP4 molecules distributed in cytoplasmic vesicular structures associated with the autosomal marker LC3, which appear in infected cells at 6 hpi; they propose that this third pool of NSP4 is involved in regulation of virus replication. Furthermore, and most interestingly, is their suggestion that “NSP4 and autophagic marker LC3-positive vesicles may serve as a lipid membrane scaffold for the formation of large viroplasms by recruiting early viroplasms or viroplasm-like structures formed by NSP2 and NSP5.” This observation is in line with our proposal that NSP4 lies in an internal protein layer within viroplasms.

An additional, and very interesting possibility to explain the internal location of NSP4 in viroplasms is the hypothesis that viroplasm morphogenesis occurs on the surface of lipid droplets (LDs) (see (2)). In that work, it was proposed that LDs serve as a platform to which NSP2 and NSP5 attach to form viroplasm-like structures, which in turn associate with pre-core complexes (consisting of VP1, VP3, and segmental plus RNA), VP2 and VP6. In this regard, it is important to have in mind that the currently accepted model for the LD biogenesis is that neutral lipids are synthesized between the leaflets of the ER membrane, and the mature LD is then thought to bud from the ER membrane to form an independent organelle that is contained within a limiting monolayer of phospholipids and associated LD proteins (3). Thus, during budding of the LDs from the ER membrane they could take along the rotavirus NSP4 ER membrane protein (topologically oriented towards the cell’s cytoplasm) that could help as a scaffold on the surface of LDs for the assembly of other rotavirus viroplasmic proteins, thus remaining in the interior of viroplasms.

The two possible explanations for the location of NSP4 in an internal shell of viroplasms just discussed above have been included in the Discussion section of the manuscript.

2) RV NSP2 octamers and NSP5 interact, and NSP5 and RV RNA compete for a binding site in the major groves of NSP2 octamers (Jiang X, et al., 2006). From the data presented it is not clear how this can happen.

Please, see the answer to comment #5 of reviewer #2. In short, in that answer it is stated that, based on the super resolution microscopy images, it seems clear that there is some overlapping between different protein layers, as is the case for NSP2 and NSP5 in Figure 1E. These overlapping regions between different protein layers are most likely the responsible for the previously reported NSP2-NSP5 biochemical interactions. We have clarified this point in the Discussion section of the main text (5th paragraph).

3) For the localization of the viral RNA an antibody was used (SciCons J2) with apparent specificity for dsRNA. It is not clear whether it recognized the stems of stem-loop structures of viral ssRNA or the complete genomic dsRNA segments, nor that it is specific for RV dsRNA.

We agree with the reviewer. There is no way to distinguish between viral structured ssRNA and dsRNA and the MAb is not specific for rotavirus dsRNA. A more specific and detailed study is necessary to locate the viral dsRNA within viroplasms. For these reasons, all the information related to the dsRNA location was removed from the article.

4) The analysis of vpl structure was carried out at 6 h post infection (pi). At this time point, vpls and infected cells can be shown to contain numerous single-layered (core) and double-layered particles which, however, are asymmetrically distributed. Since vpls appear at 2-3 h pi, those earlier structures should also be analysed.

We do not understand this comment. There is no connection, or at least we are proposing none, between the inner structure of viroplasms and the symmetrically or asymmetrically distribution of single-layered (core) and double-layered particles in the cell or within viroplasms. Our study restricts and describes that viroplasms have a structured organization and does not claim to describe the dynamics of virus particle morphogenesis within the viroplasm or of viroplasms themselves. Our study does not reach that far, and we believe that analysis of viroplasms at earlier times of infection is beyond the scope of this work. In this regard, the present study includes the analysis of a wide range of viroplasm sizes (see Figure 3) which follow a pretty scaling rule which is well described for a linear model. These results point to the conservation of structural organization from smaller (0.5 µm) to bigger (up to 2 μm width) viroplasms.

5) For further testing of the model, authors should produce transmission electron micrographs (TEM) of sections of cells infected for 6 h. Those will show a highly asymmetric distribution of the content of vpls.

This observation by the reviewer is connected with the previous one. Again, we are not claiming that the viroplasm is homogeneous about its content of double-layered or core particles. The morphogenetic process of formation of these particles, where it occurs within the viroplasms, and its dynamics of formation is far away from the purposes of this paper. Thus, the TEM analysis proposed will not limit or add to our conclusions; this sort of analysis has already been reported by our group (see, for instance, Figure 9 in Trejo et al. (4)). However, we agree with the reviewer that it will be useful to include a TEM image for comparison with the images obtained by immunofluorescence and super-resolution microscopy. This image is now the new Figure 1A, where it can be seen that viroplasms are electrodense structures but no organization of the viral proteins can be observed at first sight.

6) The authors' proposal reminded reviewer of concentric rings representing the density of RV dsRNA in the cores of particles reconstructed from cryo-EM photographs. These rings were, however, only seen when eicosahedral symmetry conditions were imposed on the image reconstruction, and not when reovirus cores were explored by cryo-EM tomography. This reviewer has the impression that the super-resolution microscopy data are based on an averaging procedure which does not recognize asymmetric structures.

The reviewer is correct into that the 3B-SRM algorithm provides an averaged structure, which is the average of all the models computed by the method. It is also correct that our working hypothesis is based on the concentric arrangement of viral protein rings that shape the viroplasms, which does not recognize asymmetric structures. However, please remind that we are describing the structure of viroplasms, not of viral particles. There is no indication that viroplasms follows an eicosahedral symmetry or any other symmetry whatsoever. The eicoshedral symmetry the reviewer refers to, is for viral particles.

Reviewer #2:[…] 1) The authors state that the viral components form ring-like structures and are arrayed as concentric layers in Figures 1B and 1C. However, it is really hard to see the ring-like NSP2 in these figures, especially in the panels of NSP2 with VP1 and dsRNA.

A more representative image has been included in Figure 1E for the pair NSP2-VP1. Regarding dsRNA, all the information was removed, because of the reasons explained in answer #3 to reviewer #1. Even when others authors suggested that the VPs have a spherical-like structure (5-7), we validated this hypothesis for all the viral proteins included in this study. In the section “Model Considerations” in the Appendix we proved, statistically that the spatial distribution of the viral proteins can be considered as concentric layers. Note that our intention is to provide a general characterization of the spatial distribution of the viral proteins, and not to understand specifics details of the distributions of each protein (like the interactions between different layers). Our segmentation algorithm provides a simple approach to quantify these distributions with a high robustness to noise (see the “Algorithm Validation” section in the Appendix), and also is the optimal solution that minimize the mean square error of the circumference fitting to the spatial distribution of the viral proteins (see the “Segmentation Algorithm” section in the Appendix).

2) The innermost layer of the viroplasm is composed of NSP5 that acts as the core scaffold for viroplasm formation. In the third paragraph of subsection “Quantitative characterization of VPs structure by a novel segmentation algorithm”, it is described that NSP5 is distributed inside the ring formed by NSP2 (Figure 1C). But it looks like there is NSP2 with the red NSP5 ring. And why there is nothing within the NSP5 circle? Is it hollow in the viroplasm sphere or is it that the anti-NSP5 antibody is not able to penetrate the protein layers enough to stain it?

For the first part of the question (interaction between the NSP2 and NSP5 protein rings), please see our answer to comment #5 below. Regarding the question about what is there inside the NSP5 circle, the answer is that it is not known. An interesting possibility is the hypothesis that viroplasm morphogenesis occurs on the surface of lipid droplets (2), since viroplasms co-localize with lipid droplets (2) (see also the answer to comment #1 of reviewer #1); thus, it is possible that the inner content of viroplasms could contain lipids. As suggested by the reviewer, it might also be that the anti-NSP5 antibody is not able to penetrate the protein layers enough to stain the center of the viroplasm. An additional alternative, although less likely, is that the space could contain cellular proteins, like those that have been shown to co-localize with viroplasms (see (8, 9)).

3) It is interesting to observe the co-localization of NSP2 and NSP4, which may play a significant role in viroplasm formation and genome replication. Please give more details about how the rabbit polyclonal sera to NSP4 was generated. How was it characterized? Please give the references if it has been used in previous publications. Does it stain the NSP4 on ER?

The hyperimmune serum to NSP4 (C-239) was generated by immunizing New Zealand white rabbits with a recombinant protein expressed in *E. coli* having a histidine-tail, and representing the carboxy-terminal end (amino acids 120 to 175) of the rhesus rotavirus RRV NSP4. This description, and the description of the hyperimmune serum to VP1 is now included in the Materials and methods section. Regarding the other part of the reviewer’s comment, the specificity of this anti-NSP4 serum was characterized by immunofluorescence and Western blot and RNAi (see Author response image 1 for the WB and RNAi data). This polyclonal antibody was previously used in Maruri-Avidal et al. (8), although the details of its generation were not described there either. In that paper, it was shown that NSP4 colocalizes with grp94, calreticulin, and PDI in the ER. This reference is now included also in the manuscript.

The paragraph of the antibody description now reads: “Monoclonal antibodies (MAbs) to VP2(3A8), VP4 (2G4), VP6 (255/60), VP7 (60) and VP7 (159) were kindly provided by H. B. Greenberg (Stanford University, Stanford, CA) (10, 11). The rabbit polyclonal sera to NSP2, NSP4 and NSP5, and the mouse polyclonal serum to NSP2 were produced in our laboratory (12). The hyperimmune serum to NSP4 (C-239) was generated in our laboratory by immunizing New Zealand white rabbits with a recombinant protein expressed in *E. coli* with a histidine-tail, representing the carboxy-terminal end (amino acids 120 to 175) of the rhesus rotavirus RRV NSP4 protein; see also reference (8) below, in which this serum was used. The hyperimmune serum to VP1 was also generated in our laboratory by immunizing BALB/c mice with a recombinant protein expressed in *E. coli* with a histidine-tail, representing amino acids 227 to 539 of the rhesus rotavirus RRV VP1 protein. Goat anti-mouse Alexa-488- and Goat anti-rabbit Alexa-568- conjugated secondary antibodies were purchased from Molecular Probes (Eugene, Oreg.).”

**Author response image 1. respfig1:** Western blot analysis of rotavirus RRV-infected cells previously transfected with an siRNA directed to NSP4 or with a control, irrelevant (Irre) siRNA, using the rabbit polyclonal serum (C-239) to NSP4.

4) Control imaging experiments should be included. For instance, an image with anti-rabbit secondary antibody in the absence of the rabbit polyclonal sera to NSP4. Does the anti-rabbit secondary antibody cross-react with the mouse polyclonal sera to NSP2?

Author response image 2 shows the requested control. As can be observed, there is a lack of reactivity of the anti-rabbit secondary antibody with the mouse polyclonal serum to NSP2 in the absence of the rabbit polyclonal sera to NSP4. It also shows that there is no cross-reactivity of the anti-mouse antibody with the rabbit polyclonal serum to NSP4 in the absence of the mouse polyclonal sera to NSP2. In general, the antibodies directed to different viral proteins that were used in this work have been extensively used in previous publications from our group and others (see at the end of this document -annex I- a list publications from our lab in which we have used the different antibodies), and their recognition patterns of viral proteins are well characterized. In Materials and methods the original publications that report the different antibodies are now included, or the method in our lab used for their generation is described. See also answer #3 to reviewer #2.

**Author response image 2. respfig2:** Anti-rabbit and anti-mouse secondary antibodies are specific for their target species. MA104 cells monolayers grown in coverslips were infected with RRV at an MOI of 1. At 6 hpi the cells were fixed and processed for immunofluorescence with either the primary antibody against NSP2 (green), NSP4 (red), or a combination of both antibodies, as described under Materials and methods. Afterwards, the cells were stained with either anti-rabbit Alexa568, anti-mouse Alexa488, or a combination of both antibodies. The cells nuclei (blue) were stained with DAPI.

5) The proposed model in Figure 4 show the viroplasm viral proteins as in different layers. How would this model explain the interplay between these proteins that coordinate the genome replication and virion assembly? Does the overlapping region (Figure 2) of different layers play a role in these processes?

The objective of this research was to provide a general model about the spatial distribution of the viral proteins in the viroplasm. Therefore, our model does not describe the interactions between the layers of different proteins. Nevertheless, and based on the super resolution microscopy images, it seems clear that, as suggested by the reviewer, there is some overlapping between different protein layers, as is the case for NSP2 and NSP5 in Figure 1E, but also between NSP2 and VP1, VP2, VP6 and NSP4. These general overlapping zones between different proteins most likely are the responsible for coordinating the genome replication and virion assembly, as suggested. To clarify this point we have extended the discussion on this subject in the Discussion section of the main text (5th paragraph). Also, we aggregated an extra figure (Figure S8) in the supplementary information showing the presence of protein projections (“spike-like”) from the central distribution of the viral protein layers, that could also contribute to the interaction of proteins mapped to different shells.

Reviewer #3:[…] 1) There are no reported controls for the antibodies. Given that these are made in the laboratory and that there are no references for them in the paper, it is essential that data are included to verify their specificity. This is particularly true for the potentially controversial finding of NSP4 – the authors must exclude potential artifacts and show that binding is specific. In general, controls are lacking in the paper.

Please, see our answers to comments #3 and #4 of reviewer #2.

2) Most of the technical detail in the paper is in the SI and devoted to the description of the algorithms for fitting the fluorescence data and locating VP components within shell structures. Given the sensitivity of the method to the photophysical behaviour of the specific dyes used, how representative are the validation and controls presented in the SI for the AF488 antibodies?

This answer has two parts, one related to the improvements of the calculation performed by 3B, and the other related to the validation of the algorithm developed to segment and analyze vpls.

First, we agree with reviewer that the outcome of 3B might be influenced by the photophysical properties of the fluorophore in study, as 3B requires as an input a P matrix which contains the transition probabilities between fluorophore’s states. This is exactly the rationale for computing P directly from the experimental data. In the Appendix, we show to the reader how to compute P directly from the experimental data, and provide some examples with distinct ATTO dyes, sowing that using such method there is a slight improvement on the resolution provide by 3B. We agree with the reviewer that the photophysical properties of ATTO and Alexa dyes are distinct, which might limit the calculations of P with our method (3B-ODE). However, we are happy to state that we have tested 3B-ODE with distinct dyes, including Alexa, fluorescein derivates and other dyes, that is DAPI, Hoechst, Sir-Actin, FM-464. All of them give a better result than by the analysis obtained with the use of 3B without computing P directly form the experimental data. As an example, we provide Author response image 3, which shows the fluorescence dynamics, as average fluorescence signal at each time step, of one viroplasm stained with AF488 antibodies (black trace).

**Author response image 3. respfig3:** 

The result of modeling the fluorescence dynamics after computing P directly from the experimental data (purple trace), note the similarity of both traces. Compare with the brown trace, a model of the fluorescence dynamics using the values used by 3B after choosing the default option within the ImageJ implementation of the method (13).

Second, the algorithm validation was performed using synthetic ground truth images (more than 40000 in this study) generated considering different levels of noise and partial occlusion angles. These images contain circumferences of different sizes and positions, for which real parameters of their implicit equations are known. This facilitates the analysis of the algorithm’s response against several plausible events, hence permitting the comparison of the obtained results with “ground-truth” circumferences that simulated the VPs. For the analysis of the sensitivity of the algorithm with specific antibodies (like AF488), it is necessary to know the ground truth segmentation result of the VPs in real images, information that is not available in the viral images repository we have (note that we are talking about super resolution microscopy images). For this reason, in this work we used synthetic images and evaluated the algorithm performance with different levels of noise, which can be considered as coarse-grained simulation of the photon yields of different dyes. Our method proved to be robust in high noise conditions, and when the level of “signal contamination” is greater than the expected in an SRM image. We obtained nanometric errors through our segmentation algorithm (under 0.17 μm in all the cases, except for the occlusion angle 3pi/2, which is an extreme partial occlusion angle, where only remains ¼ of the information of the synthetic VPs).

3) More details on the chosen approach should be included. Why was the 3B method chosen? For example, could the authors have used high signal to noise (but low resolution) widefield imaging and fitted concentric shells with appropriate noise models (see e.g. https://www.cell.com/biophysj/fulltext/S0006-3495(15)00993-5)? Nanometer fitting precision has been achieved with such an approach. The method would be easily implemented for the available data, and may work very well, given the comparatively large size of the VP structures investigated. Also, the paper gives the impression that this is the first time optical super resolution methods have been used to infer virus architecture, and notable efforts with related techniques have not been cited or discussed. E.g. in Laine et al., 2015, dSTORM was used and model based fitting was used to infer the concentric arrangement of virus proteins in HSV1 with nanometer precision. There are many similarities between the efforts presented there and those presented here. What are relative advantages and disadvantages of the available methods? I encourage the authors to review the existing literature more thoroughly and to provide guidelines to the reader on what the relative merits and disadvantages are for each technique.

At the very preliminary steps of the present research we had no idea about any degree of organization within the rotavirus viroplasms. The primary purpose of this research was to scrutinize the internal architecture of the viroplasms, by studying their protein distribution through super-resolution optical microscopy. Previous florescence/confocal microscopy studies showed that the viroplasm is composed of various structural and non-structural viral proteins, however due to the limit of the resolution of those techniques (250 nm approx) the viroplasm was observed as a structure without apparent organization of the proteins that make it up (14-16). Although viroplasms have been observed by electron microscopy the characteristics of this technique limit the level of detail that can be observed, particularly they become challenging to assign a particular degree of organization to a particular collection of molecules, so only electron-dense structures are observed without any clear degree of organization (see answer to question 5 of reviewer #1). We started by performing optical sectioning of viroplasms through total internal reflection microscopy (TIRFM), which allows to discriminate information provided by optical sections thinner than the length of the point spread function. In other words, TIRFM allows optical sectioning at 100 – 500 nm, hence overcoming the diffraction limit in Z axis. By using TIRFM, we started to observe certain degree of organization within the viroplasms (see Figure 1B of this article). That is why we proposed using super-resolution microscopy to determine if such a conformational organization exists. We chose 3B, over other super-resolution approaches because this method allow to deal samples with high density of labeling, providing data with a reasonable resolution, at the cost of higher computational effort.

We agree with the reviewer that we can use other methods to study the organization of cellular structures showing concentric arrangements as those proposed by Laine et al. (17), and Manetsberger et al. (18), which might not necessarily need to be tailored on the use of super-resolution microscopy. The similarities seen between those studies and our approach is the use of conics such as circles or ellipses to study structures showing concentric organization. The present investigation started in 2014, prior to the publication of the manuscripts recommended by this reviewer and was inspired by an algorithm which we previously published for the automatic detection and measurement of viral replication compartments by ellipse adjustment on fluorescence images (18). We agree with the reviewer that the approach provided by Manetsberger et al. can be implemented to analyze our data set, which as outcome will provide similar results. However, the approach suggested by the reviewer can provide something we had not paid attention in the present research (i.e. for simplicity reasons) which is the study of the degree of asymmetry within the viroplasms, something which we believe might be valuable to establish functional relationships between the protein distribution belts that shape these intriguing structures. We have now included in the Discussion section (first paragraph) part of information provided to the reviewer to answer this comment.

In the same vein of thinking, we want to stress to reviewer that our study represent the first approach which address the nanoscopic organization of viroplasms. The focus of the present research is not the study of viral particles. Considering that, we do not understand the rationale to think that the organization of viral particles will be mirrored form the viroplasm architecture.

[Editors’ note: the author responses to the re-review follow.]

We are prepared to consider a revised submission with no guarantees of acceptance. Our discussions about your appeal lead us to believe that it would be important to consider the following points prior to resubmission:1) The authors have accepted that the dsRNA recognizing antibody used is non-specific (within the RNAs of RVA and beyond) and have now omitted all their dsRNA related data. RVA ssRNA-ssRNA segment interactions are at the core of pre-genome assembly in viroplasms and require interaction with NSP2. This has been convincingly described by: Borodavka et al., 2017.The authors should comment on the processing of RVA RNAs in viroplasms in some more detail than by a single sentence in Discussion (p7, paragraph 3). The question of what is inside the NSP5 layer remains to be answered.

As requested, we have discussed in more detail the assembly of the rotavirus assorted +RNA complex containing NSP2, VP1 and VP3 into cores. As stated by the reviewer, the question of what is inside the NSP5 layer remains indeed to be answered.

2) The data by Berkova et al., 2006 have now been compared with the new findings in more detail in Discussion.

As suggested, we integrated into the discussion the previously reported data by Berkova et al., 2006.

3) The potential for interaction of the different viral protein layers identified has now been recognized.

This is correct. See subsection “The structural organization of VPs is independent of the reference protein chosen for pairwise comparison”.

4) Regarding potential structural asymmetries in viroplasms, authors provide a TEM photograph (Figure 1A) which shows an electron-dense viroplasm, but also many core particles and DLPs in the surrounding cytoplasm, proving that a first round of RVA particle formation has already taken place. See also Figure 1 in Altenburg et al., 1980, taken at 8 h p.i and figures in refs (Petrie et al., 1982, 1984; Richardson et al., 1986 and Eichwald et al, 2012). While the limits of TEM for multiple viral protein localization are recognized, it is still considered worthwhile to use the novel procedure on viroplasms at 2-3 h p.i., when viral particle formation has not yet taken place; preliminary data of this nature are not 'beyond the scope of this work' as authors suggest, but would be an important application.

As requested, VPs were analyzed by SRM at 3hpi. This analysis was carried out using the Super-Resolution Radial Fluctuations (SRRF) algorithm (Nat. Commun. 7:12471, 2016; see below for its description). In Author response image 4, it is clearly seen that the viral proteins in the VP show a similar “layered” organization as shown for the mature VP at 6 hpi in the manuscript. This observation indicates that this organization is already present when viral particle formation has not yet taken place, suggesting that it might be important for the assembly of DLPs within VPs. Of interest, at 3 hpi the layer formed by the NSP4 protein does not coincide (at the resolution achieved with the SRRF analysis) with the NSP2 layer, as it was found by 3B-SRM in VPs at 6 hpi, but it lies slightly outside the NSP2 layer, strongly suggesting that the organization of the proteins within VPs is dynamic and can change during maturation of the VPs. Further analysis carrying out a detailed characterization of the organization of the viral proteins in VPs a different times post-infection and using the 3B-SRM conditions described in the manuscript should be carried out to confirm these preliminary observations.

Due to problems beyond our control, we could not use the clusters or supercomputers of our university to generate the 3B-SRM images; as a consequence, we decided to use SRRF even when the resolution of the resulting images is less than in the case of 3B-SRM. The images generated with SRRF (Author response image 4), however, allow us to show, as a proof of concept, that at 3hpi a layered organization in the distribution of the viral proteins already exists. The experimental design for this analysis was the same as that described for 6hpi.

In the SRRF algorithm, for each image in a sequential series of frames (image’s stack), it magnifies each pixel of the image into subpixels and uses the degree of local gradient convergence (radial symmetries) to compute the probability that a subpixel contains a fluorophore. The previous step generates a stack of radiality maps using the spatial information of each image, and therefore it is possible to process these radiality maps through a higher-order temporal analysis to generate a single SRM image (see Nat. Commun. 7:12471, 2016 for details). Like 3B-SRM, SRRF allows the creation of SR images without the necessity for special prepared samples or for a specific microscope system.

**Author response image 4. respfig4:** The VPs of rotavirus present a ring-like structure at early times of infection. MA104 cells grown in coverslips were infected with RRV (MOI of 3). At 3 hours post-infection, the cells were fixed and co-inmunostained with the indicate antibodies. The nanoscale distribution of viral proteins within the VP was then analyzed through SRRF-SRM. Scale bar is 0.5 um.

5) Authors' argument that they “describe the structure of viroplasms, not virus particles” should be emphasized in Discussion.

The fact that we describe the structure of viroplasms, not virus particles, in our work, has been emphasized in the Discussion section.

6) Authors state that they “proved statistically that the spatial distribution of the viral proteins can be considered as concentric layers”. Although they further state that the novel observations cannot be used to understand specific details, such as the interactions between different layers, Discussion could be more outspoken about the use of the novel procedure to analyse RVA replication steps within viroplasms.

As requested, we have extended the discussion regarding the potential use of the SRM procedure described, to analyze replication steps within viroplasms beyond the determination of the organization of viral proteins within these structures.

References

1) Z. Berkova, SE. Crawford, G. Trugnan, T. Yoshimori, AP. Morris, and MK. Estes. Rotavirus NSP4 induces a novel vesicular compartment regulated by calcium and associated with viroplasms. Journal of Virology, 80(12):6061–6071, 2006.

2) W. Cheung, M. Gill, A. Esposito, CF. Kaminski, N. Courousse, S. Chwetzoff, G. Trugnan, N. Keshavan, and U. Desselberger. Rotaviruses associate with cellular lipid droplet components to replicate in viroplasms, and compounds disrupting or blocking lipid droplets inhibit viroplasm formation and viral replication. Journal of Virology, 84(13):6782–6798, 2010.

3) TC. Walther and RV. Farese. Lipid droplets and cellular lipid metabolism. Annual Review of Biochemistry, 81(1):687–714, 2012.

4) Ó. Trejo-Cerro, C. Eichwald, EM. Schraner, D. Silva-Ayala, S. López, CF. Arias. Actin-Dependent Nonlytic Rotavirus Exit and Infectious Virus Morphogenetic Pathway in Nonpolarized Cells. Journal of Virology, 92, e02076-17.

5) C. Eichwald, JF. Rodriguez, and OR. Burrone. Characterization of rotavirus NSP2/NSP5 interactions and the dynamics of viroplasm formation. Journal of General Virology, 85(3):625–634, 2004.

6) C. Cabral-Romero and L. Padilla-Noriega. Association of rotavirus viroplasms with microtubules through NSP2 and NSP5. Mem. Inst. Oswaldo Cruz [online], 101(6), 2006.

7) M. Campagna, L. Marcos-Villar, F. Arnoldi, CCH. Felipe, PP. Gallego, JC. González-Santamaría, DI. González, F. Lopitz-Otsoa, MS. Rodríguez, OR. Burrone, and C. Rivas. Rotavirus viroplasm proteins interact with the cellular SUMOylation system: implications for viroplasm-like structure formation. Journal of virology, 87(2):807–817, 2013.

8) L. Maruri-Avidal, S. López, and CF. Arias. Endoplasmic reticulum chaperones are involved in the morphogenesis of rotavirus infectious particles. Journal of Virology, 82(11):5368–5380, 2008.

9) P. Dhillon, VN. Tandra, SG. Chorghade, ND. Namsa, L. Sahoo, and CD. Rao. Cytoplasmic re-localization and colocalization with viroplasms of host cell proteins, and their role in rotavirus infection. Journal of Virology, 92(15), 2018.

10) RD. Shaw, PT. Vo, PA. Offit, BS. Coulsont, and HB. Greenberg. Antigenic mapping of the surface proteins of rhesus rotavirus. Virology, 155(2):434 – 451, 1986.

11) HB. Greenberg, V. McAuliffe, J. Valdesuso, R. Wyatt, J. Flores, A. Kalica, Y. Hoshino, and N. Singh. Serological analysis of the subgroup protein of rotavirus, using monoclonal antibodies. Infection and Immunity, 39(1):91–99, 1983.

12) RA. González, MA. Torres-Vega, S. López, and CF. Arias. in vivo interactions among rotavirus nonstructural proteins. Archives of Virology, 143(5):981–996, 1998.

13) E. Rosten, G. Jones, and S. Cox. Imagej plug-in for bayesian analysis of blinking and bleaching. Nature Methods, 10(2):97–98, 2013.

14) MK. Estes and HB. Greenberg. Fields Virology, chapter Rotaviruses, pages 1347–1401. Wolters Kluwer Health/Lippincott Williams & Wilkins, Philadelphia, PA, 6th edition, 2013.

15) A. Navarro, L. Williamson, M. Angel, and JT. Patton. Chapter 2.3 – rotavirus replication and reverse genetics. In Viral Gastroenteritis, pages 121 – 143. Academic Press, Boston, 2016.

16) SD. Trask, SM. McDonald, and JT. Patton. Structural Insights into the Coupling of Virion Assembly and Rotavirus Replication. Nature Reviews Microbiology. 10(3): 165–177, 2012.

17) RF. Laine, A. Albecka, S. Van de Linde, EJ. Rees, CM. Crump, and CF. Kaminski. Structural analysis of herpes simplex virus by optical super-resolution imaging. Nature Communications, 6(5980), 2015.

18) J. Manetsberger, JD. Manton, MJ. Erdelyi, H. Lin, D. Rees, G. Christie, and EJ. Rees. Ellipsoid localization microscopy infers the size and order of protein layers in bacillus spore coats. Biophysical Journal, 109(10), 2015.